# One-shot Conditional Sampling: MMD meets Nearest Neighbors

**Anirban Chatterjee** [* 1]   **Sayantan Choudhury** [* † 2]   **Rohan Hore** [* 3]

## Abstract

How can we generate samples from a conditional distribution that we never fully observe? This question arises across a broad range of applications in both modern machine learning and classical statistics, including image post-processing in computer vision, approximate posterior sampling in simulation-based inference, and conditional distribution modeling in complex data settings. In such settings, compared with unconditional sampling, additional feature information can be leveraged to enable more adaptive and efficient sampling. Building on this, we introduce Conditional Generator using MMD (CG-MMD), a novel framework for conditional sampling. Unlike many contemporary approaches, our method frames the training objective as a simple, adversary-free direct minimization problem. A key feature of CGMMD is its ability to produce conditional samples in a single forward pass of the generator, enabling practical one-shot sampling with low test-time complexity. We establish rigorous theoretical bounds on the loss incurred when sampling from the CGMMD sampler, and prove convergence of the estimated distribution to the true conditional distribution. In the process, we also develop a uniform concentration result for nearest-neighbor based functionals, which may be of independent interest. Finally, we show that CGMMD performs competitively on synthetic tasks involving complex conditional densities, as well as on practical applications such as image denoising and image super-resolution.

## 1. Introduction

A fundamental problem in statistics and machine learning is to model the relationship between a response $Y \in \mathcal{Y}$ and a predictor $X \in \mathcal{X}$. Classical regression methods (Hastie et al., 2009; Koenker & Bassett, 1978), typically summarize this relationship through summary statistics, which are often insufficient for many downstream tasks that require the knowledge of the entire conditional law. Access to the full conditional distribution enables quantification of uncertainty associated with prediction (Castillo & Randrianarisoa, 2022), uncovers latent structure (Mimno et al., 2015), supports dimension reduction (Reich et al., 2011), and graphical modeling (Chen et al., 2026). In modern scientific applications, it provides a foundation for simulation-based inference (Cranmer et al., 2020) across various domains, including computer vision (Gupta et al., 2024), neuroscience (von Krause et al., 2022), and the physical sciences (Hou et al., 2024; Mastandrea et al., 2024).

Classical approaches such as distributional regression and conditional density estimation (Rosenblatt, 1969; Fan et al., 1996; Hothorn et al., 2014) model the full conditional distribution directly but often rely on strong assumptions and offer limited flexibility. In contrast, recent advances in generative models like Generative Adversarial Networks (GANs) (Zhou et al., 2023; Mirza & Osindero, 2014; Odena et al., 2017), Variational Autoencoders (VAEs) (Harvey et al., 2022; Doersch, 2016; Mishra et al., 2018), and diffusion models (Rombach et al., 2022; Saharia et al., 2022; Zhan et al., 2025) provide more flexible, assumption lean alternatives for conditional distribution learning across applications in vision, language, and scientific simulation. A more detailed discussion of related work, background, and connections to simulation-based inference is provided in Section A.

GANs, introduced by Goodfellow et al. (2014) as a two-player minimax game optimizing the Jensen–Shannon divergence (Fuglede & Topsoe, 2004), are a widely adopted class of generative models, known for their flexibility and empirical success. However, training remains delicate and unstable, even in the unconditional setting (Arjovsky & Bottou, 2017; Salimans et al., 2016). As Arjovsky & Bottou (2017) point out, the generator and target distributions often lie on low-dimensional manifolds that do not intersect, ren-

---

*Equal contribution . [1]Department of Statistics, University of Chicago, Chicago, USA. [2]Department of Statistics and Data Science, MBZUAI, Abu Dhabi, UAE. [3]Department of Statistics and Data Science, Carnegie Mellon University, Pittsburgh, USA. [†]Part of this work was done while this author was at Johns Hopkins University, USA. Correspondence to: Anirban Chatterjee <anirbanc@uchicago.edu>.

*Proceedings of the 43rd International Conference on Machine Learning*, Seoul, South Korea. PMLR 306, 2026. Copyright 2026 by the author(s).

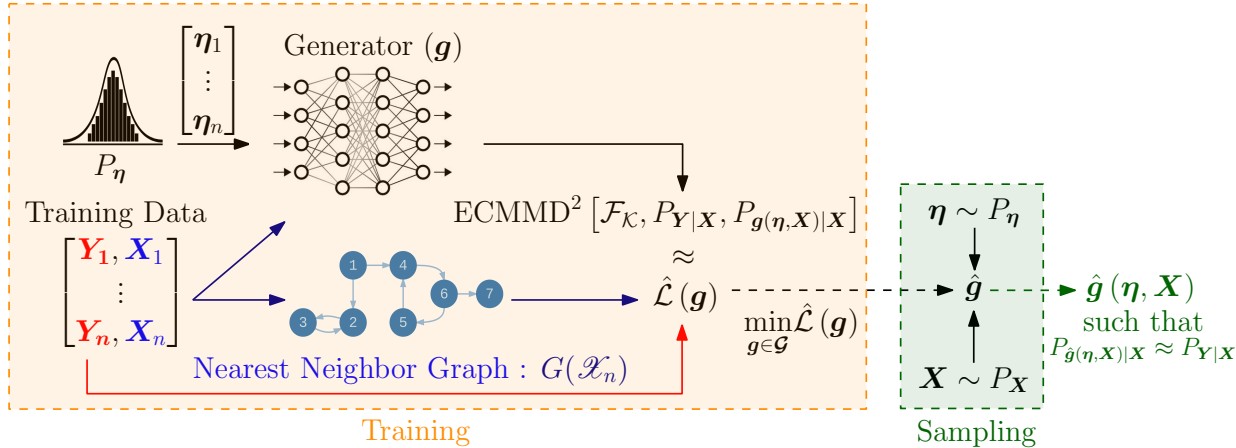

*Figure 1.* Schematic overview of CGMMD: Given training data $(\boldsymbol{Y}_1, \boldsymbol{X}_1), \ldots, (\boldsymbol{Y}_n, \boldsymbol{X}_n)$, the samples $\mathscr{X}_n = \{\boldsymbol{X}_1, \ldots, \boldsymbol{X}_n\}$ and auxiliary noise $\boldsymbol{\eta}_1, \ldots, \boldsymbol{\eta}_n$ are passed through the generator $\boldsymbol{g}$ to produce samples $\boldsymbol{g}(\boldsymbol{\eta}_1, \boldsymbol{X}_1), \ldots, \boldsymbol{g}(\boldsymbol{\eta}_n, \boldsymbol{X}_n)$. These outputs are compared with the observed $\boldsymbol{Y}_1, \ldots, \boldsymbol{Y}_n$ values using a nearest-neighbor ($G(\mathscr{X}_n)$) based estimate of the ECMMD discrepancy (see (1.2)) between true and generated conditional distributions. Edges are color-coded to highlight the dependence of each section on the corresponding inputs. After training, sampling is immediate: for any new input $\boldsymbol{X}$, independently generate new $\boldsymbol{\eta} \sim P_{\boldsymbol{\eta}}$, the trained model $\hat{\boldsymbol{g}}$ then produces $\hat{\boldsymbol{g}}(\boldsymbol{\eta}, \boldsymbol{X})$ as the conditional output. Each component is described in greater details in Section 2 and Section 3.

dering divergences like Jensen–Shannon or KL constant or infinite and thus providing no useful gradient. To address this, alternative objectives based on Integral Probability Metrics (IPMs) (Müller, 1997), such as the Wasserstein distance (Villani, 2009) and Maximum Mean Discrepancy (MMD) (Gretton et al., 2012), have been proposed for more stable training in unconditional sampling using GANs.

Building on the success of MMD-GANs (Li et al., 2015; Dziugaite et al., 2015; Bińkowski et al., 2018; Huang et al., 2022b), we propose an MMD-based loss using nearest neighbors to quantify discrepancies between conditional distributions. While MMD has been used in conditional generation, to the best of our knowledge we are the first to provide sharp theoretical guarantees for MMD based conditional sampling, offering a principled foundation for training conditional generators. Initially developed for two-sample testing by Gretton et al. (2012), MMD has since seen broad adoption across the statistical literature (Gretton et al., 2007; Fukumizu et al., 2007; Chwialkowski et al., 2016; Sutherland et al., 2017). It quantifies the discrepancy between two probability distributions as the maximum difference in expectations over functions $f$ drawn from the unit ball of a Reproducing Kernel Hilbert Space (RKHS) defined on $\mathcal{Y}$ (Aronszajn, 1950). Formally, let $\mathcal{Y}$ be a separable metric space equipped with $\mathcal{B}_{\mathcal{Y}}$, the sigma-algebra generated by the open sets of $\mathcal{Y}$. Let $\mathcal{P}(\mathcal{Y})$ be the collection of all probability measures on $(\mathcal{Y}, \mathcal{B}_{\mathcal{Y}})$. Then for any $P_{\boldsymbol{Y}}, P_{\boldsymbol{Z}} \in \mathcal{P}(\mathcal{Y})$,

$$\mathrm{MMD}(\mathcal{F}_{\mathcal{K}}, P_{\boldsymbol{Y}}, P_{\boldsymbol{Z}}) = \sup_{f \in \mathcal{F}_{\mathcal{K}}} \mathbb{E}[f(\boldsymbol{Y})] - \mathbb{E}[f(\boldsymbol{Z})], \quad (1.1)$$

where $\mathcal{F}_{\mathcal{K}}$ is the unit ball of a reproducing kernel Hilbert space (RKHS) $\mathcal{K}$ on $\mathcal{Y}$.

## 1.1. Conditional Generator using Maximum Mean Discrepancy (CGMMD)

To extend MMD to the conditional setting, we employ the expected conditional MMD (ECMMD) from Chatterjee et al. (2024) (also see (Huang et al., 2022b)), which naturally generalizes the MMD distance to a discrepancy between conditional distributions. Formally, for $\boldsymbol{X} \sim P_{\boldsymbol{X}}$, conditional distributions $P_{\boldsymbol{Y}|\boldsymbol{X}}$ and $P_{\boldsymbol{Z}|\boldsymbol{X}}$ supported on $\mathcal{Y}$, the squared ECMMD can be defined as,

$$\mathrm{ECMMD}^2(\mathcal{F}_{\mathcal{K}}, P_{\boldsymbol{Y}|\boldsymbol{X}}, P_{\boldsymbol{Z}|\boldsymbol{X}})$$
$$:= \mathbb{E}_{\boldsymbol{X} \sim P_{\boldsymbol{X}}}\left[\mathrm{MMD}^2(\mathcal{F}_{\mathcal{K}}, P_{\boldsymbol{Y}|\boldsymbol{X}}, P_{\boldsymbol{Z}|\boldsymbol{X}})\right]. \quad (1.2)$$

We discuss simplified formulations of this measure later in Section 2.1. By Chatterjee et al. (2024, Proposition 2.3), ECMMD is indeed a strict scoring rule, meaning that $\mathrm{ECMMD}^2(\mathcal{F}_{\mathcal{K}}, P_{\boldsymbol{Y}|\boldsymbol{X}}, P_{\boldsymbol{Z}|\boldsymbol{X}}) = 0$ if and only if $P_{\boldsymbol{Y}|\boldsymbol{X}} = P_{\boldsymbol{Z}|\boldsymbol{X}}$ almost surely. This property establishes ECMMD as a principled and reliable tool for comparing conditional distributions.

Instead of estimating the target conditional distribution $P_{\boldsymbol{Y}|\boldsymbol{X}}$ directly, we follow the generative approach from Zhou et al. (2023) and Song et al. (2025). By the noise outsourcing lemma (see Lemma 2.2), the problem of nonparametric conditional density estimation can be reformulated as a generalized nonparametric regression problem. In particular, for a given predictor value $\boldsymbol{X} = \boldsymbol{x}$, our goal is to learn a conditional generator $\boldsymbol{g}(\boldsymbol{\eta}, \boldsymbol{x})$, where $\boldsymbol{\eta}$ is drawn from a simple reference distribution (e.g., Gaussian or uniform). The generator is trained so that $\boldsymbol{g}(\boldsymbol{\eta}, \boldsymbol{x})$ approximates the conditional distribution of $\boldsymbol{Y} \mid \boldsymbol{X} = \boldsymbol{x}$ for all $\boldsymbol{x}$. Discrepancy between the true conditional distribution $P_{\boldsymbol{Y}|\boldsymbol{X}}$ and

the model distribution $P_{g(\eta, X)|X}$ is measured using the squared ECMMD. Once training is complete, conditional sampling becomes a one-shot procedure: draw $\eta$ from the reference distribution and sample $g(\eta, x)$. In this way, the generator provides an explicit and efficient representation of the conditional distribution of $Y \mid X$. We refer to $g(\eta, x)$ as the Conditional Generator using Maximum Mean Discrepancy, or CGMMD for short. We provide the schematic overview of the method in Figure 1. Now, we turn to the main contributions of our proposed method.

### 1.2. Main Contributions

Our main contributions are summarized below.

- **Direct Minimization.** Similar to MMD-GANs in the unconditional setting, CGMMD avoids adversarial min-max optimization and instead enables direct minimization of an ECMMD based objective (see (1.1)), offering a more straightforward and tractable alternative to GAN-based training (Zhou et al., 2023; Song et al., 2025; Ramesh et al., 2022). This design helps avoid common issues in conditional GANs, such as mode collapse and unstable min–max dynamics.

- **One-shot Sampling.** While diffusion models have demonstrated remarkable success in generating high-quality and diverse samples, their iterative denoising procedure (Ho et al., 2020) makes sampling computationally expensive and time-consuming. In contrast, CGMMD enables efficient one-shot sampling, i.e., conditional samples are obtained in a single forward pass of the generator. Specifically, to sample from $Y \mid X = x$, one simply draws $\eta$ from a simple reference distribution (e.g., Gaussian or uniform) and evaluates $\hat{g}(\eta, x)$, where $\hat{g}$ is a solution of (3.2).

- **Theoretical Guarantees.** We provide rigorous theoretical guarantees for CGMMD. Theorem 4.4 gives a non-asymptotic finite-sample bound on the error of the conditional sampler $\hat{g}(\eta, x)$, and Corollary 4.5 establishes convergence to the true conditional distribution as the sample size increases. Together, these results provide strong theoretical justification for CGMMD.

  To the best of our knowledge, this is the first application of tools from uniform concentration of nonlinear functionals, nearest neighbor methods, and generalization theory to conditional generative modeling. In the process, we also establish a general uniform concentration result for a broad class of nearest-neighbor-based functionals (Appendix G), which may be of independent interest.

- **Numerical Experiments.** Finally, we provide experiments on both synthetic and real data (mainly in image post-processing tasks) to evaluate the performance of CGMMD and compare it with existing approaches in the literature. Overall, our proposed approach performs reliably across different settings and often matches or exceeds the alternative approaches in more challenging cases.

## 2. Technical Background

In this section, we introduce the necessary concepts and previous works required to understand our proposed framework, CGMMD. To that end, we begin with the necessary formalism.

Let $\mathcal{X}, \mathcal{Y}$ be Polish spaces, that is, complete separable metric spaces equipped with the corresponding Borel-sigma algebras $\mathcal{B}_{\mathcal{X}}$ and $\mathcal{B}_{\mathcal{Y}}$ respectively. Let $\mathcal{P}(\mathcal{X})$ and $\mathcal{P}(\mathcal{Y})$ be the collection of all probability measures defined on $(\mathcal{X}, \mathcal{B}_{\mathcal{X}})$ and $(\mathcal{Y}, \mathcal{B}_{\mathcal{Y}})$ respectively. Recalling the RKHS $\mathcal{K}$ defined on $\mathcal{Y}$ from (1.1), the Riesz representation theorem (Reed & Simon, 1980, Therorem II.4) guarantees the existence of a positive definite kernel $\mathsf{K} : \mathcal{Y} \times \mathcal{Y} \to \mathbb{R}$ such that for every $y \in \mathcal{Y}$, the feature map $\phi_y \in \mathcal{K}$ satisfies $\mathsf{K}(y, \cdot) = \phi_y(\cdot)$ and $\mathsf{K}(y_1, y_2) = \langle \phi_{y_1}, \phi_{y_2} \rangle_{\mathcal{K}}$.

The definition of feature maps can now be extended to embed any distribution $P \in \mathcal{P}(\mathcal{Y})$ into $\mathcal{K}$. In particular, for $P \in \mathcal{P}(\mathcal{Y})$ we can define the kernel mean embedding $\mu_P$ as $\langle f, \mu_P \rangle_{\mathcal{K}} = \mathbb{E}_{Y \sim P}[f(Y)]$. Moreover, by the canonical form of the feature maps, it follows that $\mu_P(t) := \mathbb{E}_{Y \sim P}[\mathsf{K}(Y, t)]$ for all $t \in \mathcal{Y}$. Henceforth, we make the following assumptions on kernel $\mathsf{K}$.

**Assumption 2.1.** The kernel $\mathsf{K} : \mathcal{Y} \times \mathcal{Y} \to \mathbb{R}$ is positive definite and satisfies the following:

1. The kernel $\mathsf{K}$ is bounded, that is $\|\mathsf{K}\|_\infty < K$ for some $K > 0$ and Lipschitz continuous.

2. The kernel mean embedding $\mu : \mathcal{P}(\mathcal{Y}) \to \mathcal{K}$ is a one-to-one (injective) function. This is also known as the *characteristic kernel* property (Sriperumbudur et al., 2011).

Assumption 2.1 ensures that the mean embedding $\mu_P \in \mathcal{K}$ (see Lemma 3 in Gretton et al. (2012) and Lemma 2.1 in Park & Muandet (2020)), and that MMD defines a metric on $\mathcal{P}(\mathcal{Y})$. While these properties can be guaranteed under weaker conditions on the kernel $\mathsf{K}$, we adopt the above assumption for technical convenience. With the above notations the MMD (recall (1.1)) can be equivalently expressed as $\mathrm{MMD}^2(\mathcal{F}_{\mathcal{K}}, P_Y, P_Z) = \|\mu_{P_Y} - \mu_{P_Z}\|_{\mathcal{K}}^2$ (see Lemma 4 from Gretton et al. (2012)) where $\| \cdot \|_{\mathcal{K}}$ is the norm induced by the inner product $\langle \cdot, \cdot \rangle_{\mathcal{K}}$. In the following, we express the ECMMD in an equivalent form and leverage it to obtain a consistent empirical estimator.

### 2.1. ECMMD: Representation via Kernel Embeddings

Recalling the definition of ECMMD from (1.2), we note that it admits an equivalent formulation. In particular, for

distributions $P_{Y|X}$ and $P_{Z|X}$ (which exists by Klenke (2008, Theorem 8.37)), define the conditional mean embeddings $\mu_{P_{Y|X}}(t) := \mathbb{E}[\mathsf{K}(Y, t) \mid X]$ and $\mu_{P_{Z|X}}(t) := \mathbb{E}[\mathsf{K}(Z, t) \mid X]$ for all $t \in \mathcal{Y}$. Under Assumption 2.1, the conditional mean embeddings are indeed well defined by Park & Muandet (2020, Lemma 3.2). Consequently, $\|\mu_{P_{Y|X=x}} - \mu_{P_{Z|X=x}}\|_{\mathcal{K}}^2$ is the squared MMD metric between the conditional distributions for a particular value of $X = x$. Averaging this quantity over the marginal distribution of $X$ yields the squared ECMMD distance:

$$\mathrm{ECMMD}^2(\mathcal{F}_{\mathcal{K}}, P_{Y|X}, P_{Z|X})$$
$$= \mathbb{E}_{X \sim P_X}\left[\|\mu_{P_{Y|X}} - \mu_{P_{Z|X}}\|_{\mathcal{K}}^2\right] \quad (2.1)$$

However, to use ECMMD as a loss function for estimating the conditional sampler, we require a consistent estimator of the expression in (2.1). To that end, the well-known *kernel trick* enables a more tractable reformulation of ECMMD, making it amenable to estimation from observed data. By Chatterjee et al. (2024, Proposition 2.4) (also see Huang et al. (2022b) and Park & Muandet (2020)), the squared ECMMD admits the tractable form

$$\mathrm{ECMMD}^2(\mathcal{F}_{\mathcal{K}}, P_{Y|X}, P_{Z|X}) = \mathbb{E}\left[\mathsf{H}(W, W')\right] \quad (2.2)$$

where $W = (Y, Z), W' = (Y', Z')$, $\mathsf{H}(W, W') = \mathsf{K}(Y, Y') + \mathsf{K}(Z, Z') - \mathsf{K}(Y, Z') - \mathsf{K}(Z, Y')$ and $(Y, Y', Z, Z', X)$ is generated by first sampling $X \sim P_X$, then drawing $(Y, Z)$ and $(Y', Z')$ independently from $P_{Y|X} \times P_{Z|X}$. Note that when $Y, Z$ are independent of $X$, the expression from (2.2) is equivalent to the classical expression of squared MMD (Gretton et al., 2012).

## 2.2. ECMMD: Consistent Estimation using Nearest Neighbors

Towards estimating the ECMMD, we leverage the equivalent expression from (2.2). By the tower property of conditional expectations, (2.2) can be further expanded as,

$$\mathrm{ECMMD}^2(\mathcal{F}_{\mathcal{K}}, P_{Y|X}, P_{Z|X}) = \mathbb{E}\left[\mathbb{E}\left[\mathsf{H}(W, W') \mid X\right]\right].$$

To estimate ECMMD, we observe that it involves averaging a conditional expectation over the distribution $P_X$. Given observed samples $\{(Y_i, Z_i, X_i) : 1 \leq i \leq n\}$ drawn from the joint distribution $P_{YZX} = P_{Y|X} \times P_{Z|X} \times P_X$, we proceed by first estimating the inner conditional expectation given $X = X_i$, and then averaging these estimates over the observed values $X_1, \ldots, X_n$. To estimate the inner conditional expectation given $X = X_i$, one can, in principle, average the inner function over sample indices whose corresponding predictors are 'close' to $X_i$. A natural way to quantify such proximity is through nearest-neighbor graphs. Formally we construct the estimated ECMMD as follows.

Fix $k = k_n \geq 1$ and let $G(\mathscr{X}_n)$ be the directed $k-$nearest neighbor graph on $\mathscr{X}_n =$

$\{X_1, \ldots, X_n\}$. Moreover let $N_{G(\mathscr{X}_n)}(i) := \{j \in [n] : X_i \to X_j \text{ is an edge in } G(\mathscr{X}_n)\}$ for all $i \in [n]$. Now the $k-$NN based estimator of ECMMD can be defined as,

$$\widehat{\mathrm{ECMMD}}^2\left(\mathcal{F}_{\mathcal{K}}, P_{Y|X}, P_{Z|X}\right)$$
$$:= \frac{1}{n} \sum_{i=1}^{n} \frac{1}{k_n} \sum_{j \in N_{G(\mathscr{X}_n)}(i)} \mathsf{H}(W_i, W_j) \quad (2.3)$$

where $W_i = (Y_i, Z_i)$ for all $i \in [n]$ and $\mathsf{H}(W_i, W_j) = \mathsf{K}(Y_i, Y_j) - \mathsf{K}(Y_i, Z_j) - \mathsf{K}(Z_i, Y_j) + \mathsf{K}(Z_i, Z_j)$ for all $1 \leq i, j \leq n$. Chatterjee et al. (2024, Theorem 3.2) shows that under mild conditions, this estimator is consistent for the oracle ECMMD. We exploit this nearest-neighbor construction to define the CGMMD objective in Section 3.

## 2.3. Generative Representation of Conditional Distribution

As outlined in Section 1.1, conditional density estimation can be reformulated as a generalized nonparametric regression problem. Suppose $(Y, X) \in \mathcal{X} \times \mathcal{Y}$ follows some joint distribution $P_{YX}$, and we observe $n$ independent samples $\{(Y_1, X_1), \ldots, (Y_n, X_n)\}$ from $P_{YX}$. Our goal is to generate samples from the unknown conditional distribution $P_{Y|X}$. The *noise outsourcing lemma* (see Kallenberg (2021, Theorem 5.10), Zhou et al. (2023, Lemma 2.1) and Bloem-Reddy & Teh (2020, Lemma 5)) formally connects conditional distribution estimation with conditional sample generation. For completeness, we state it below.

**Lemma 2.2** (Noise Outsourcing Lemma)**.** *Suppose $(Y, X) \sim P_{YX}$. Then, for any $m \geq 1$, there exist a random vector $\eta \sim P_\eta = \mathrm{N}(\mathbf{0}_m, \mathbf{I}_m)$ and a Borel-measurable function $\bar{g} : \mathbb{R}^m \times \mathcal{X} \to \mathcal{Y}$ such that $\eta$ is generated independent of $X$ and $(Y, X) = (\bar{g}(\eta, X), X)$ almost surely.*

Under appropriate conditions the above result also follows from Brenier's theorem (Villani, 2003, Theorem 3.8). Moreover, by Zhou et al. (2023, Lemma 2.2), $(Y, X) \stackrel{d}{=} (\bar{g}(\eta, X), X)$ if and only if $\bar{g}(\eta, x) \sim P_{Y|X=x}$ for every $x \in \mathcal{X}$. This identifies $\bar{g}$ as a conditional generator. Consequently, to draw from $P_{Y|X}$, we sample $\eta \sim \mathrm{N}(\mathbf{0}_m, \mathbf{I}_m)$ and output $\bar{g}(\eta, X)$.

This perspective places conditional density estimation firmly within the realm of generative modeling. The task reduces to: given $n$ independent samples from $P_{YX}$, learn the conditional generator $\bar{g}$. (Zhou et al., 2023; Ramesh et al., 2022; Song et al., 2025; Liu et al., 2021) leveraged this idea to develop a GAN-based (respectively Wasserstein-GAN) framework for conditional sampling. In contrast, our approach follows a similar path but replaces the potentially unstable min–max optimization of GANs with a principled minimization objective based on ECMMD discrepancy. The precise formulation is given in the following section.

# 3. ECMMD Based Objective for CGMMD

Building on the generative representation of conditional distributions and the ECMMD discrepancy introduced earlier, our goal is to learn a conditional generator $\bar{g}$ by minimizing the ECMMD distance between the true conditional distribution $Y \mid X$ and the generated conditional distribution $\bar{g}(\eta, X) \mid X$. We restrict our attention to a parameterized function class $\mathcal{G}$, as solving this unconstrained minimization problem over all measurable functions is intractable. To that end, we begin by defining the population objective

$$
\begin{aligned}
\mathcal{L}(\boldsymbol{g}) :&= \text{ECMMD}^2 \left[ \mathcal{F}_{\mathcal{K}}, P_{Y|X}, P_{g(\eta, X)|X} \right] \\
&= \mathbb{E}_{X \sim P_X} \left[ \| \mu_{P_{Y|X}} - \mu_{P_{g(\eta, X)|X}} \|_{\mathcal{K}}^2 \right].
\end{aligned}
$$

The target generator is then given by $g^\star \in \arg\min_{g \in \mathcal{G}} \mathcal{L}(g)$. Since the oracle objective $\mathcal{L}(\cdot)$ is not directly available, we employ the estimation strategy outlined in Section 2.2 to construct a consistent empirical approximation of $\mathcal{L}(g)$. Given $n$ independent samples $(Y_1, X_1), \ldots, (Y_n, X_n) \sim P_{YX}$ and independent draws of noise variables $\eta_1 \ldots, \eta_n \sim P_\eta$, we define the empirical objective,

$$
\begin{aligned}
\hat{\mathcal{L}}(\boldsymbol{g}) :&= \widehat{\text{ECMMD}}^2 \left( \mathcal{F}_{\mathcal{K}}, P_{Y|X}, P_{g(\eta, X)|X} \right) \\
&= \frac{1}{n k_n} \sum_{i=1}^{n} \sum_{j \in N_{G(\mathscr{X}_n)}(i)} \mathsf{H} \left( \boldsymbol{W}_{i, \boldsymbol{g}}, \boldsymbol{W}_{j, \boldsymbol{g}} \right) \quad (3.1)
\end{aligned}
$$

where $\mathsf{H}$ is defined from (2.3) and $\boldsymbol{W}_{i, \boldsymbol{g}} := (Y_i, \boldsymbol{g}(\eta_i, X_i))$ for all $1 \le i \le n$. Our estimate of the conditional generator is then defined as

$$
\hat{\boldsymbol{g}} \in \arg\min_{\boldsymbol{g} \in \mathcal{G}} \hat{\mathcal{L}}(\boldsymbol{g}). \quad (3.2)
$$

With the framework now in place, we emphasize that CGMMD offers substantial flexibility to practitioners. In our experiments, we restrict $\mathcal{G}$ to deep neural networks, i.e., $\mathcal{G} = \left\{ g_\theta : \mathbb{R}^m \times \mathcal{X} \to \mathcal{Y} \mid \theta \in \mathbb{R}^{\mathcal{S}} \right\}$ where $\mathcal{S}$ is the total number of parameters of the neural network $g_\theta$. Here, (3.2) reduces to solving $\hat{\theta} \in \arg\min_{\theta \in \mathbb{R}^{\mathcal{S}}} \hat{\mathcal{L}}(g_\theta)$. A corresponding pseudo-code is provided in Algorithm 1.

In practice, the user may tailor the method by selecting the kernel $\mathsf{K}$, the function class $\mathcal{G}$, number of neighbors $k_n$, and the manner in which the auxiliary noise variable $\eta$ is incorporated into $g(\cdot, x)$. We discuss some of these potential choices as well as refinements to the CGMMD objective when $P_X$ has discrete support in Appendix D.

# 4. Analysis and Convergence Guarantees

In this section, we analyze the error of estimating the true conditional sampler $\bar{g}$ (see Lemma 2.2). This section is further divided into two parts. In Section 4.1 we begin by

---

**Algorithm 1** CGMMD Training

**Require:** Data $\{(Y_i, X_i)\}_{i=1}^n$, generator $g_\theta$, auxillary kernel function $\mathsf{H}$, noise $P_\eta$, stepsize $\alpha$, epochs $E$, batch size $B$, neighbors $k_B$.
1: Sample $\{\eta_i\}_{i=1}^n \sim P_\eta$
2: **for** $e = 1$ to $E$ **do**
3:     **for** mini-batch $I \subset [n]$, $|I| = B$ **do**
4:       $G \leftarrow k_B$-NN graph on $\{X_i\}_{i \in I}$
5:       $g_i \leftarrow g_\theta(\eta_i, X_i)$, $W_{i,g} \leftarrow (Y_i, g_i) \ \forall i \in I$
6:       $\hat{\mathcal{L}} \leftarrow \frac{1}{B k_B} \sum_{i \in I} \sum_{j \in N_G(i)} \mathsf{H}(W_{i,g}, W_{j,g})$
7:       $\theta \leftarrow \theta - \alpha \nabla_\theta \hat{\mathcal{L}}$
8:     **end for**
9: **end for**
10: **return** $\hat{\theta} \leftarrow \theta$

---

deriving a finite-sample bound on the error arising from replacing the true conditional sampler $\bar{g}$ with its empirical estimate $\hat{g}$. As a further contribution in Section 4.2, we establish the convergence of the conditional distribution induced by the empirical sampler to the true conditional distribution. For clarity and ease of exposition, we present simplified versions of the assumptions and main results here, while deferring the complete statements and proofs to Appendix E.

## 4.1. Non-Asymptotic Error Bounds

For the estimated empirical sampler $\hat{g}$ defined in (3.2) the estimation error can be defined as (recall Definition 1.2),

$$
\begin{aligned}
\mathcal{L}(\hat{\boldsymbol{g}}) &= \text{ECMMD}^2 \left[ \mathcal{F}, P_{\bar{g}(\eta, X)|X}, P_{\hat{g}(\eta, X)|X} \right] \\
&= \mathbb{E} \left[ \| \mu_{P_{\bar{g}(\eta, X)|X}} - \mu_{P_{\hat{g}(\eta, X)|X}} \|_{\mathcal{K}}^2 \mid \hat{\boldsymbol{g}} \right], \quad (4.1)
\end{aligned}
$$

where the expectations are taken over the randomness of $\eta$ and $X$ keeping the empirical sampler $\hat{g}$ fixed. In other words, the estimation error evaluates the squared ECMMD between the conditional distributions of $\bar{g}(\eta, X)$ and $\hat{g}(\eta, X)$ given $X$. In the following, we will provide non-asymptotic bounds on the estimation error $\mathcal{L}(\hat{g})$. To that end, for the rest of the article, we assume $\mathcal{Y} \subseteq \mathbb{R}^p$ for some $p \ge 1$ and we begin by rigorously defining the class of functions $\mathcal{G}$.

**Details of $\mathcal{G}$:** Let $\mathcal{G} = \mathcal{G}_{\mathcal{H}, \mathcal{W}, \mathcal{S}, \mathcal{B}}$ be the set of ReLU neural networks $g : \mathbb{R}^m \times \mathbb{R}^d \to \mathbb{R}^p$ with depth $\mathcal{H}$, width $\mathcal{W}$, size $\mathcal{S}$ and $\|g\|_\infty \le \mathcal{B}$. In particular, $\mathcal{H}$ denotes the number of hidden layers and $(w_0, w_1, \ldots, w_{\mathcal{H}})$ denotes the width of each layer, where $w_0 = d + m$ and $w_{\mathcal{H}} = p$ denotes the input and output dimension, respectively. We take $\mathcal{W} = \max\{w_0, w_1, \ldots, w_{\mathcal{H}}\}$. Finally, size $\mathcal{S} = \sum_{i=1}^{\mathcal{H}} w_i (w_{i-1} + 1)$ refers to the total number of parameters of the network. To establish the error bounds, we make the following assumption about the parameters of $\mathcal{G}$.

**Assumption 4.1.** The network parameters of $\mathcal{G}$ satisfies $\mathcal{B} \geq 1$ and $\mathcal{H}, \mathcal{W} \to \infty$ such that,

$$\frac{\mathcal{H}\mathcal{W}}{(\log n)^{\frac{d+m}{2}}} \xrightarrow{n \to \infty} \infty \quad \text{and} \quad \frac{\mathcal{B}^2 \mathcal{H} \mathcal{S} \log \mathcal{S} \log n}{n} \xrightarrow{n \to \infty} 0.$$

The imposed conditions require that the neural network's size grows with the sample size, specifically that the product of its depth and width increases with $n$. These assumptions are flexible enough to accommodate a wide range of architectures, but a key constraint is that the network size must remain smaller than the sample size. This arises from the use of empirical process theory (Van Der Vaart & Wellner, 1996; Bartlett et al., 2019) to control the stochastic error in the estimated generator. Similar conditions appear in recent work on conditional sampling (Zhou et al., 2023; Liu et al., 2021; Song et al., 2025) and in convergence analyses for deep nonparametric regression (Schmidt-Hieber, 2020; Kohler & Langer, 2021; Nakada & Imaizumi, 2020). We also make the following technical assumptions.

**Assumption 4.2.** The following conditions on $P_{YX}$, the kernel $\mathsf{K}$, the true conditional sampler $\bar{g}$ and the class $\mathcal{G}$ holds.

1. $P_X$ is supported on $\mathcal{X} \subseteq \mathbb{R}^d$ for some $d > 0$ and $\|X_1 - X_2\|_2$ has a continuous distribution for $X_1, X_2 \sim P_X$.

2. Moreover $X \sim P_X$ is sub-gaussian, that is [1], $\mathbb{P}(\|X\|_2 > t) \lesssim \exp(-t^2)$ for all $t > 0$.

3. The target conditional sampler $\bar{g}: \mathbb{R}^m \times \mathbb{R}^d \to \mathbb{R}^p$ is uniformly continuous with $\|\bar{g}\|_\infty \leq 1$.

4. For any $g \in \mathcal{G}$ consider

$$h_g(x) = \mathbb{E}\left[\mathsf{K}(Y, \cdot) - \mathsf{K}(g(\eta, X), \cdot) | X = x\right]$$

and assume that $|\langle h_g(x), h_g(x_1) - h_g(x_2)\rangle| \lesssim \|x_1 - x_2\|_2$, for all $x, x_1, x_2 \in \mathcal{X}$ where the constant is independent of $g$.

The first two assumptions are standard in the nearest neighbor literature and have been studied in the context of conditional independence testing using nearest neighbor-based methods (Huang et al., 2022a; Deb et al., 2020; Azadkia & Chatterjee, 2021; Borgonovo et al., 2025; Dasgupta & Kpotufe, 2014). The first, concerning uniqueness in nearest neighbor selection, can be relaxed via tie-breaking schemes (see Section 7.3 in Deb et al. (2020)), though we do not pursue this direction. The second, on the tail behavior of

---

[1] We use the notation $a \lesssim_\theta b$ to imply $a \leq C_\theta b$ for some constant $C_\theta > 0$ depending on the parameter $\theta$. In particular $a \lesssim b$ implies $a \leq Cb$ for some universal constant $C > 0$. Henceforth take $\theta = (d, m, p, \mathsf{K})$.

the predictor $X$, can be weakened to include heavier-tailed distributions, such as those satisfying sub-Weibull conditions (Vladimirova et al., 2020) (also see (E.1)). The third assumption is mainly for technical convenience; similar conditions appear in prior work on neural network-based conditional sampling (Zhou et al., 2023; Song et al., 2025; Liu et al., 2021). Its uniform continuity condition can also be relaxed to continuity (see Appendix E).

*Remark* 4.3. Assumption 4.2.4 is arguably the most critical in our analysis. It quantifies the sensitivity of the conditional mean embeddings to changes in the predictor $X$, and is essential for establishing concentration of the nearest-neighbor-based ECMMD estimator (see (2.3)) around its population counterpart. Similar assumptions have been used in prior work on nearest neighbor methods (Huang et al., 2022a; Deb et al., 2020; Azadkia & Chatterjee, 2021; Dasgupta & Kpotufe, 2014). As noted in Azadkia & Chatterjee (2021, Section 4), omitting such regularity conditions can lead to arbitrarily slow convergence rates. While the locally lipschitz-type condition can be relaxed, for example to Hölder continuity upto polynomial factors (see (E.2)) it remains a key assumption for our theoretical guarantees. We further elaborate on this assumption in Appendix F.

Under the above assumptions, we are now ready to present our main theorem on the error incurred by using the empirical sampler $\hat{g}$.

**Theorem 4.4** (Simpler version of Theorem E.4). *Adopt Assumption 2.1, Assumption 4.1 and Assumption 4.2. Moreover take $\omega_{\bar{g}}(r) := \sup\{\|\bar{g}(x) - \bar{g}(y)\|_2 : x, y \in \mathbb{R}^{d+m}, \|x - y\|_2 \leq r\}$ to be the optimal modulus of continuity of the true conditional sampler $\bar{g}$. Let $k_n = o(\sqrt{n})$, then for any $\delta \in (0, 1)$, with probability at least $1 - \delta$,*

$$\mathcal{L}(\hat{g}) \lesssim_\theta \frac{\text{poly} \log n}{n^{\frac{1}{2d}}} + \sqrt{\frac{\mathcal{B}^2 \mathcal{H} \mathcal{S} \log \mathcal{S} \log n}{n}}$$
$$+ \omega_{\bar{g}}\left(\frac{2\sqrt{\log n}}{(\mathcal{H}\mathcal{W})^{\frac{1}{d+m}}}\right) + \sqrt{\frac{\log(1/\delta)}{n}}.$$

The first two terms capture the stochastic error from the uniform concentration of the empirical loss around the population ECMMD objective. The third term reflects approximation error from estimating the true conditional sampler $\bar{g}$ using neural networks in $\mathcal{G}$. While we defer the proof of this result and its generalization to Appendix B.1 and Appendix E, respectively, we highlight the main novelty of our analysis here. Specifically, it integrates tools from recent advances in uniform concentration for non-linear functionals (Maurer & Pontil, 2019; Ni & Huo, 2024), nearest neighbor methods (Azadkia & Chatterjee, 2021; Deb et al., 2020), and generalization theory, including neural network approximation of smooth functions (Shen et al.,

2020; Zhang et al., 2022). To our knowledge, this is the first application of these techniques to conditional generative modeling with nonparametric nearest neighbor objectives. Additionally, we establish a uniform concentration result for a broad class of nearest-neighbor-based functionals (Appendix G), which may be of independent interest.

**Adaptation to Intrinsic Dimensionality.** The explicit dependence on feature dimension $d$ in Theorem 4.4 (see also Theorem E.4) likely reflects the intrinsic hardness of conditional sampling. Such curse-of-dimensionality phenomenon are widely known to be a fundamental hardness in related problems of density estimation, conditional density estimation, and conditional mean estimation (Tsybakov, 2009; Bilodeau et al., 2023; Li et al., 2022).

Importantly, CGMMD adapts to the intrinsic dimension of the conditioning variable. Specifically, we say that $\boldsymbol{X}$ has intrinsic dimension $\bar{d}$ if, for all $t > 0$ and $\varepsilon \in (0, t]$, the set $B(t) \cap \text{Support}(\boldsymbol{X})$ can be covered by at most $O((t/\varepsilon)^{\bar{d}})$ closed balls of radius $\varepsilon$. This notion, studied in prior works of Deb et al. (2020) and Huang et al. (2022a), is closely related to the *Assouad dimension* (Fraser, 2020) and includes settings where $\boldsymbol{X}$ lies on a lower-dimensional manifold (Huang et al., 2022a). Under this definition, the dimensional dependence in Theorem 4.4 (and similarly in Theorem E.4) can be improved by replacing $d$ with $\bar{d}$. In light of the manifold hypothesis (Whiteley et al., 2026), which posits that many high-dimensional datasets concentrate near low-dimensional manifolds, this adaptation is particularly relevant in practice. We validate this adaptation with a synthetic experiment in Appendix C.2.

### 4.2. Convergence of the Empirical Sampler

As outlined earlier, in this section, we leverage the bound established in Theorem 4.4 to demonstrate the convergence of the conditional distribution identified by the estimated sampler $\hat{\boldsymbol{g}}(\boldsymbol{\eta}, \boldsymbol{X})$ to the true conditional distribution.

While Theorem 4.4 provides a finite-sample quantitative guarantee on the loss incurred by using the estimated sampler in place of the true sampler $\boldsymbol{g}$, we now show that the conditional distribution induced by $\hat{\boldsymbol{g}}$ converges to the true conditional distribution. Furthermore, we strengthen this result by establishing convergence in terms of characteristic functions as well. By a classical result by Bochner (see Theorem H.2) every continuous positive definite function $\psi$ is associated with a finite non-negative Borel measure $\Lambda_\psi$. With this notation, we have the following convergence result with proof given in Appendix B.2.

**Corollary 4.5.** *Suppose the assumptions from Theorem 4.4 hold. Then,*

$$\mathbb{E}\left[\text{MMD}^2\left[\mathcal{F}, P_{\hat{\boldsymbol{g}}(\boldsymbol{\eta},\boldsymbol{X})|\boldsymbol{X}}, P_{\bar{\boldsymbol{g}}(\boldsymbol{\eta},\boldsymbol{X})|\boldsymbol{X}}\right]\right] \longrightarrow 0. \quad (4.2)$$

*Moreover, if the kernel* $\mathsf{K}(\boldsymbol{x}, \boldsymbol{y}) = \psi(\boldsymbol{x} - \boldsymbol{y})$ *for some*

*bounded, lipschitz continuous positive definite function* $\psi$. *Then,*

$$\int \left(\phi_{\hat{\boldsymbol{g}}(\boldsymbol{\eta},\boldsymbol{X})|\boldsymbol{X}}(\boldsymbol{t}) - \phi_{\bar{\boldsymbol{g}}(\boldsymbol{\eta},\boldsymbol{X})|\boldsymbol{X}}(\boldsymbol{t})\right)^2 \mathrm{d}\Lambda_\psi(\boldsymbol{t}) \xrightarrow{P} 0 \quad (4.3)$$

*where* $\phi_{\hat{\boldsymbol{g}}(\boldsymbol{\eta},\boldsymbol{X})|\boldsymbol{X}}$ *and* $\phi_{\bar{\boldsymbol{g}}(\boldsymbol{\eta},\boldsymbol{X})|\boldsymbol{X}}$ *are the characteristic functions of the conditional distributions* $P_{\hat{\boldsymbol{g}}(\boldsymbol{\eta},\boldsymbol{X})|\boldsymbol{X}}$ *and* $P_{\bar{\boldsymbol{g}}(\boldsymbol{\eta},\boldsymbol{X})|\boldsymbol{X}}$ *respectively.*

The above results demonstrate the efficacy of CGMMD. In particular, they show that the conditional distribution learned by the conditional sampler in CGMMD closely approximates the true conditional distribution.

## 5. Numerical Experiments

We begin our empirical study with toy examples of bivariate conditional sample generation, then move to practical applications such as image denoising and super-resolution on MNIST (Yann, 2010), denoising on CelebHQ (Karras et al., 2018), super-resolution on STL10 (Coates et al., 2011) and inpainting on FashionMNIST (Xiao et al., 2017). We compare CGMMD with the methods in Zhou et al. (2023) and Song et al. (2025) on synthetic data and also add comparisons with conditional normalizing flows in synthetic benchmarks. Moreover, to assess test-time complexity, we compare CGMMD with a diffusion model using classifier-free guidance (Ho & Salimans, 2021). Due to space constraints, only selected results are shown here; full details appear in Appendix C. In all the experiments presented here, we have used the Gaussian kernel and batch-size 200. Codes are available at https://github.com/anirbanc96/cgmmd.

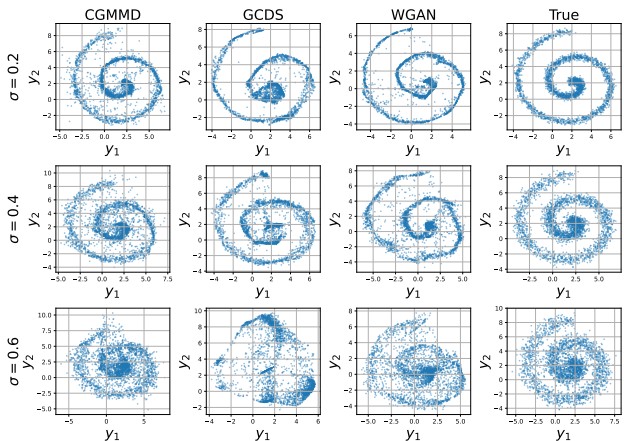

*Figure 2.* Comparison of conditional generators on the Helix benchmark at $\boldsymbol{X} = 1$.

### 5.1. Conditional Bivariate Sampling

In this section, we compare our proposed CGMMD with two baseline approaches: the GCDS (Zhou et al., 2023), a

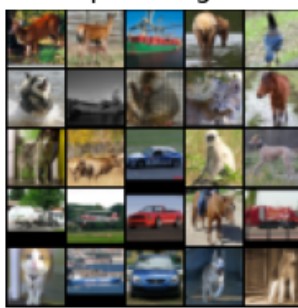 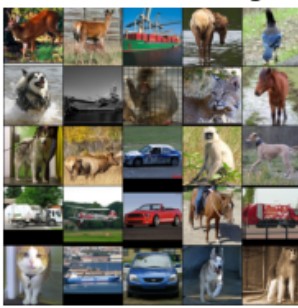 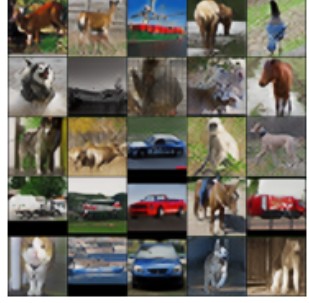 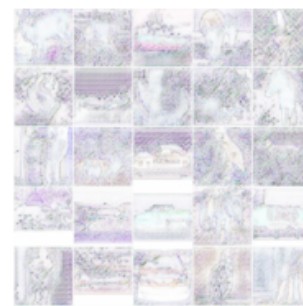

*Figure 3.* High resolution reconstructions of STL10 images from low resolution inputs. From left to right: The low resolution input images, the true high resolution images, mean of reconstructed images from CGMMD, pixel-wise standard deviation of the reconstructed images.

vanilla GAN framework, and a Wasserstein-based modification, WGAN (trained with pure Wasserstein loss) (Song et al., 2025). We consider a synthetic setup with $X \sim$ $N(0,1)$, $U \sim \text{Unif}[0, 2\pi]$, and $\varepsilon_1, \varepsilon_2 \overset{iid}{\sim} N(0, \sigma^2)$. The response variables are $Y_1 = 2X + U\sin(2U) + \varepsilon_1$, $Y_2 = 2X + U\cos(2U) + \varepsilon_2$, and our goal is to generate conditional samples from $(Y_1, Y_2) \mid X$ at varying noise levels ($\sigma$). All three methods use the same two-hidden-layer feed-forward ReLU generator with noise $\eta$ concatenated to the generator input, and are evaluated at noise levels $\sigma \in \{0.2, 0.4, 0.6\}$. At low noise ($\sigma = 0.2$), all three methods recover the helix structure well. As the noise level rises, however, CGMMD maintains the overall curvature, in particular at the 'eye' (the center of the helix), while the reconstructions from GCDS and WGAN degrade noticeably (See Figure 2). In this regard we have noticed that without $\ell_1$ regularisation WGAN training is often unstable. We also explore an additional conditional bivariate setting (which imitates circular structure), with qualitatively similar results deferred to Appendix C.1 and Appendix C.3.

### 5.2. Image Super-Resolution and Denoising

In this section, we evaluate the performance of CGMMD across two tasks: image super-resolution and image denoising. For this, we use the MNIST, STL-10 and CelebHQ datasets.

**Super-Resolution.** We implement CGMMD for the $4\times$ image super-resolution task on MNIST ($7 \times 7$ to $28 \times 28$) and STL-10 (Coates et al., 2011) ($3 \times 24 \times 24$ to $3 \times 96 \times 96$). This task can be naturally formulated as a conditional generation problem, where the goal is to produce a high-resolution image given a low-resolution input. Figure 4 and Figure 3 show that CGMMD accurately reconstructs high-resolution images from low-resolution inputs for both MNIST and STL-10. Moreover, the pixel-wise standard deviation image in Figure 3 indicates that our method yields

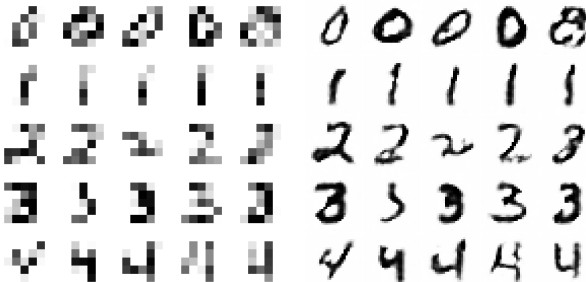

*Figure 4.* MNIST Super-Resolution with digits $\{0, 1, 2, 3, 4\}$.

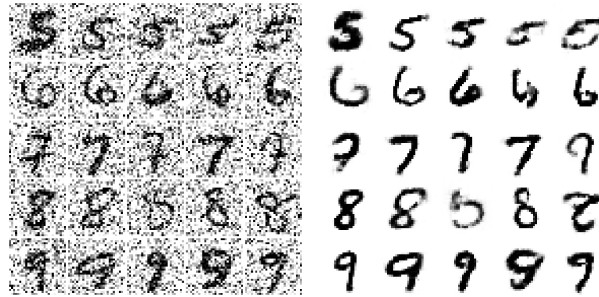

*Figure 5.* MNIST Denoising with digits $\{5, 6, 7, 8, 9\}$ at $\sigma = 0.5$.

substantial diversity in the generated outputs. We emphasize that our goal is not to outperform state-of-the-art super-resolution methods (Kim et al., 2016; Zhang et al., 2018), but rather to demonstrate the flexibility of our approach. Additional results and experimental details are provided in Appendix C.4 and Appendix C.6.

**Image Denoising.** We evaluate CGMMD on the image denoising task using the MNIST ($28 \times 28$ iamges) and CelebHQ ($3 \times 64 \times 64$ images) datasets. In this task, the inputs are images (digits for MNIST and facial images for CelebHQ) corrupted with additive Gaussian noise ($\sigma = 0.5, 0.25$ for MNIST and CelebHQ respectively). We

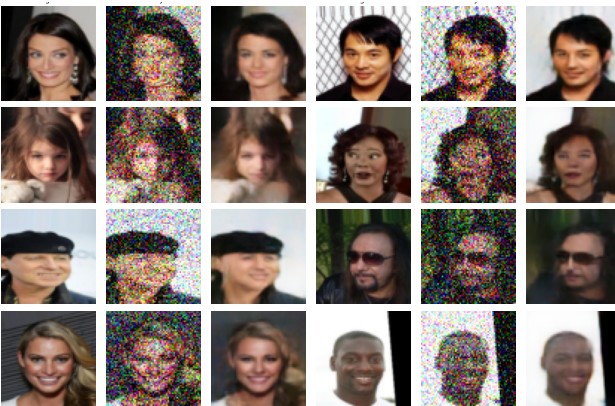

*Figure 6.* CelebHQ denoising using CGMMD at $\sigma = 0.25$. From left to right: Original, Noisy and Denoised.

can indeed formulate this as a conditional generation problem. In Figure 5, the left 5 columns represent the noisy digit images while the right 5 columns are the clean images reconstructed using CGMMD. Additional experiments and details are given in Appendix C.4. For the CelebHQ experiment, Figure 6 shows original images (left), noisy inputs (middle), and denoised outputs produced by CGMMD (right). The results demonstrate that our model effectively reconstructs clean facial images from noisy inputs and preserves quality even under high noise levels. Additional details are given in Appendix C.5.

*Table 1.* Comparison of CGMMD with Diffusion Model (DM) and Distilled Diffusion (DD).

| Model | PSNR | SSIM | Time/batch (s) | Time/img (s) |
|---|---|---|---|---|
| DM | 13.326 | 0.861 | 6.94 | $5.42 \times 10^{-2}$ |
| DD | 10.658 | 0.508 | $1.18 \times 10^{-1}$ | $9.2 \times 10^{-4}$ |
| CGMMD | 8.922 | 0.718 | $7.21 \times 10^{-2}$ | $5.6 \times 10^{-4}$ |

**Comparison with Conditional Diffusion Models.** In Table 1, we compare CGMMD with a diffusion model using classifier-free guidance (Ho & Salimans, 2021) and progressive distilled diffusion (Meng et al., 2023; Salimans & Ho, 2022) on the MNIST image denoising task ($\sigma = 0.9$). The results in Table 1 indicate that the diffusion model achieves higher-quality reconstructions but at a substantially higher computational cost compared to CGMMD. Distilled diffusion offers comparable performance to CGMMD, achieving better PSNR but lower SSIM, while incurring a moderate increase in computation. Overall, CGMMD provides a favorable trade-off, generating images of reasonable quality much faster, making it particularly well-suited for applications where rapid conditional sampling is essential.

# 6. Conclusion

We introduced CGMMD, a conditional generative framework that learns the full conditional law of $Y \mid X$ by directly minimizing an empirical ECMMD objective. Unlike adversarial approaches, the proposed method avoids min–max optimization while retaining the flexibility of neural conditional generators and enabling efficient one-shot sampling. Our nearest-neighbor-based construction provides a practical estimator of the conditional MMD loss and leads to a simple and scalable training algorithm.

On the theoretical side, we established non-asymptotic finite-sample error bounds for the learned sampler and proved convergence of the induced conditional distribution to the true conditional law. These guarantees connect conditional generative modeling with tools from kernel embeddings, nearest-neighbor estimation, uniform concentration, and neural network approximation theory.

Empirically, CGMMD demonstrates consistent performance across synthetic conditional sampling tasks as well as image denoising, super-resolution, and inpainting benchmarks, achieving a favorable trade-off between accuracy and computational efficiency. We emphasize, however, that the primary goal of this work was not to surpass state-of-the-art methods on these benchmarks, but rather to illustrate the flexibility of the proposed framework and provide empirical support for the underlying theoretical developments. An important direction for future work is the further refinement and optimization of CGMMD to enhance its empirical performance.

While our assumptions on the network architecture require the network size to grow with the sample size to guarantee accurate approximation, we believe the analysis can be extended to fixed-architecture networks achieving arbitrarily small approximation error. Another promising direction is the study of flow-based conditional sampling methods (Hagemann et al., 2024; Hertrich et al., 2024). We believe that the proof techniques developed in this work could also yield theoretical guarantees for flow-based generative sampling approaches trained with MMD objectives; we leave these directions for future work.

# Acknowledgments

We thank Bhaswar Bhattacharya for helpful discussions.

# Impact Statement

This paper presents work whose goal is to advance the field of Machine Learning. There are many potential societal consequences of our work, none which we feel must be specifically highlighted here.

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

# Supplementary Materials

## Contents

# A. Selected Background and Influences

Here we provide a concise overview of the most directly relevant lines of work that align with our approach to conditional generative modeling. We concentrate on selected contributions that either motivate or underpin our methodology, rather than attempting a full survey of the field.

**Statistical foundations of conditional density estimation**    A rich line of work in statistics addresses conditional density estimation through nonparametric methods. Classical approaches include kernel and local-polynomial smoothing (Rosenblatt, 1969; Hyndman et al., 1996; Chen et al., 2000; Hall & Yao, 2005) and regression-style formulations for conditional densities (Fan et al., 1996; Fan & Yim, 2004). Alternative strategies exploit nearest-neighbor ideas (Lincheng & Zhijun, 1985) or expansions in suitable basis functions (Izbicki & Lee, 2016; Sugiyama et al., 2010). More recent frameworks, such as distributional regression (Hothorn et al., 2014; Rigby & Stasinopoulos, 2005; Kock & Klein, 2025), model the entire conditional distribution directly rather than focusing on low-order summaries. Together, these approaches form the statistical foundation for modern methods of conditional density estimation.

**Conditional GAN and MMD Gradient Flows.**    Alongside classical approaches, Conditional Generative Adversarial Networks (cGANs) extend the original GAN framework (Goodfellow et al., 2014) by conditioning both the generator and discriminator on side information such as labels or auxiliary features (Zhou et al., 2023; Mirza & Osindero, 2014; Baptista et al., 2024; Odena et al., 2017). Variants employ projection-based discriminators for improved stability (Miyato & Koyama, 2018) or architectures tailored to structured outputs such as image-to-image translation (Isola et al., 2017; Denton et al., 2015; Reed et al., 2016). Despite strong empirical results, cGANs often inherit the instability and mode-collapse issues of adversarial training, motivating alternative losses based on integral probability metrics such as MMD or Wasserstein distances (Ren et al., 2016; Liu et al., 2021; Huang et al., 2022b; Song et al., 2025), which in turn inspire our ECMMD-based conditional generator. Among the most closely related works are Ren et al. (2016) and Huang et al. (2022b). Ren et al. (2016) introduce an RKHS-to-RKHS operator-based embedding to measure pointwise differences between conditional distributions. However, their formulation relies on strong assumptions that may not hold in continuous domains (Song et al., 2009), and the estimator incurs a high computational cost, up to $O(n^3)$ or $O(B^3)$, where $B$ is the batch size. In a related direction, Huang et al. (2022b) propose a measure equivalent to ECMMD for aleatoric uncertainty quantification and conditional sample generation. While their approach demonstrates strong empirical performance, it requires Monte Carlo sampling and potentially repeated sampling from both the generative model and the true conditional distribution, making it computationally intensive (up to $O(B^2)$). Furthermore, it remains unclear whether the learned generator consistently approximates the true conditional distribution. Another closely related work is Bouchacourt et al. (2016), where the authors introduce a similar loss function using Rao's dissimilarity coefficients (Rao, 1982) for posterior sampling.

Recently, another line of work has focused on (un)conditional sampling using Maximum Mean Discrepancy (MMD) gradient flows. In particular, (Arbel et al., 2019; Hagemann et al., 2024; Hertrich et al., 2024; Galashov et al., 2025) have proposed constructing Wasserstein gradient flows of the MMD and leveraging them for both conditional and unconditional sample generation. Notably, the recent work of (Hagemann et al., 2024) considers the same conditional sampling problem studied in this paper and proposes a flow-based model based on the energy distance (equivalently, a negative distance kernel). However, the key distinction between their work and ours lies in our MMD-GAN–based formulation, flexibility in the choice of kernels, as well as the rigorous theoretical analysis we provide, including finite-sample guarantees and comprehensive convergence results.

**Simulation-based inference.**    A parallel line of work on conditional sample generation appears in the simulation-based inference literature. One of the earliest and most popular approaches is Approximate Bayesian Computation (ABC) (see (Martin et al., 2024) and references within), which aims to draw approximate samples from the posterior distribution. Recent advances leverage modern machine learning to improve this process, typically by learning surrogate posteriors from simulations using neural networks (see (Cranmer et al., 2020) for a survey). For example, Ramesh et al. (2022) propose a GAN-based approach, while others employ normalizing flows as a powerful alternative (Rezende & Mohamed, 2015; Papamakarios et al., 2021; Linhart et al., 2022). We refer readers to Zammit-Mangion et al. (2024) for a comprehensive review of recent developments.

# B. Proofs of Theorem 4.4 and Corollary 4.5

## B.1. Proof of Theorem 4.4

Under Assumption 2.1, Assumption 4.2 and Assumption 4.1 Theorem 4.4 follows as a special case of Theorem E.4. To that end, from Theorem E.4 note that for any $\delta > 0$ with probability atleast $1 - \delta$, there exists an universal constant $C > 0$ such that,

$$\mathcal{L}(\hat{\boldsymbol{g}}) \lesssim_{\boldsymbol{\theta}} \sqrt{\frac{\mathcal{B}^2 \mathcal{H}\mathcal{S} \log \mathcal{S} \log n}{n} + \frac{\text{poly}\log(n)}{n^{\frac{1}{2d}}}} \tag{B.1}$$

$$+ \underbrace{1 - \Phi\left(R\right)^m \left(1 - C\exp\left(-R^2\right)\right)}_{L_1} + \underbrace{\sqrt{d+m}\,\omega_{\bar{\boldsymbol{g}}}^E \left(2R\left(\mathcal{H}\mathcal{W}\right)^{-\frac{1}{d+m}}\right)}_{L_2} + \sqrt{\frac{\log\left(1/\delta\right)}{n}}$$

for any $R > 0$ with $E = [-R, R]^d$ and,

$$\omega_{\bar{\boldsymbol{g}}}^E(r) = \sup\left\{\|\bar{\boldsymbol{g}}(\boldsymbol{x}) - \bar{\boldsymbol{g}}(\boldsymbol{y})\|_2 : \|\boldsymbol{x} - \boldsymbol{y}\|_2 \leq r, \boldsymbol{x}, \boldsymbol{y} \in E\right\}.$$

Note that from Assumption 4.2 we know $\bar{\boldsymbol{g}}$ is uniformly continuous, hence,

$$\omega_{\bar{\boldsymbol{g}}}^E(r) \leq \omega_{\bar{\boldsymbol{g}}}(r) \text{ for all } r > 0. \tag{B.2}$$

Moreover, take $R = R_n = \sqrt{(\log n)}$ then we can simplify the terms $L_1$ and $L_2$ as follows. To that end recall the expression $L_1$ and note that $\Phi$ is the CDF of standard Gaussian distribution. Then as $n \to \infty$ we have the lower bound

$$\Phi(R_n) \geq 1 - \frac{\exp(-R_n^2/2)}{\sqrt{2\pi}R_n},$$

and hence by Taylor series expansion,

$$\Phi(R_n)^m \geq 1 - \frac{m\exp\left(-R_n^2/2\right)}{\sqrt{2\pi}R_n} + O\left(\frac{\exp\left(-R_n^2\right)}{R_n^2}\right).$$

Then as $n \to \infty$ and recalling $R_n = \sqrt{\log n}$,

$$L_1 = 1 - \Phi(R_n)^m \left(1 - Ce^{-R_n^2}\right) \lesssim \frac{m\exp\left(-R_n^2/2\right)}{\sqrt{2\pi}R_n} + e^{-R_n^2} \lesssim \frac{1}{\sqrt{n}}. \tag{B.3}$$

With this choice of $R = R_n$ and recalling (B.2) we can simplify $L_2$ as,

$$L_2 \lesssim \omega_{\bar{\boldsymbol{g}}}\left(\frac{2\sqrt{\log n}}{(\mathcal{H}\mathcal{W})^{\frac{1}{d+m}}}\right). \tag{B.4}$$

The proof is now completed by combining the bounds from (B.1), (B.3) and (B.4).

## B.2. Proof of Corollary 4.5

The proof of the first convergence follows directly by observing that $\omega_{\bar{\boldsymbol{g}}}(r) \to 0$ as $r \to 0$ by definition, and applying Theorem 4.4, the expression for $\mathcal{L}(\hat{\boldsymbol{g}})$ in (4.1), and the Dominated Convergence Theorem (DCT).

The proof for the second convergence is an immediate consequence of the first convergence and Sriperumbudur et al. (2010, Corollary 4).

## C. Additional Experiments

In this section, we present full details about the experiments from Section 5 and additional experiments to depict usefulness of our approach CGMMD across varied tasks. In all of the experiments, unless otherwise stated we take K to be the Gaussian kernel, and use the AdamW optimizer with default parameters.

### C.1. Synthetic setup: Circle Generation

Much like the helix-generation experiment in Section 5.1, we now consider a synthetic sampling setup where the task remains to generate conditional samples from a bivariate distribution, but here the conditional distribution follows a circular rather than a spiral structure.

Specifically, let $X \sim \mathrm{N}(0, 1)$, $U \sim \mathrm{Unif}[0, 2\pi]$, and $\varepsilon_1, \varepsilon_2 \overset{\mathrm{iid}}{\sim} \mathrm{N}(0, \sigma^2)$. Define the response variables as

$$Y_1 = X + 3\sin(U) + \varepsilon_1, \quad Y_2 = X + 3\cos(U) + \varepsilon_2. \tag{C.1}$$

In this experiment we compare our proposed CGMMD with the GCDS method of Zhou et al. (2023). As before, both methods employ the same two-hidden-layer feed-forward ReLU generator with noise $\eta$ concatenated to the input, and we evaluate performance at noise levels $\sigma \in \{0.2, 0.4, 0.6\}$.

At low level noises both methods perform similarly. However, at higher noise levels, CGMMD preserves the circular shape of the conditional distribution (Figure 7), whereas GCDS tends to produce elliptical distortions.

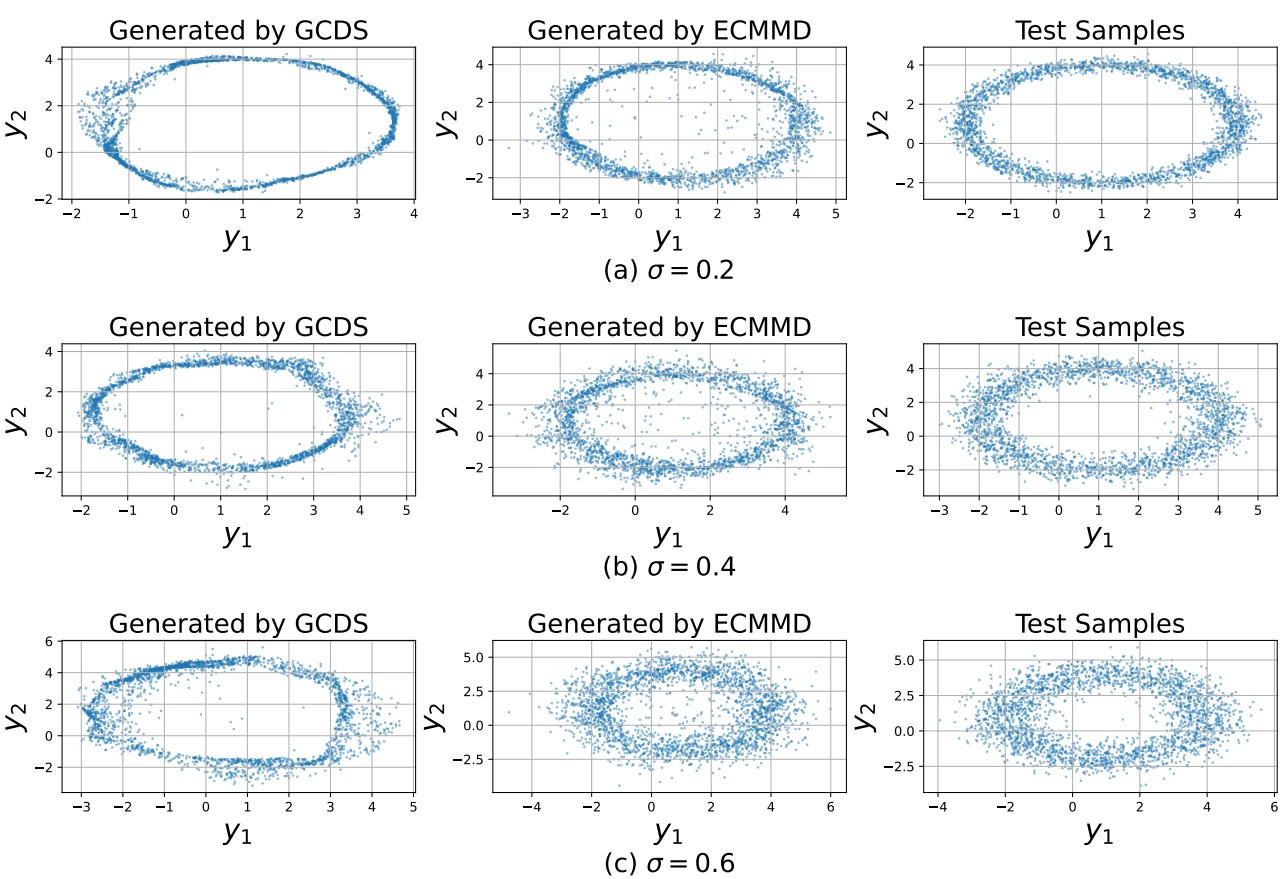

*Figure 7.* Comparison of conditional generators on the Circle benchmark

In Figure 8, we also demonstrate how quickly our approach CGMMD picks up the circular structure for the setting laid out in Section 5.1 at no more than 100 epochs even with a small two-hidden-layer feed-forward ReLU generator network.

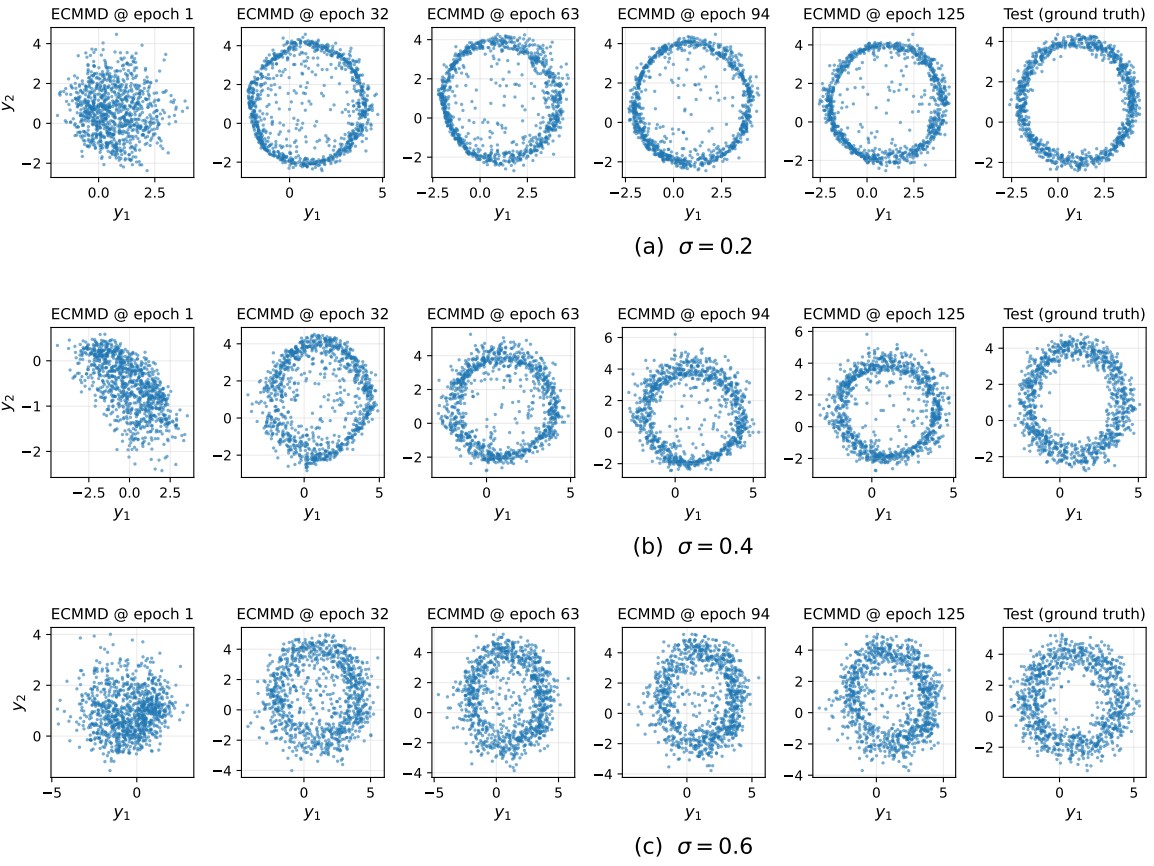

*Figure 8.* Conditional samples of $(\boldsymbol{Y}_1, \boldsymbol{Y}_2) \mid \boldsymbol{X} = 1$ for circle experiment, generated by CGMMD while training.

## C.2. Adaptation to Intrinsic Dimension

In this section, we conduct a synthetic experiment to validate our claim from Section 4 that CGMMD adapts to intrinsic dimensionality. Specifically, we take $\boldsymbol{Z} \sim \mathrm{N}_5(\boldsymbol{0}, \boldsymbol{I}_5)$ and define $\boldsymbol{X} = \boldsymbol{AZ}$, where $\boldsymbol{A} \in \mathbb{R}^{d \times 5}$ has orthonormal columns. We then generate the response variable as $\boldsymbol{Y} = \boldsymbol{X}^\top \boldsymbol{1}/\sqrt{d} + \varepsilon$, where $\varepsilon \sim \mathrm{N}(0, 1)$. In this experiment, we use CGMMD to train a conditional generator for the distribution of $\boldsymbol{Y} \mid \boldsymbol{X}$. We evaluate the quality of generated samples by computing the MSE between the conditional mean and standard deviation estimated from the learned distribution and the corresponding quantities under the true conditional distribution at fixed conditioning values corresponding to $\boldsymbol{Z} = 0.5\boldsymbol{1}_5$ and $\boldsymbol{Z} = -0.5\boldsymbol{1}_5$. We report the variation in MSE values across ambient dimensions $d \in \{25, 50, 100, 200, 300, 400, 500\}$ in Figure 9 with

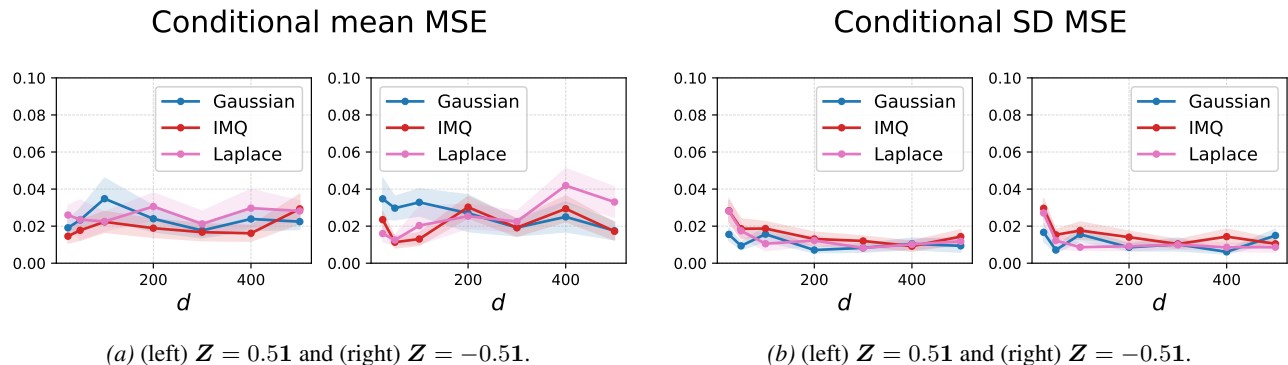

*(a)* (left) $\boldsymbol{Z} = 0.5\boldsymbol{1}$ and (right) $\boldsymbol{Z} = -0.5\boldsymbol{1}$.        *(b)* (left) $\boldsymbol{Z} = 0.5\boldsymbol{1}$ and (right) $\boldsymbol{Z} = -0.5\boldsymbol{1}$.

*Figure 9.* MSE of (a) Conditional Mean and (b) Conditional SD with increasing ambient dimension $d$.

Gaussian, Laplace and IMQ kernels. From Figures 9a and 9b, we observe that the MSE values remain largely unchanged as the ambient dimension increases. This supports the claim that the error behavior of CGMMD depends primarily on the intrinsic dimensionality rather than the ambient dimensionality.

### C.3. Comparisons with Normalizing Flows

In this section we compare the CGMMD with conditional normalizing flows in two settings. For the first experiment we consider the setting from Section C.1 and for the second setting we consider the two-moons benchmarking example from simulation based inference (Lueckmann et al., 2021; Ramesh et al., 2022).

#### C.3.1. CIRCLE GENERATION

Recall the conditional distribution $(Y_1, Y_2) \mid X$ from (C.1). In this experiment, we compare our proposed CGMMD with a conditional normalizing flow (cNF) following the general framework of (Winkler et al., 2019). Unlike their coupling-layer-based architecture, our flow uses 2–3 Masked Autoregressive Transform (MAF) layers (Papamakarios et al., 2017), interleaved with permutation layers, as the core building blocks. For CGMMD as before we employ two-hidden-layer feed-forward ReLU generator with noise $\eta$ concatenated to the input, and we evaluate performance at noise levels $\sigma \in \{0.4, 0.8\}$.

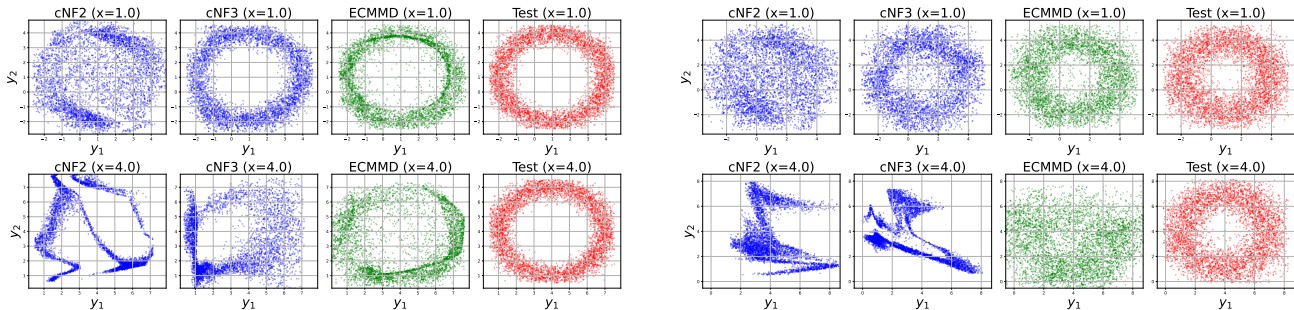

*Figure 10.* Conditional samples of $(Y_1, Y_2) \mid X$ from the circle experiment generated by CGMMD and cNF. The left panel corresponds to $\sigma = 0.4$ and the right panel to $\sigma = 0.8$. The top row shows samples conditional on $X = 1$, and the bottom row shows samples conditional on $X = 4$.

In Figure 10, we plot the conditional samples generated by CGMMD and cNF for $X = 1$ and $X = 4$ at noise levels $\sigma = 0.4$ and $0.8$. We observe that when $X$ belongs to a high-probability region ($X = 1$), both CGMMD and cNF produce accurate conditional samples. However, when $X$ belongs to a low-probability region ($X = 4$), CGMMD is able to retain the semblance of the circular structure, whereas cNF fails to capture the underlying circular conditional distribution.

#### C.3.2. TWO MOONS

In this section, we consider sampling from the unknown posterior distribution in the two-moons benchmarking task from simulation-based inference (Lueckmann et al., 2021; Ramesh et al., 2022). The true posterior exhibits global bimodality and a locally crescent-shaped structure, making it a challenging benchmarking problem.

Here the data generating process has the following structure. Generate $Y = (Y_1, Y_2)$ from the uniform distribution on the unit square $[-1, 1]^2$ and then given $Y$ generate $X$ as follows:

$$X \mid Y = (r \cos(\alpha) + 0.25, r \sin(\alpha)) + \left( -\frac{|Y_1 + Y_2|}{\sqrt{2}}, \frac{Y_2 - Y_1}{\sqrt{2}} \right)$$

where $\alpha \sim \text{Unif}(-\pi/2, \pi/2)$ and $r \sim \text{N}\left(0.1, 0.01^2\right)$. Given paired samples from the above data generating procedure, the objective is to learn the posterior distribution of $Y \mid X$. To that end we implement the CGMMD and flow-based neural posterior estimation (SNPE) using MAF from the sbi (Tejero-Cantero et al., 2020) package. For CGMMD we implement a ResNet-style generator using LayerNorm residual blocks (MLP) and also a MDN-based generator with LayerNorm residual blocks producing full-covariance Gaussian mixtures.

In Figure 11, we show conditional samples generated by CGMMD and SNPE for $X = (-0.64, 0.162)$ and $(-0.25, 0.633)$. These $X$ values are chosen from the sbibm package (Lueckmann et al., 2021), which provides reference posterior samples

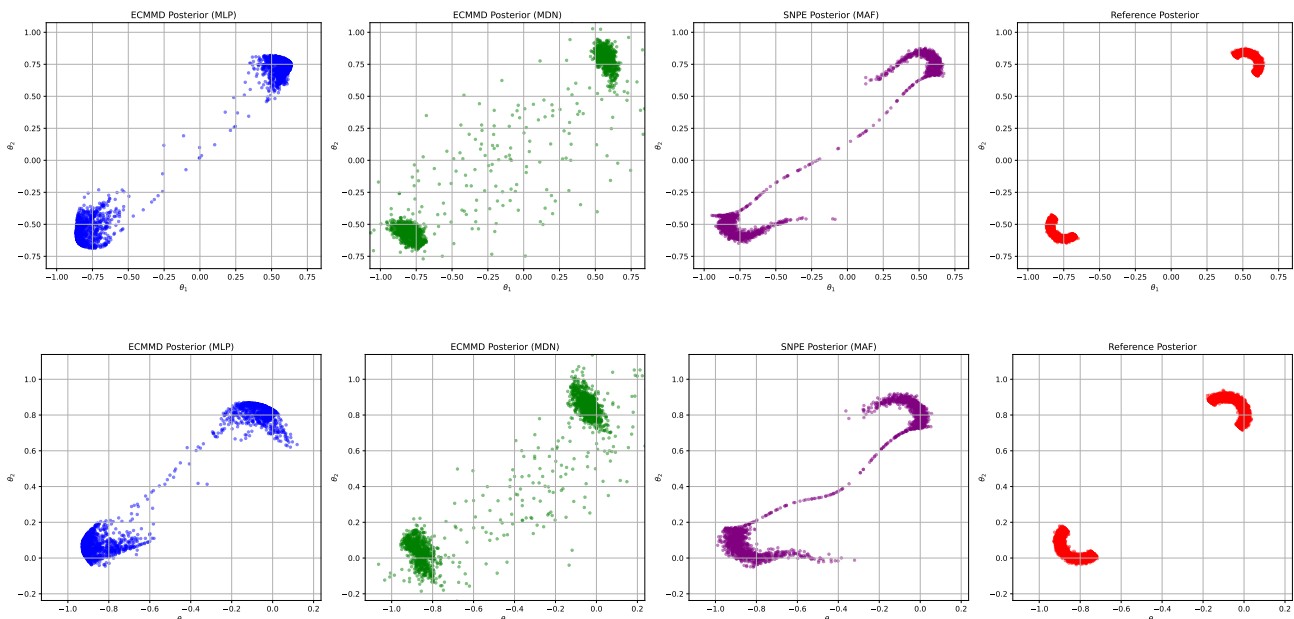

*Figure 11.* Conditional samples of $Y \mid X$ for the two-moons experiment, generated by CGMMD and SNPE from `sbi` (Tejero-Cantero et al., 2020). The top row shows samples conditional on $X = (-0.64, 0.162)$, while the bottom row corresponds to $X = (-0.25, 0.633)$. Reference posterior samples are taken from the `sbibm` package (Lueckmann et al., 2021).

for comparison. In both cases, SNPE captures the bimodality and the local crescent-shaped structure, whereas CGMMD preserves the bimodality but does not fully capture the local crescent shape. The MLP model, however, captures the presence of local curvature. This aligns with observations in (Ramesh et al., 2022), where GAN-based models were noted to struggle in capturing the local crescent structure.

### C.4. Additional results on MNIST super-resolution and denoising

Here, we present the complete results (performance for all digits in $\{0, 1, \ldots, 9\}$) for the image denoising and image super resolution task laid out in Section 5.2. For both denoising ( see Figure 12 and Figure 13) and $4X$ super-resolution task (see Figure 14 and Figure 15), we present the average reconstructed images generated by CGMMD along with the corresponding standard-deviation images for all the digits. We conclude that on average our method can reconstruct the original images with good precision. Moreover, the non-trivial pixel-wise standard deviation indicates substantial diversity in the generated images, supporting the effectiveness of the conditional sampling objective of CGMMD.

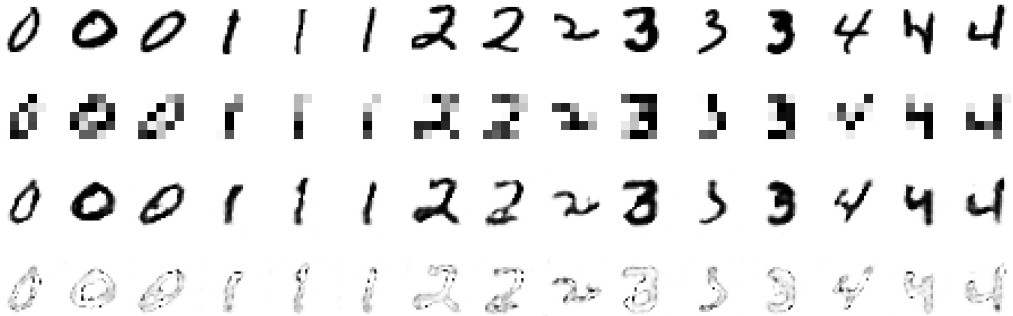

*Figure 12.* Additional MNIST super-resolution results for digits $\{0, 1, 2, 3, 4\}$. Rows show (top to bottom): ground-truth images, corresponding low-resolution inputs, high-resolution mean reconstructions, and pixel-wise standard deviations.

For the $4X$ super-resolution task on MNIST we use the following architechture: The model begins with two convolutional layers, interspersed with Batch Normalization and ReLU activations. The resulting feature maps are then concatenated with

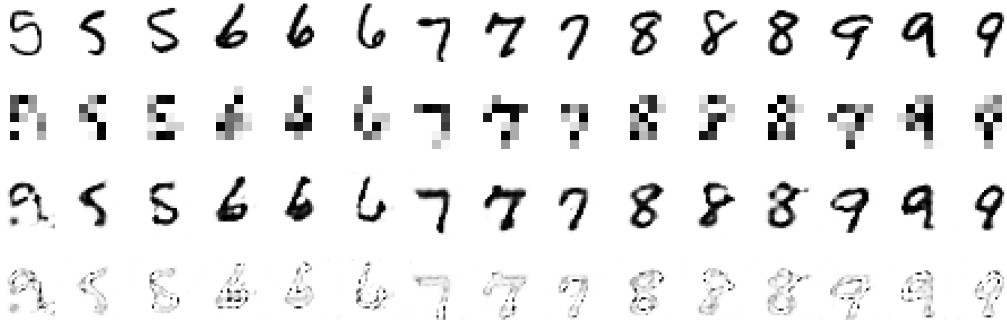

*Figure 13.* Additional MNIST super-resolution results for digits $\{0, 1, 2, 3, 4\}$. Rows show (top to bottom): ground-truth images, corresponding low-resolution inputs, high-resolution mean reconstructions, and pixel-wise standard deviations.

the auxiliary noise input and passed through two transposed convolutional layers for upsampling, each again interspersed with Batch Normalization and ReLU. A final convolutional layer with a sigmoid activation generates the high-resolution output.

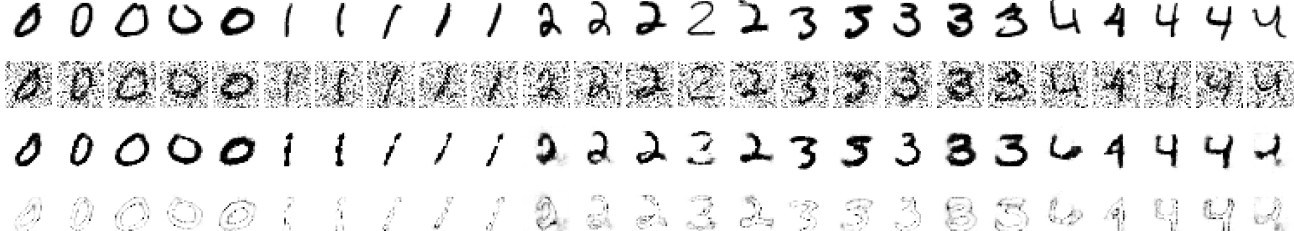

*Figure 14.* Additional MNIST denoising results for digits $\{0, 1, 2, 3, 4\}$. Rows show (top to bottom): ground-truth images, corresponding noisy inputs, denoised mean images, and pixel-wise standard deviations.

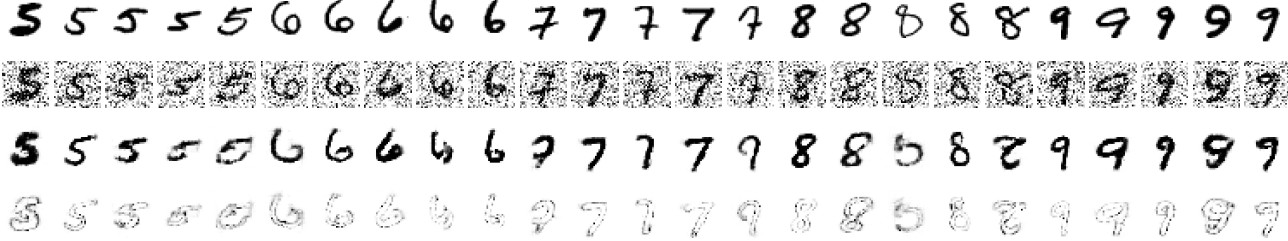

*Figure 15.* Additional MNIST denoising results for digits $\{5, 6, 7, 8, 9\}$. Rows show (top to bottom): ground-truth images, corresponding noisy inputs, denoised mean images, and pixel-wise standard deviations.

For the denoising task on MNIST, we use a CNN-based autoencoder architecture. The model begins with an encoder composed of two convolutional layers interspersed with ReLU activations and max-pooling operations. The encoded features are flattened and passed through two fully connected layers with ReLU activations. After feature extraction, the auxiliary noise is concatenated with the feature representation, and the combined vector is processed by another set of fully connected layers with ReLU activations. The resulting tensor is reshaped and passed through a decoder consisting of two transposed convolutional layers, the first followed by a ReLU activation and the second by a sigmoid activation, producing the denoised output.

### C.5. Additional results on image denoising with CelebHQ dataset

Here we present additional examples of the image denoising task on the CelebA-HQ dataset (Karras et al., 2018) from Section 5. The dataset consists of 30,000 high-quality images of celebrity faces. For our experiments, we downsampled the images to $64 \times 64$ resolution and added Gaussian noise with standard deviation $\sigma = 0.25$. To generate Figure 16, we selected images at random and applied $\ell_1$ regularization to enhance sharpness.

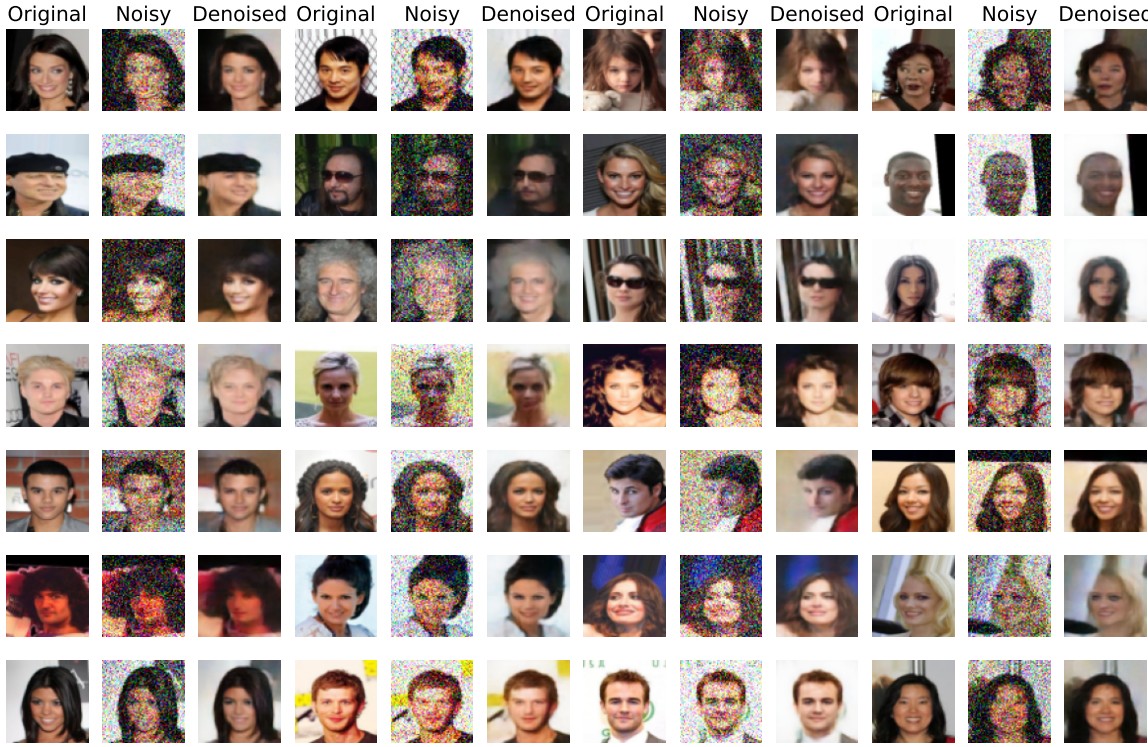

Original  Noisy  Denoised  Original  Noisy  Denoised  Original  Noisy  Denoised  Original  Noisy  Denoised

*Figure 16.* Performance of CGMMD on image denoising task. For each image, we plot the original clean image, the noisy image and the denoised image generated by CGMMD.

## C.6. Super-resolution with STL10 dataset

In this section, we add details to the super-resolution experiment from Section 5.2. Since nearest-neighbor methods scale poorly in high dimensions, we embed images in a lower dimensional space via a ResNet-18 encoder followed by PCA and perform neighborhood computations in this space. Real-world data are usually high-dimensional, but almost always reside on low-dimensional manifolds; leveraging such embeddings improves reconstruction quality, as also noted by prior work (Li et al., 2015; Ren et al., 2016; Huang et al., 2022b). We additionally apply $\ell_1$ regularization to obtain sharper reconstructions. To reiterate, as shown in Figure 3, similar to the MNIST experiments, our method is able to generate high-resolution images that closely resemble the ground truth.

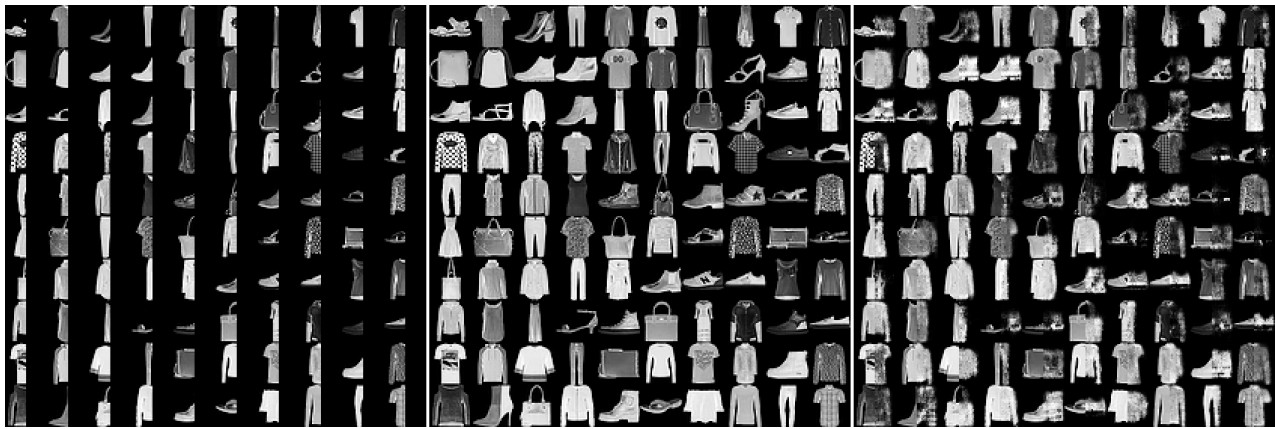

*Figure 17.* Inpainted reconstructions of FashionMNIST (Xiao et al., 2017) images. From left to right: the left-half input, the original full image, and the inpainted output produced by CGMMD, respectively.

## C.7. Image inpainting with FashionMNIST

In this section, we address the task of image inpainting on the FashionMNIST dataset (Xiao et al., 2017), where the goal is to reconstruct the right half of each fashion product image from its left half. In our setup, the model receives the left $28 \times 14$ portion of the image as input and produces a full $28 \times 28$ image, with the generated $28 \times 14$ right half augmented with the original left half.

In Figure 17, we present the performance of CGMMD in reconstructing full images for each FashionMNIST product category. For most examples, the reconstructions resemble the true items, and the results further demonstrate that CGMMD effectively captures the diversity across categories.

## D. Design Choices and Practical Considerations

**Choice of** $\mathsf{K}$ **and** $k_n$. While various kernels $\mathsf{K}$ can be used, standard choices like Gaussian $\mathsf{K}_\sigma^{\mathrm{gauss}}(x,y) = \exp\left(-\frac{\|x-y\|_2^2}{2\sigma^2}\right)$ or Laplace $\mathsf{K}_\sigma^{\mathrm{lap}}(x,y) = \exp\left(-\frac{\|x-y\|_2}{2\sigma}\right)$ kernels usually perform well empirically. Prior work also supports rational quadratic kernels and linear combinations of kernels (Bińkowski et al., 2018), with recent studies showing that using multiple kernels can yield more powerful discrepancy measures (Chatterjee & Bhattacharya, 2025; Schrab et al., 2023; 2022). In particular for a collection of kernels $\mathcal{K}_r := \{\mathsf{K}_1, \ldots, \mathsf{K}_r\}$ the loss function can be defined as,

$$\hat{\mathcal{L}}_{\mathrm{multi}}(\boldsymbol{g}) := \sum_{m=1}^{r} \frac{w_m}{nk_n} \sum_{i=1}^{n} \sum_{j \in N_G(\mathscr{X}_n)(i)} \mathsf{H}_m\left(\boldsymbol{W}_{i,g}, \boldsymbol{W}_{j,g}\right)$$

where $\mathsf{H}_m$ is defined using $\mathsf{K}_m$ as in (2.3) and $w_m$ is the weight associated with the kernel $\mathsf{K}_m$. Moreover for computational gains it is possible to implement low-rank kernel approximations like Random Fourier Features (Rahimi & Recht, 2007).

In our experiments, we use a Gaussian kernel with bandwidth set to $\sqrt{p}$, where $p$ is the dimension of $\mathbf{Y}$, following the recommendation in (Ramdas et al., 2015). However, there is no universal consensus on how to choose the bandwidth parameter. A widely used alternative in the two-sample testing literature is the median heuristic (Gretton et al., 2012), which sets the bandwidth to the median of the pairwise distances.

To sidestep bandwidth selection altogether, some works on unconditional generative modeling with MMD employ linear combinations of kernels with manually chosen bandwidths (Bińkowski et al., 2018; Li et al., 2015). Recently, Li et al. (2017) proposed learning the bandwidth (equivalently, learning the kernel itself) via *adversarial kernel learning*, in which both the generator and the kernel are jointly optimized through a min–max formulation. An analogous extension of CGMMD is conceivable, but lies beyond the scope of the present work.

Additionally ImageNet-pretrained networks are a promising and practical choice for the kernel. Proposition 1 of (Santos et al., 2019) shows that pretrained perceptual feature maps can define characteristic kernels when their feature representations are dense in the space of continuous functions. Thus, such pretrained networks can naturally be incorporated into the CGMMD framework.

In addition to the kernel $\mathsf{K}$, CGMMD also requires choosing the number of nearest neighbors $k_n$. Choosing $k_n$ too large increases the computational overhead as the nearest-neighbor is recomputed in each batch, while choosing $k_n$ too small leads to loss of local information. In our experiments, we select $k_n$ manually based on the specific experimental setting. This practice is consistent with the observations and recommendations in Deb et al. (2020) and Huang et al. (2022a). Moreover, for ultra high-dimensional conditioning variables, constructing a $k$-NN graph directly in the raw input space may be unreliable. A standard alternative is to first learn a lower-dimensional representation (for example via a ResNet encoder) and then construct the graph in the learned feature space (Ren et al., 2016; Li et al., 2015). Motivated by the manifold hypothesis, this often yields more meaningful neighborhood structure in practice. Indeed we follow this in our STL10 experiments (see Appendix C.6), where the $k$-NN graph is built using learned embeddings rather than raw pixel values, leading to stable neighborhoods and strong sample quality.

To study the choice of kernels (characterized by the bandwidth for Gaussian and Laplace kernels) and the number of nearest neighbors $k_n$, we conduct the following ablation study. We consider the data-generating process

$$\boldsymbol{X} = (X_1, \ldots, X_5)^\top \sim \mathcal{N}_5(\mathbf{0}_5, \boldsymbol{I}_5),$$

and generate

$$\boldsymbol{Y} = X_1^2 + \exp\left(\frac{X_2 + X_3}{3}\right) + \sin(X_4 X_5) + \varepsilon, \qquad \varepsilon \sim \mathcal{N}(0, 1).$$

We use CGMMD to learn a conditional sampler that approximates the conditional distribution of $\boldsymbol{Y} \mid \boldsymbol{X}$. We then report the mean squared error (MSE) between the conditional mean and conditional standard deviation estimated from generated samples and the corresponding quantities under the true conditional distribution, evaluated at the fixed conditioning value $\boldsymbol{X} = \mathbf{0}_5$.

In Table 2 we report these values for Gaussian and Laplace kernels with kernel bandwidth $\sigma \in (10^{-4}, 10^2)$ and $k_n \in \{5, 15, 30, 45, 60, 75, 90\}$ many nearest neighbors, used for computing the loss. For this experiment we choose a training

*(a)* Gaussian – MSE of Conditional Mean

| $\sigma$ | $k_n$=5 | $k_n$=15 | $k_n$=30 | $k_n$=45 | $k_n$=60 | $k_n$=75 | $k_n$=90 |
|---|---|---|---|---|---|---|---|
| 0.0001 | 0.3743 | 1.3554 | 1.1875 | 1.1657 | 1.1732 | 1.1917 | 1.2153 |
| 0.0005 | 0.0035 | 0.0120 | 0.0195 | 0.1281 | 1.3850 | 1.2650 | 1.1820 |
| 0.0022 | 0.0073 | 0.0104 | 0.0002 | 0.0004 | 0.0038 | 0.4610 | 1.1312 |
| 0.0100 | 0.0141 | 0.0063 | 0.0009 | 0.0009 | 0.0213 | 0.8550 | 1.0946 |
| 0.0464 | 0.0279 | 0.0020 | 0.0047 | 0.0327 | 0.6478 | 1.2770 | 0.9922 |
| 0.2154 | 0.0020 | 0.0059 | 0.0708 | 0.6410 | 1.2945 | 1.1707 | 0.9130 |
| 1.0000 | 0.4108 | 0.5750 | 1.1302 | 1.1866 | 1.0802 | 0.8733 | 0.7812 |
| 4.6416 | 1.0612 | 1.0141 | 0.8623 | 0.7110 | 0.6148 | 0.5883 | 0.5877 |
| 21.5443 | 0.3734 | 0.4763 | 0.4850 | 0.3944 | 0.4462 | 0.4209 | 0.4430 |
| 100.0000 | 0.3534 | 0.3043 | 0.3819 | 0.3396 | 0.4081 | 0.3038 | 0.3567 |

*(b)* Gaussian – MSE of Conditional SD

| $\sigma$ | $k_n$=5 | $k_n$=15 | $k_n$=30 | $k_n$=45 | $k_n$=60 | $k_n$=75 | $k_n$=90 |
|---|---|---|---|---|---|---|---|
| 0.0001 | 0.4494 | 0.5700 | 0.5903 | 0.5723 | 0.5720 | 0.5669 | 0.5645 |
| 0.0005 | 0.3821 | 0.4793 | 0.5313 | 0.2796 | 0.0333 | 0.1579 | 0.2575 |
| 0.0022 | 0.3623 | 0.4048 | 0.2910 | 0.3349 | 0.1713 | 0.2127 | 0.4654 |
| 0.0100 | 0.2361 | 0.0878 | 0.0609 | 0.0416 | 0.0024 | 0.4289 | 0.5173 |
| 0.0464 | 0.0934 | 0.0438 | 0.0838 | 0.0601 | 0.1019 | 0.3714 | 0.3762 |
| 0.2154 | 0.0300 | 0.0161 | 0.0027 | 0.1122 | 0.2687 | 0.2909 | 0.2994 |
| 1.0000 | 0.0459 | 0.0909 | 0.2006 | 0.2268 | 0.2339 | 0.2538 | 0.2592 |
| 4.6416 | 0.2222 | 0.1893 | 0.2272 | 0.2026 | 0.2137 | 0.2118 | 0.2073 |
| 21.5443 | 0.2196 | 0.2177 | 0.1748 | 0.1859 | 0.1969 | 0.2000 | 0.1743 |
| 100.0000 | 0.3365 | 0.1898 | 0.2105 | 0.2358 | 0.1784 | 0.2224 | 0.2214 |

*(c)* Laplace – MSE of Conditional Mean

| $\sigma$ | $k_n$=5 | $k_n$=15 | $k_n$=30 | $k_n$=45 | $k_n$=60 | $k_n$=75 | $k_n$=90 |
|---|---|---|---|---|---|---|---|
| 0.0001 | 0.8058 | 0.7666 | 0.7467 | 0.7423 | 0.7448 | 0.7507 | 0.7597 |
| 0.0005 | 0.7177 | 0.8524 | 0.7949 | 0.7808 | 0.7794 | 0.7834 | 0.7911 |
| 0.0022 | 0.0124 | 0.0494 | 0.1031 | 0.4121 | 0.9426 | 0.9148 | 0.8874 |
| 0.0100 | 0.0023 | 0.0046 | 0.0253 | 0.0746 | 0.4292 | 1.0731 | 1.0403 |
| 0.0464 | 0.0023 | 0.0023 | 0.0177 | 0.0850 | 0.5762 | 1.1609 | 1.0748 |
| 0.2154 | 0.0067 | 0.0029 | 0.0048 | 0.2244 | 1.2316 | 1.1762 | 0.9380 |
| 1.0000 | 0.2551 | 0.4658 | 1.1840 | 1.2665 | 1.0799 | 0.8863 | 0.7984 |
| 4.6416 | 0.5159 | 0.5729 | 0.5123 | 0.5398 | 0.5051 | 0.4864 | 0.4442 |
| 21.5443 | 0.2099 | 0.1124 | 0.1589 | 0.2428 | 0.1814 | 0.1824 | 0.2165 |
| 100.0000 | 3.2993 | 0.0091 | 0.1256 | 0.0211 | 0.0908 | 1.5399 | 0.0147 |

*(d)* Laplace – MSE of Conditional SD

| $\sigma$ | $k_n$=5 | $k_n$=15 | $k_n$=30 | $k_n$=45 | $k_n$=60 | $k_n$=75 | $k_n$=90 |
|---|---|---|---|---|---|---|---|
| 0.0001 | 0.3064 | 0.2878 | 0.2632 | 0.2384 | 0.2137 | 0.1904 | 0.1694 |
| 0.0005 | 0.2498 | 0.3081 | 0.2938 | 0.2914 | 0.2915 | 0.2922 | 0.2937 |
| 0.0022 | 0.0009 | 0.0025 | 0.0137 | 0.1566 | 0.3383 | 0.3243 | 0.3153 |
| 0.0100 | 0.0246 | 0.0110 | 0.0032 | 0.0046 | 0.1709 | 0.3506 | 0.3562 |
| 0.0464 | 0.0123 | 0.0116 | 0.0043 | 0.0088 | 0.2235 | 0.3561 | 0.3601 |
| 0.2154 | 0.0163 | 0.0143 | 0.0052 | 0.0711 | 0.3256 | 0.3417 | 0.3494 |
| 1.0000 | 0.0177 | 0.0876 | 0.2296 | 0.2399 | 0.2541 | 0.2610 | 0.2593 |
| 4.6416 | 0.2043 | 0.1903 | 0.1763 | 0.1776 | 0.1782 | 0.2151 | 0.2005 |
| 21.5443 | 0.8484 | 0.2484 | 0.3155 | 0.2056 | 0.2213 | 0.2117 | 0.2150 |
| 100.0000 | 0.6223 | 2.0629 | 1.1401 | 0.5936 | 0.1712 | 0.9108 | 0.2254 |

*Table 2.* Ablation study on kernel bandwidth $\sigma$ and neighborhood size $k_n$ for Gaussian and Laplace kernels. Rows are bandwidth values; columns are number of nearest neighbors.

sample size of 5000 and report the (estimated) MSE values by averaging over 100 repetitions. Very large bandwidths are consistently worse, which is expected as they tend to wash out local information. For the MSE of conditional mean, small-to-moderate neighborhoods perform best, whereas for the MSE of conditional standard deviation, moderate to moderately large neighborhoods yield more stable estimates. For the bandwidth, the extremes consistently underperform while the middle range is almost always preferable. Taken together, a moderate bandwidth combined with a moderate neighborhood size offers a reliable default.

**Choice of batch size.** In the experimental setting of Section 5.1, we examine how batch size affects the quality of generated samples. At noise level $\sigma = 0.2$, in the top row of Figure 18, we present the scatterplots of generated (by CGMMD) samples $(\boldsymbol{Y}_1, \boldsymbol{Y}_2)$ conditional on $\boldsymbol{X} = 1$ at batch sizes $\{200, 400, 600, 800\}$ along with the conditional samples from true conditional distribution. In the second and third rows of Figure 18, we further present the scatterplots restricted to the regions $\boldsymbol{Y}_1 \leq -0.5$ and $\boldsymbol{Y}_2 \geq 3$, corresponding to low-mass tail areas.

We observe that as the batch size increases, the overall scatter decreases and the proportion of outliers in the tail regions becomes smaller, resulting in a closer match to the true helix structure. However, larger batch sizes come with additional computational cost, and across all our experiments, we have found that using a batch size of a few hundred typically provides a good balance between performance and efficiency.

**Refinement for Discrete Supports.** The estimator $\hat{\boldsymbol{g}}$ based on $\widehat{\mathrm{ECMMD}}$ in (3.2) is well-defined for both continuous and discrete $P_{\boldsymbol{X}}$. However, for discrete supports, nearest neighbor estimates may introduce redundancy or omit relevant structure depending on $k_n$. To mitigate this, when $P_{\boldsymbol{X}}$ has discrete support we refine the empirical objective as:

$$\hat{\mathcal{L}}_D(\boldsymbol{g}) := \frac{1}{n} \sum_{i=1}^{n} \frac{1}{|\{j : \boldsymbol{X}_j = \boldsymbol{X}_i\}|} \sum_{j:\boldsymbol{X}_j = \boldsymbol{X}_i} \mathsf{H}(\boldsymbol{W}_{i,\boldsymbol{g}}, \boldsymbol{W}_{j,\boldsymbol{g}}),$$

and obtain the generator via $\min_{\boldsymbol{g} \in \mathcal{G}} \hat{\mathcal{L}}_D(\boldsymbol{g})$. Such refinements for discrete supports are also discussed in prior work on nearest neighbor methods (Deb et al., 2020; Huang et al., 2022a). We apply the proposed objective to generate digit images conditioned on class labels using the MNIST dataset. Figure 19 shows the average of the generated samples for each digit class, indicating that the outputs are consistent, with non-trivial variation across individual samples.

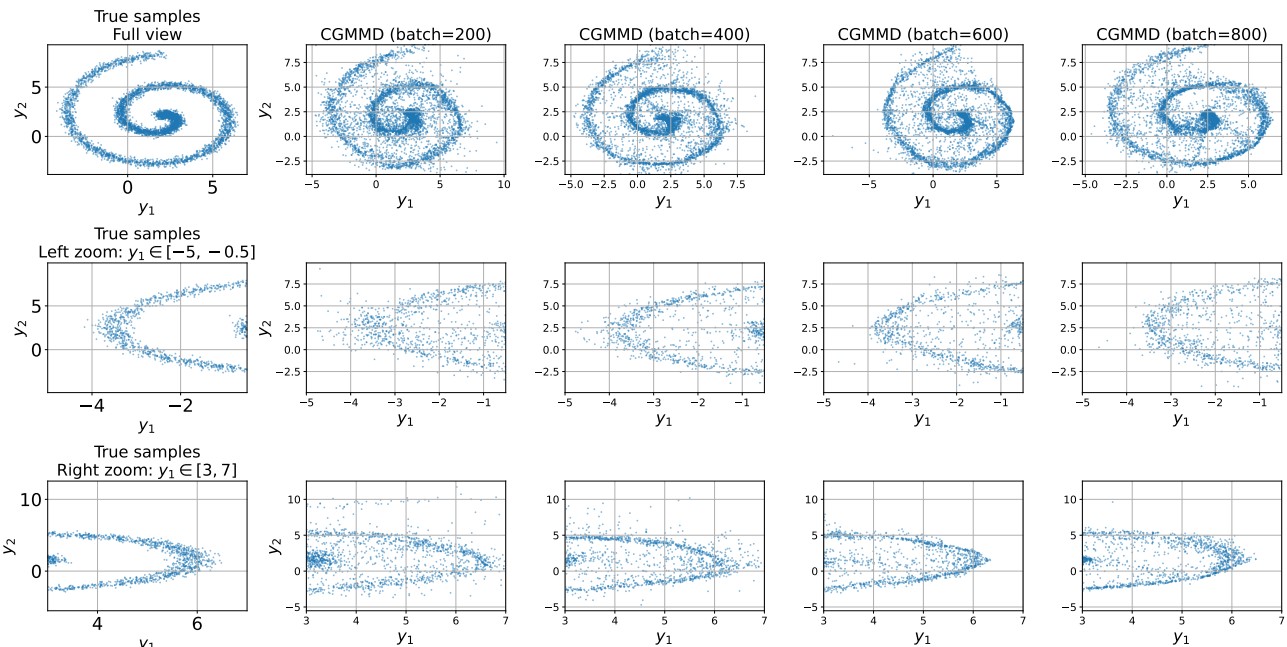

*Figure 18.* Effect of batch size on the generation quality of CGMMD in the simulation setting of Section 5.1.

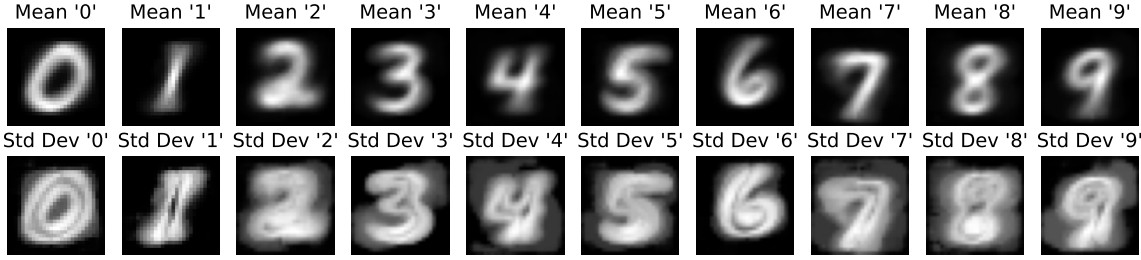

*Figure 19.* Mean and standard deviation of generated digit images.

**Computational Complexity.** For $k_n = O(1)$, the estimator in (3.1) can be computed in near-linear time $O(n \log n)$ by first constructing the $k$-NN graph in $O(n \log n)$ time (Friedman et al., 1977), followed by an $O(n)$ summation. This is substantially more efficient than standard MMD objectives, which require $O(n^2)$ time. Moreover, it may be insightful to leverage approximate nearest neighbor methods (Douze et al., 2024; Malkov & Yashunin, 2020) to accelerate training. In our experiments, however, we implement a helper function that computes nearest neighbors via brute-force search, which incurs a computational cost of $O(B^2)$ where $B$ denotes the batch size. This can be improved to $O(B \log B)$ by implementing efficient nearest-neighbor search or approximate nearest neighbor methods. While our focus is on conditional generation, the same objective can be applied to unconditional generation by taking $\boldsymbol{X}$ independent of $\boldsymbol{Y}$ and solving the corresponding optimization problem. Although outside the scope of this work, this approach may offer improved computational efficiency at the cost of sample quality.

### D.1. Derandomized CGMMD

Recall the ECMMD-based objective for CGMMD from Section 3. In the empirical objective from (3.1), we introduce additional noise variables $\boldsymbol{\eta}_1, \ldots, \boldsymbol{\eta}_n \sim P_{\boldsymbol{\eta}}$ to train the generative model $\boldsymbol{g}$. However, this introduces an extra source of randomness in the training procedure. As a result, different runs of the same algorithm on the same observed dataset may produce different conditional samplers, thereby introducing inconsistencies in the learned model due to finite-sample

variability.

To mitigate this issue, in this section we introduce a derandomization procedure, albeit at the cost of additional computational overhead.

Note that the noise variables are sampled from a known distribution $P_{\boldsymbol{\eta}}$, which is typically chosen to be either Gaussian or Uniform. Leveraging this, we propose the following algorithm to modify the empirical loss $\hat{\mathcal{L}}$ accordingly.

1. Fix $M_n \geq 1$. Then generate i.i.d. samples $\{\boldsymbol{\eta}_{i,1}, \ldots, \boldsymbol{\eta}_{i,M_n} : 1 \leq i \leq n\} \sim P_{\boldsymbol{\eta}}$.

2. Let $\boldsymbol{W}_{i,m,\boldsymbol{g}} = (\boldsymbol{Y}_i, \boldsymbol{g}(\boldsymbol{\eta}_{i,m}, \boldsymbol{X}_i))$, for all $1 \leq i \leq n$ and $1 \leq m \leq M_n$. Now define,

$$\hat{\mathcal{L}}_{\mathrm{DR}}(\boldsymbol{g}) := \frac{1}{nk_n} \sum_{i=1}^{n} \sum_{j \in N_{G(\mathscr{X}_n)}(i)} \frac{1}{M_n} \sum_{m=1}^{M_n} \mathsf{H}\left(\boldsymbol{W}_{i,m,\boldsymbol{g}}, \boldsymbol{W}_{j,m,\boldsymbol{g}}\right).$$

3. Approximate the conditional sampler by solving $\hat{\boldsymbol{g}}_{\mathrm{DR}} = \arg\min_{\boldsymbol{g} \in \boldsymbol{\mathcal{G}}} \hat{\mathcal{L}}_{\mathrm{DR}}(\boldsymbol{g})$.

Note that for $M_n = 1$, the derandomized objective $\hat{\mathcal{L}}_{\mathrm{DR}}$ reduces to the original empirical loss $\hat{\mathcal{L}}$ from (3.1). The inner averaging over the generated noise variables is expected to reduce the variance introduced by the stochasticity of the noise, thereby mitigating the additional randomness in the training procedure.

Moreover, Theorem 5.2 from Chatterjee et al. (2024) shows that, under mild conditions (in fact, without imposing any restrictions on the choice of $M_n$), the derandomized loss $\hat{\mathcal{L}}_{\mathrm{DR}}$ converges to the true ECMMD objective. Therefore, we can expect similar convergence guarantees as those established in Theorem 4.4 to hold in this setting as well.

# E. Convergence of the Empirical Sampler

In this section we establish convergence of the empirical sampler from (3.2) under more general settings. For the reader's convenience we briefly recall the notations, assumptions and details about the class of neural networks from Section 4.

Recall that we observe samples $\{(\boldsymbol{Y}_i, \boldsymbol{X}_i) : 1 \leq i \leq n\}$ from a joint distribution $P_{\boldsymbol{Y}\boldsymbol{X}}$ on $\mathbb{R}^p \times \mathbb{R}^d$ such that the regular conditional distribution $P_{\boldsymbol{Y}|\boldsymbol{X}}$ exists. Our aim is to generate samples from this conditional distribution. Towards that, by the *noise outsourcing lemma* (see Theorem 5.10 from Kallenberg (2021) and Lemma 2.1 from Zhou et al. (2023)) we know there exists a measurable function $\bar{\boldsymbol{g}}$ such that $P_{\bar{\boldsymbol{g}}(\boldsymbol{\eta}, \boldsymbol{X})|\boldsymbol{X}} = P_{\boldsymbol{Y}|\boldsymbol{X}}$ for $\boldsymbol{\eta}$ generated independently from $\mathrm{N}_m\left(\boldsymbol{0}, \boldsymbol{I}_m\right)$ for any $m \geq 1$. From Section 3 recall that to estimate the conditional sampler $\bar{\boldsymbol{g}}$, we consider the ECMMD from Chatterjee et al. (2024) as a discrepancy measure. In particular we take a kernel K satisfying the following.

**Assumption E.1.** The kernel $\mathsf{K} : \mathbb{R}^p \times \mathbb{R}^p \to \mathbb{R}$ is positive definite and satisfies the following:

1. The kernel K is uniformly bounded, that is $\|\mathsf{K}\|_\infty < K$ for some $K > 0$ and Lipschitz continuous with Lipschitz constant $L_{\mathsf{K}}$.

2. The kernel mean embedding $\mu : \mathcal{P}(\mathcal{Y}) \to \mathcal{H}$ is a one-to-one (injective) function. This is also known as the *characteristic kernel* property (Sriperumbudur et al., 2011).

Now fix $m \geq 1$, generate independent samples $\boldsymbol{\eta}_1, \boldsymbol{\eta}_2, \ldots, \boldsymbol{\eta}_n$ from $\mathrm{N}_m\left(\boldsymbol{0}, \boldsymbol{I}_m\right)$ and take a class of neural networks $\mathcal{G}$ (defined below). Next, we construct the $k_n$-nearest neighbor graph $G\left(\mathscr{X}_n\right)$ on the samples $\mathscr{X}_n := \{\boldsymbol{X}_1, \ldots, \boldsymbol{X}_n\}$ with respect to the $\|\cdot\|_2$. For any $\boldsymbol{g} \in \mathcal{G}$ let $\boldsymbol{W}_{i,\boldsymbol{g}} = (\boldsymbol{Y}_i, \boldsymbol{g}\left(\boldsymbol{\eta}_i, \boldsymbol{X}_i\right))$ for all $i \in [n]$ and define,

$$\mathsf{H}\left(\boldsymbol{W}_{i,\boldsymbol{g}}, \boldsymbol{W}_{j,\boldsymbol{g}}\right) := \mathsf{K}\left(\boldsymbol{Y}_i, \boldsymbol{Y}_j\right) - \mathsf{K}\left(\boldsymbol{Y}_i, \boldsymbol{g}\left(\boldsymbol{\eta}_j, \boldsymbol{X}_j\right)\right) - \mathsf{K}\left(\boldsymbol{g}\left(\boldsymbol{\eta}_i, \boldsymbol{X}_i\right), \boldsymbol{Y}_j\right) + \mathsf{K}\left(\boldsymbol{g}\left(\boldsymbol{\eta}_i, \boldsymbol{X}_i\right), \boldsymbol{g}\left(\boldsymbol{\eta}_j, \boldsymbol{X}_j\right)\right)$$

for all $1 \leq i \neq j \leq n$ and for any $\boldsymbol{g} \in \mathcal{G}$ take,

$$\hat{\mathcal{L}}\left(\boldsymbol{g}\right) := \frac{1}{n k_n} \sum_{i=1}^n \sum_{j \in N_{G(\mathscr{X}_n)}(i)} \mathsf{H}\left(\boldsymbol{W}_{i,\boldsymbol{g}}, \boldsymbol{W}_{j,\boldsymbol{g}}\right).$$

With the above definition, we estimate the true function $\bar{\boldsymbol{g}}$ as,

$$\hat{\boldsymbol{g}} := \arg\min_{\boldsymbol{g} \in \mathcal{G}} \mathcal{L}\left(\boldsymbol{g}\right)$$

For establishing convergence guarantees for the estimated conditional sampler $\hat{\boldsymbol{g}}$ we make the following technical assumptions.

**Assumption E.2.** The following conditions on $P_{\boldsymbol{Y}\boldsymbol{X}}$, the kernel K, the true conditional sampler $\bar{\boldsymbol{g}}$ and the class $\mathcal{G}$ holds.

1. $P_{\boldsymbol{X}}$ is supported on $\mathcal{X} \subseteq \mathbb{R}^d$ for some $d > 0$ and $\|\boldsymbol{X}_1 - \boldsymbol{X}_2\|_2$ has a continuous distribution for $\boldsymbol{X}_1, \boldsymbol{X}_2 \sim P_{\boldsymbol{X}}$.

2. There exists $\alpha, C_1, C_2 > 0$ such that for $\boldsymbol{X} \sim P_{\boldsymbol{X}}$,

$$\mathbb{P}\left(\|\boldsymbol{X}\|_2 > t\right) \leq C_1 \exp\left(-C_2 t^\alpha\right), \quad \forall t > 0. \tag{E.1}$$

3. The target conditional sampler $\bar{\boldsymbol{g}} : \mathbb{R}^m \times \mathbb{R}^d \to \mathbb{R}^p$ is continuous with $\|\bar{\boldsymbol{g}}\|_\infty \leq C_0$ for some constant $C_0 > 0$.

4. For any $\boldsymbol{g} \in \mathcal{G}$ consider $h_{\boldsymbol{g}}(x) = \mathbb{E}\left[\mathsf{K}(\boldsymbol{Y}, \cdot) - \mathsf{K}\left(\boldsymbol{g}\left(\boldsymbol{\eta}, \boldsymbol{X}\right), \cdot\right) | \boldsymbol{X} = x\right]$ and assume that there exists $\beta_1, \beta_2 > 0$ such that,

$$\left|\langle h_{\boldsymbol{g}}(\boldsymbol{x}), h_{\boldsymbol{g}}(\boldsymbol{x}_1) - h_{\boldsymbol{g}}(\boldsymbol{x}_2)\rangle\right| \leq C_3 \left(1 + \|\boldsymbol{x}\|_2^{\beta_1} + \|\boldsymbol{x}_1\|_2^{\beta_1} + \|\boldsymbol{x}_2\|_2^{\beta_1}\right) \|\boldsymbol{x}_1 - \boldsymbol{x}_2\|_2^{\beta_2}, \tag{E.2}$$

for all $\boldsymbol{x}, \boldsymbol{x}_1, \boldsymbol{x}_2 \in \mathcal{X}$ where $C_3$ is a constant independent of $\boldsymbol{g}$.

We take $\mathcal{G}$ to be a class of neural networks with the following details.

**Details of $\mathcal{G}$:** Let $\mathcal{G} = \mathcal{G}_{\mathcal{H},\mathcal{W},\mathcal{S},\mathcal{B}}$ be the set of ReLU neural networks $\boldsymbol{g} : \mathbb{R}^m \times \mathbb{R}^d \to \mathbb{R}^p$ with depth $\mathcal{H}$, width $\mathcal{W}$, size $\mathcal{S}$ and $\|\boldsymbol{g}\|_\infty \leq \mathcal{B}$. In particular, $\mathcal{H}$ denotes the number of hidden layers and $(w_0, w_2, \ldots, w_{\mathcal{H}})$ denotes the width of each layer where $w_0 = d+m$ and $w_{\mathcal{H}} = p$ denotes the input and output dimension respectively. We take $\mathcal{W} = \max\{w_0, w_1, \ldots, w_{\mathcal{H}}\}$. Finally size $\mathcal{S} = \sum_{i=1}^{\mathcal{H}} w_i(w_{i-1} + 1)$ refers to the total number of parameters of the network.

Moreover, we make the following assumptions about the parameters of the class $\mathcal{G}$.

**Assumption E.3.** The network parameters of $\mathcal{G}$ satisfies $\mathcal{H}, \mathcal{W} \to \infty$ such that,

$$\mathcal{H}\mathcal{W} \to \infty \text{ and } \frac{\mathcal{B}^2 \mathcal{H}\mathcal{S} \log \mathcal{S} \log n}{n} \to 0$$

as $n \to \infty$. Additionally $\mathcal{B} \geq C_0$ where $C_0$ is defined in Assumption E.2.

Before stating our main result, for a function $f$, uniformly continuous on a set $E$, define the optimal modulus of continuity on the set $E$ as,

$$\omega_f^E(r) := \sup\{\|f(\boldsymbol{x}) - f(\boldsymbol{y})\| : \|\boldsymbol{x} - \boldsymbol{y}\| \leq r, \boldsymbol{x}, \boldsymbol{y} \in E\}.$$

We are now ready to state our result on convergence of the empirical sampler.

**Theorem E.4.** *Adopt Assumption E.1, Assumption E.3 and Assumption E.2. Take $\varepsilon_n = \left(\frac{k_n \log n}{n}\right)^{1/d} (\log n)^{1/\alpha}$ and,*

$$\nu_n = \begin{cases} \frac{k_n \log n}{n}(\log n)^{2\beta_2/\alpha} & \text{if } d < 2\beta_2 \\ \frac{k_n \log n}{n}(\log n)^{1+d/\alpha} & \text{if } d = 2\beta_2 \\ \left(\frac{k_n \log n}{n}\right)^{2\beta_2/d}(\log n)^{2\beta_2/\alpha} & \text{if } d > 2\beta_2. \end{cases}$$

*Let $k_n = o(\sqrt{n})$, then for any $\delta > 0$ with $E = [-R, R]^{d+m}$,*

$$\mathcal{L}(\hat{\boldsymbol{g}}) \lesssim_{\boldsymbol{\theta}} \frac{1}{\sqrt{n}} + \sqrt{\frac{\mathcal{B}^2 \mathcal{H}\mathcal{S} \log \mathcal{S} \log n}{n}} + \varepsilon_n^{\beta_2} + \sqrt{\nu_n}$$

$$+ 1 - \Phi(R)^m (1 - C_1 \exp(-C_2 R^\alpha)) + \sqrt{d+m}\,\omega_{\bar{\boldsymbol{g}}}^E\left(2R(\mathcal{H}\mathcal{W})^{-\frac{1}{d+m}}\right) + \sqrt{\frac{\log(1/\delta)}{n}}$$

*for all $R > 0$ with probability atleast $1 - \delta$.*

The above theorem provides finite sample bounds on the loss incurred by using the estimated conditional sampler $\hat{\boldsymbol{g}}$. We can use the explicit bound from Theorem E.4 to confirm that the conditional distribution induced by the empirical sampler indeed converge to the true conditional distribution.

**Corollary E.5.** *Adopt Assumption E.1, Assumption E.3 and Assumption E.2. Then for $k_n = o(\sqrt{n})$,*

$$\mathbb{E}\left[\text{MMD}^2\left[\mathcal{F}, P_{\hat{\boldsymbol{g}}(\boldsymbol{\eta},\boldsymbol{X})|\boldsymbol{X}}, P_{\bar{\boldsymbol{g}}(\boldsymbol{\eta},\boldsymbol{X})|\boldsymbol{X}}\right] \mid \hat{\boldsymbol{g}}\right] \to 0 \text{ a.s.}.$$

Finally to complete this section on convergence guarantees for the empirical sampler, using DCT the result from Corollary E.5 can be relaxed to claim,

$$\mathbb{E}\left[\text{MMD}^2\left[\mathcal{F}, P_{\hat{\boldsymbol{g}}(\boldsymbol{\eta},\boldsymbol{X})|\boldsymbol{X}}, P_{\bar{\boldsymbol{g}}(\boldsymbol{\eta},\boldsymbol{X})|\boldsymbol{X}}\right]\right] \to 0.$$

## E.1. Proof of Theorem E.4

For simplicity we will assume that $p = 1$. The proof for general $p > 1$ is similar but with additional notational complexities. To begin with by Proposition 2.3 from Chatterjee et al. (2024) we know that $\mathcal{L}(\bar{\boldsymbol{g}}) = 0$ for the true conditional sampler $\bar{\boldsymbol{g}}$. Then we get the decomposition,

$$\mathcal{L}(\hat{\boldsymbol{g}}) = \mathcal{L}(\hat{\boldsymbol{g}}) - \mathcal{L}(\bar{\boldsymbol{g}}) \leq \sup_{\boldsymbol{g} \in \mathcal{G}}\left|\hat{\mathcal{L}}(\boldsymbol{g}) - \mathcal{L}(\boldsymbol{g})\right| + \left|\hat{\mathcal{L}}(\tilde{\boldsymbol{g}}) - \mathcal{L}(\tilde{\boldsymbol{g}})\right| + |\mathcal{L}(\tilde{\boldsymbol{g}}) - \mathcal{L}(\bar{\boldsymbol{g}})|$$

for any $\tilde{g}$ in $\mathcal{G}$. We can now relax the upper bound to get,

$$\mathcal{L}\left(\hat{g}\right) \leq \underbrace{2 \sup_{g \in \mathcal{G}} \left|\hat{\mathcal{L}}(g) - \mathcal{L}(g)\right|}_{T_1} + \underbrace{\inf_{\tilde{g} \in \mathcal{G}} |\mathcal{L}(\tilde{g}) - \mathcal{L}(\bar{g})|}_{T_2} \tag{E.3}$$

We will bound terms $T_1$ and $T_2$ individually. We first start with $T_2$.

**Lemma E.6.** *Adopt the conditions and notations of Theorem E.4 and recall $T_2$ from (E.3). Then for any $R > 0$,*

$$T_2 \lesssim_{\mathsf{K}} 1 - \Phi\left(R\right)^m \left(1 - C_1 \exp\left(-C_2 R^\alpha\right)\right) + \sqrt{d+m}\,\omega_{\bar{g}}^E \left(2R\left(\mathcal{H}\mathcal{W}\right)^{-\frac{1}{d+m}}\right)$$

*where $\omega_{\bar{g}}^E \left(\cdot\right)$ is the optimal modulus of continuity of $\bar{g}$ on the subset $E = [-R, R]^{d+m}$.*

Next we bound the term $T_1$ from (E.3). To that end we start by decomposing $T_1$. Note that,

$$T_1 \leq \underbrace{\sup_{g \in \mathcal{G}} \left|\hat{\mathcal{L}}\left(g\right) - \mathbb{E}\left[\hat{\mathcal{L}}\left(g\right) \mid \mathscr{X}_n\right]\right|}_{T_{1,1}} + \underbrace{\sup_{g \in \mathcal{G}} \left|\mathbb{E}\left[\hat{\mathcal{L}}\left(g\right) \mid \mathscr{X}_n\right] - \frac{1}{n}\sum_{i=1}^{n} \|h_g\left(\mathbf{X}_i\right)\|_{\mathcal{K}}^2\right|}_{T_{1,2}}$$

$$+ \underbrace{\sup_{g \in \mathcal{G}} \left|\frac{1}{n}\sum_{i=1}^{n} \|h_g\left(\mathbf{X}_i\right)\|_{\mathcal{K}}^2 - \mathcal{L}\left(g\right)\right|}_{T_{1,3}}. \tag{E.4}$$

In the following we bound each of the terms $T_{1,1}, T_{1,2}$ and $T_{1,3}$ separately. First we bound the term $T_{1,1}$.

**Lemma E.7.** *Adopt the conditions and notations of Theorem E.4 and recall $T_{1,1}$ from (E.4). Then for any $\delta > 0$, with probability at least $1 - \delta$,*

$$T_{1,1} \lesssim_{\mathsf{K},d} \frac{1}{\sqrt{n}} + \sqrt{\frac{\mathcal{B}^2 \mathcal{H}\mathcal{S} \log \mathcal{S} \log n}{n}} + \sqrt{\frac{\log\left(2/\delta\right)}{n}}.$$

Next we bound the term $T_{1,2}$.

**Lemma E.8.** *Adopt the conditions and notations of Theorem E.4 and recall $T_{1,2}$ from (E.4). Recall $\varepsilon_n = \left(\frac{k_n \log n}{n}\right)^{1/d} (\log n)^{1/\alpha}$ and,*

$$\nu_n = \begin{cases} \frac{k_n \log n}{n}(\log n)^{2\beta_2/\alpha} & \text{if } d < 2\beta_2 \\ \frac{k_n \log n}{n}(\log n)^{1+d/\alpha} & \text{if } d = 2\beta_2 \\ \left(\frac{k_n \log n}{n}\right)^{2\beta_2/d}(\log n)^{2\beta_2/\alpha} & \text{if } d > 2\beta_2. \end{cases}$$

*Then for $k_n = o\left(\sqrt{n}\right)$ and any $\delta > 0$, with probability $1 - \delta$,*

$$T_{1,2} \lesssim_{d,\mathsf{K}} \frac{1}{n^2} + \varepsilon_n^{\beta_2} + \sqrt{\nu_n} + \sqrt{\frac{\log\left(1/\delta\right)}{n}}.$$

Finally we bound the remaining term $T_{1,3}$.

**Lemma E.9.** *Adopt the conditions and notations of Theorem E.4 and recall $T_{1,3}$ from (E.4). Then for any $\delta > 0$, with probability at least $1 - \delta$,*

$$T_{1,3} \lesssim_{\mathsf{K}} \frac{1}{\sqrt{n}} + \sqrt{\frac{\mathcal{B}^2 \mathcal{H}\mathcal{S} \log \mathcal{S} \log n}{n}} + \sqrt{\frac{\log\left(1/\delta\right)}{n}}.$$

Now to complete the proof of Theorem E.4 we combine the bound from (E.3) and the bounds from Lemma E.6, Lemma E.7, Lemma E.8 and Lemma E.9 to conclude,

$$
\mathcal{L}(\hat{\boldsymbol{g}}) \lesssim_{d,\mathsf{K}} \frac{1}{\sqrt{n}} + \sqrt{\frac{\mathcal{B}^2 \mathcal{H} \mathcal{S} \log \mathcal{S} \log n}{n} + \varepsilon_n^{2\beta_2} + \sqrt{\nu_n}}
$$

$$
+ 1 - \Phi(R)^m (1 - C_1 \exp(-C_2 R^\alpha)) + \sqrt{d+m}\, \omega_{\bar{\boldsymbol{g}}}^E \left( 2R\mathcal{H}^{-\frac{1}{d+m}} \mathcal{W}^{-\frac{1}{d+m}} \right) + \sqrt{\frac{\log(1/\delta)}{n}}
$$

for any $R > 0$ with probability atleast $1 - \delta$.

### E.1.1. PROOF OF LEMMA E.6

Recalling the definition of $\mathcal{L}$ from (2.2), for any $\tilde{\boldsymbol{g}} \in \mathcal{G}$ we get,

$$
|\mathcal{L}(\tilde{\boldsymbol{g}}) - \mathcal{L}(\bar{\boldsymbol{g}})| \lesssim \mathbb{E}\left[|\mathsf{K}(\boldsymbol{Y}, \bar{\boldsymbol{g}}(\boldsymbol{\eta}, \boldsymbol{X})) - \mathsf{K}(\boldsymbol{Y}, \tilde{\boldsymbol{g}}(\boldsymbol{\eta}, \boldsymbol{X}))|\right]
$$
$$
+ \mathbb{E}\left[|\mathsf{K}(\bar{\boldsymbol{g}}(\boldsymbol{\eta}, \boldsymbol{X}), \bar{\boldsymbol{g}}(\boldsymbol{\eta}', \boldsymbol{X})) - \mathsf{K}(\tilde{\boldsymbol{g}}(\boldsymbol{\eta}, \boldsymbol{X}), \tilde{\boldsymbol{g}}(\boldsymbol{\eta}', \boldsymbol{X}))|\right]
$$

where $\boldsymbol{\eta}, \boldsymbol{\eta}' \sim \mathrm{N}_m(\boldsymbol{0}, \boldsymbol{I}_m)$ are generated independent of $\boldsymbol{X}$. Now take $E = [-R, R]^{d+m}$ for any $R > 0$. Then recalling the bound on $\mathsf{K}$ from Assumption E.1 we can now relax the above upper bound as,

$$
\left| \mathcal{L}(\tilde{\boldsymbol{g}}) - \mathcal{L}(\bar{\boldsymbol{g}}) \right| \lesssim \mathbb{P}((\boldsymbol{\eta}, \boldsymbol{X}) \in E^c)
$$
$$
+ \mathbb{E}\left[|\mathsf{K}(\boldsymbol{Y}, \bar{\boldsymbol{g}}(\boldsymbol{\eta}, \boldsymbol{X})) - \mathsf{K}(\boldsymbol{Y}, \tilde{\boldsymbol{g}}(\boldsymbol{\eta}, \boldsymbol{X}))| \mathbb{1}\{(\boldsymbol{\eta}, \boldsymbol{X}) \in E\}\right]
$$
$$
+ \mathbb{E}\left[|\mathsf{K}(\bar{\boldsymbol{g}}(\boldsymbol{\eta}, \boldsymbol{X}), \bar{\boldsymbol{g}}(\boldsymbol{\eta}', \boldsymbol{X})) - \mathsf{K}(\tilde{\boldsymbol{g}}(\boldsymbol{\eta}, \boldsymbol{X}), \tilde{\boldsymbol{g}}(\boldsymbol{\eta}', \boldsymbol{X}))| \mathbb{1}\{(\boldsymbol{\eta}, \boldsymbol{X}), (\boldsymbol{\eta}', \boldsymbol{X}) \in E\}\right]
$$

Next we use the Lipschitz property of $\mathsf{K}$ from Assumption E.1 to further relax the above bound as,

$$
\left| \mathcal{L}(\tilde{\boldsymbol{g}}) - \mathcal{L}(\bar{\boldsymbol{g}}) \right| \lesssim_{\mathsf{K}} \mathbb{P}((\boldsymbol{\eta}, \boldsymbol{X}) \in E^c) + \mathbb{E}\left[\|\bar{\boldsymbol{g}}(\boldsymbol{\eta}, \boldsymbol{X}) - \tilde{\boldsymbol{g}}(\boldsymbol{\eta}, \boldsymbol{X})\|_2 \mathbb{1}\{(\boldsymbol{\eta}, \boldsymbol{X}) \in E\}\right]
$$
$$
\lesssim_{\mathsf{K}} \mathbb{P}((\boldsymbol{\eta}, \boldsymbol{X}) \in E^c) + \|(\tilde{\boldsymbol{g}} - \bar{\boldsymbol{g}}) \mathbb{1}_E\|_\infty \tag{E.5}
$$

Now by (E.1) and recalling that $\boldsymbol{\eta}$ is independent of $\boldsymbol{X}$ we know,

$$
\mathbb{P}((\boldsymbol{\eta}, \boldsymbol{X}) \in E^c) \leq 1 - \Phi(R)^m (1 - C_1 \exp(-C_2 R^\alpha)). \tag{E.6}
$$

Hence continuing the trail of inequalities from (E.5) and recalling that the choice of $\tilde{\boldsymbol{g}} \in \mathcal{G}$ was arbitrary we can show,

$$
\inf_{\tilde{\boldsymbol{g}} \in \mathcal{G}} \left| \mathcal{L}(\tilde{\boldsymbol{g}}) - \mathcal{L}(\boldsymbol{g}) \right| \lesssim_{\mathsf{K}} 1 - \Phi(R)^m (1 - C_1 \exp(-C_2 R^\alpha)) + \inf_{\tilde{\boldsymbol{g}} \in \mathcal{G}} \|(\tilde{\boldsymbol{g}} - \bar{\boldsymbol{g}}) \mathbb{1}_E\|_\infty
$$

Now by Assumption E.2 recall that the target conditional sampler $\bar{\boldsymbol{g}}$ is continuous and $\|\bar{\boldsymbol{g}}\|_\infty \leq C_0$. Now for all $n$ large enough, take $L = \lfloor \sqrt{\mathcal{H}} \rfloor$ and $N = \lfloor \sqrt{\mathcal{W}} \rfloor$. Then by Theorem 4.3 from Shen et al. (2020) there exists a ReLU network $\tilde{\boldsymbol{g}}_0$ with depth $12L + 14 + 2(d+m)$, maximum width $3^{d+m+3} \max\left\{ (d+m) \left\lfloor N^{\frac{1}{d+m}} \right\rfloor, N+1 \right\}$ and $\|\tilde{\boldsymbol{g}}_0\|_\infty \leq C_0$ such that,

$$
\|(\tilde{\boldsymbol{g}}_0 - \bar{\boldsymbol{g}}) \mathbb{1}_E\|_\infty \lesssim \sqrt{d+m}\, \omega_{\bar{\boldsymbol{g}}}^E \left( 2R N^{-\frac{2}{d+m}} L^{-\frac{2}{d+m}} \right)
$$

where $\omega_{\bar{\boldsymbol{g}}}^E(\cdot)$ is the optimal modulus of continuity of $\bar{\boldsymbol{g}}$ on the set $E$ (note that this is well defined since $\bar{\boldsymbol{g}}$ is uniformly continuous on E). Now note that by definition of $L$ and $N$, we can easily extend $\tilde{\boldsymbol{g}}_0$ to a ReLU network $\tilde{\boldsymbol{g}} \in \mathcal{G}$ such that $\tilde{\boldsymbol{g}}_0 = \tilde{\boldsymbol{g}}$. Hence,

$$
\inf_{\tilde{\boldsymbol{g}} \in \mathcal{G}} \|(\tilde{\boldsymbol{g}} - \bar{\boldsymbol{g}}) \mathbb{1}_E\|_\infty \leq \|(\tilde{\boldsymbol{g}}_0 - \bar{\boldsymbol{g}}) \mathbb{1}_E\|_\infty \lesssim \sqrt{d+m}\, \omega_{\bar{\boldsymbol{g}}}^E \left( 2R\mathcal{H}^{-\frac{1}{d+m}} \mathcal{W}^{-\frac{1}{d+m}} \right).
$$

E.1.2. PROOF OF LEMMA E.7

From Assumption E.1 recall K is bounded and Lipschitz. Hence applying Corollary G.3, we get that,

$$
\mathbb{P}\left[T_{1,1} \lesssim_{\mathsf{K}} \frac{1}{n}\mathbb{E}\left[\sup_{\boldsymbol{g}\in\boldsymbol{\mathcal{G}}}\sum_{i=1}^{n}\sqrt{1+\frac{d_i\left(\mathscr{X}_n\right)}{k_n}}Z_i\boldsymbol{g}\left(\boldsymbol{\eta}_i,\boldsymbol{X}_i\right)\mid\mathscr{X}_n\right]+\sqrt{\frac{\log\left(2/\delta\right)}{n}}\mid\mathscr{X}_n\right]\geq 1-\delta
$$

where $Z_1,\ldots,Z_n$ are generated independently from $N(0,1)$ and $d_i\left(\mathscr{X}_n\right)$ is the degree (in-degree + out-degree) of $\boldsymbol{X}_i$ in $G\left(\mathscr{X}_n\right)$ for all $i\in[n]$. A simple application of tower property of conditional expectation shows that with probability at least $1-\delta$,

$$
T_{1,1}\lesssim_{\mathsf{K}}\frac{1}{n}\mathbb{E}\left[\sup_{\boldsymbol{g}\in\boldsymbol{\mathcal{G}}}\sum_{i=1}^{n}\sqrt{1+\frac{d_i\left(\mathscr{X}_n\right)}{k_n}}Z_i\boldsymbol{g}\left(\boldsymbol{\eta}_i,\boldsymbol{X}_i\right)\mid\mathscr{X}_n\right]+\sqrt{\frac{\log\left(2/\delta\right)}{n}}. \tag{E.7}
$$

Now consider the set,

$$
\boldsymbol{\mathcal{G}}_n:=\{(\boldsymbol{g}\left(\boldsymbol{\eta}_1,\boldsymbol{X}_1\right),\ldots,\boldsymbol{g}\left(\boldsymbol{\eta}_n,\boldsymbol{X}_n\right)):\boldsymbol{g}\in\boldsymbol{\mathcal{G}}\}
$$

and for any $\boldsymbol{v}_1=(v_{1,1},\ldots,v_{n,1})$ and $\boldsymbol{v}_2=(v_{1,2},\ldots,v_{n,2})$ consider the empirical distance,

$$
d_{n,\infty}\left(\boldsymbol{v}_1,\boldsymbol{v}_2\right):=\max_{i=1}^{n}\left|v_{i,1}-v_{i,2}\right|. \tag{E.8}
$$

Fix $\varepsilon>0$ and take $\mathcal{C}_{n,\varepsilon}$ to be the covering number of $\boldsymbol{\mathcal{G}}_n$ at scale $\varepsilon$ with respect to the empirical distance $d_{n,\infty}$ and let $\boldsymbol{\mathcal{G}}_{n,\varepsilon}$ to be one such covering set. By Lemma 2.1 from Jaffe et al. (2020) we know that,

$$
d_i\left(\mathscr{X}_n\right)\lesssim_d k_n\text{ for all }i\in[n]. \tag{E.9}
$$

Then by considering elements in $\boldsymbol{\mathcal{G}}_{n,\varepsilon}$ we can now easily show,

$$
\frac{1}{n}\mathbb{E}\left[\sup_{\boldsymbol{g}\in\boldsymbol{\mathcal{G}}}\sum_{i=1}^{n}\sqrt{1+\frac{d_i\left(\mathscr{X}_n\right)}{k_n}}Z_i\boldsymbol{g}\left(\boldsymbol{\eta}_i,\boldsymbol{X}_i\right)\mid\mathscr{X}_n\right]
$$
$$
\lesssim_d\varepsilon+\frac{1}{n}\mathbb{E}\left[\sup_{\boldsymbol{v_g}\in\boldsymbol{\mathcal{G}}_{n,\varepsilon}}\sum_{i=1}^{n}\sqrt{1+\frac{d_i\left(\mathscr{X}_n\right)}{k_n}}Z_i\boldsymbol{v_{g,i}}\mid\mathscr{X}_n\right] \tag{E.10}
$$

where $\boldsymbol{v_g}=(\boldsymbol{v_{g,1}},\ldots,\boldsymbol{v_{g,n}})$ with $\boldsymbol{v_{g,i}}=\boldsymbol{g}\left(\boldsymbol{\eta}_i,\boldsymbol{X}_i\right)$ for all $i\in[n]$ and $g\in\boldsymbol{\mathcal{G}}$. Now by applying Lemma H.1 and once again using the bound from (E.9) we get,

$$
\frac{1}{n}\mathbb{E}\left[\sup_{\boldsymbol{v_g}\in\boldsymbol{\mathcal{G}}_{n,\varepsilon}}\sum_{i=1}^{n}\sqrt{1+\frac{d_i\left(\mathscr{X}_n\right)}{k_n}}Z_i\boldsymbol{v_{g,i}}\mid\bar{\boldsymbol{\eta}}_n,\mathscr{X}_n\right]\lesssim\frac{\sqrt{\log\mathcal{C}_{n,\varepsilon}}}{n}\sup_{\boldsymbol{v_g}\in\boldsymbol{\mathcal{G}}_{n,\varepsilon}}\sqrt{\sum_{i=1}^{n}\left(1+\frac{d_i\left(\mathscr{X}_n\right)}{k_n}\right)\left|\boldsymbol{v_{g,i}}\right|^2}
$$
$$
\lesssim_d\frac{\sqrt{\log\mathcal{C}_{n,\varepsilon}}}{n}\sup_{\boldsymbol{v_g}\in\boldsymbol{\mathcal{G}}_{n,\varepsilon}}\sqrt{\sum_{i=1}^{n}\left|\boldsymbol{v_{g,i}}\right|^2}
$$
$$
\lesssim\mathcal{B}\sqrt{\frac{\log\mathcal{C}_{n,\varepsilon}}{n}} \tag{E.11}
$$

where $\bar{\boldsymbol{\eta}}=(\boldsymbol{\eta}_1,\ldots,\boldsymbol{\eta}_n)$ and the final bound follows by recalling that $\|\boldsymbol{g}\|_\infty\leq\mathcal{B}$ for all $\boldsymbol{g}\in\boldsymbol{\mathcal{G}}$. Now take $p_{\dim}\left(\boldsymbol{\mathcal{G}}\right)$ to be the pseudo-dimension of the class $\boldsymbol{\mathcal{G}}$. Then by Theorem 12.2 from Anthony & Bartlett (1999) we know that for large enough $n$,

$$
\log\mathcal{C}_{n,\varepsilon}\leq p_{\dim}\left(\boldsymbol{\mathcal{G}}\right)\log\left(\frac{2e\mathcal{B}n}{\varepsilon p_{\dim}\left(\boldsymbol{\mathcal{G}}\right)}\right)\leq p_{\dim}\left(\boldsymbol{\mathcal{G}}\right)\log\left(\frac{2e\mathcal{B}n}{\varepsilon}\right)
$$

Now substituting bounds on $p_{\dim}(\mathcal{G})$ from Bartlett et al. (2019) we get,

$$\log \mathcal{C}_{n,\varepsilon} \lesssim \mathcal{HS} \log \mathcal{S} \log \frac{2e\mathcal{B}n}{\varepsilon} \tag{E.12}$$

Choosing $\varepsilon = 1/\sqrt{n}$ and combining (E.7), (E.10), (E.11) and (E.12) we get,

$$T_{1,1} \lesssim_{\mathsf{K},d} \frac{1}{\sqrt{n}} + \sqrt{\frac{\mathcal{B}^2 \mathcal{HS} \log \mathcal{S} \log\left(2e\mathcal{B}n^{3/2}\right)}{n}} + \sqrt{\frac{\log\left(1/\delta\right)}{n}} \tag{E.13}$$

with probability at least $1 - \delta$. Now to further simplify the upper bound note that, by definition $\mathcal{H} \geq 1$ and hence,

$$\frac{\mathcal{B}^2 \mathcal{HS} \log \mathcal{S} \log\left(2e\mathcal{B}n^2\right)}{n} \lesssim \frac{\mathcal{B}^2 \mathcal{HS} \log \mathcal{S} \log n}{n} + \frac{\mathcal{B}^2 \mathcal{HS} \log \mathcal{S} \log \mathcal{B}}{n}.$$

By definition note that $w_0 = d + m \geq 2$ and $w_i \geq 1$ for all $1 \leq i \leq \mathcal{H}$. Then $\mathcal{S} \geq 4$ and hence recalling Assumption E.3 we get $\mathcal{B}^2 = o\left(n/\log n\right)$, implying $\log \mathcal{B} = O\left(\log n\right)$. Hence we can simplify the upper bound as,

$$\frac{\mathcal{B}^2 \mathcal{HS} \log \mathcal{S} \log\left(2e\mathcal{B}n^2\right)}{n} \lesssim \frac{\mathcal{B}^2 \mathcal{HS} \log \mathcal{S} \log n}{n}. \tag{E.14}$$

Now substituting in (E.13) we conclude,

$$T_{1,1} \lesssim_{\mathsf{K},d} \frac{1}{\sqrt{n}} + \sqrt{\frac{\mathcal{B}^2 \mathcal{HS} \log \mathcal{S} \log n}{n}} + \sqrt{\frac{\log\left(1/\delta\right)}{n}}$$

with probability at least $1 - \delta$.

### E.1.3. PROOF OF LEMMA E.8

Recall the function $h_{\boldsymbol{g}}$ from (E.2). Then note that,

$$T_{1,2} = \sup_{\boldsymbol{g} \in \mathcal{G}} \left| \frac{1}{nk_n} \sum_{i=1}^{n} \sum_{j \in N_{G(\mathscr{X}_n)}(i)} \langle h_{\boldsymbol{g}}\left(\boldsymbol{X}_i\right), h_{\boldsymbol{g}}\left(\boldsymbol{X}_i\right) - h_{\boldsymbol{g}}\left(\boldsymbol{X}_j\right) \rangle_{\mathcal{K}} \right|.$$

Now by Assumption E.2 we get,

$$
\begin{aligned}
\mathbb{E}\left[T_{1,2}\right] &\lesssim \mathbb{E}\left[\frac{1}{nk_n} \sum_{i=1}^{n} \sum_{j \in N_{G(\mathscr{X}_n)}(i)} \left(1 + \|\boldsymbol{X}_i\|_2^{\beta_1} + \|\boldsymbol{X}_j\|_2^{\beta_1}\right) \|\boldsymbol{X}_i - \boldsymbol{X}_j\|_2^{\beta_2}\right] \\
&= \mathbb{E}\left[\frac{1}{k_n} \sum_{j \in N_{G(\mathscr{X}_n)}(1)} \left(1 + \|\boldsymbol{X}_1\|_2^{\beta_1} + \|\boldsymbol{X}_j\|_2^{\beta_1}\right) \|\boldsymbol{X}_1 - \boldsymbol{X}_j\|_2^{\beta_2}\right] \\
&= \mathbb{E}\left[\left(1 + \|\boldsymbol{X}_1\|_2^{\beta_1} + \|\boldsymbol{X}_{\mathsf{N}(1)}\|_2^{\beta_1}\right) \|\boldsymbol{X}_1 - \boldsymbol{X}_{\mathsf{N}(1)}\|_2^{\beta_2}\right],
\end{aligned}
\tag{E.15}
$$

where the first equality follows by exchangeability and the second follows by choosing $\mathsf{N}(1)$ to be an uniformly selected index from $N_{G(\mathscr{X}_n)}(1)$, the neighbors of vertex $\boldsymbol{X}_1$. Now take $M_n = C\left(\log n\right)^{1/\alpha}$, where $C > 0$ is a universal constant, and let $E_n = \left\{\max\left\{\|\boldsymbol{X}_1\|_2, \|\boldsymbol{X}_{\mathsf{N}(1)}\|_2\right\} \leq M_n\right\}$. Now,

$$
\begin{aligned}
\mathbb{E}\bigg[\bigg(1 &+ \|\boldsymbol{X}_1\|_2^{\beta_1} + \|\boldsymbol{X}_{\mathsf{N}(1)}\|_2^{\beta_1}\bigg) \|\boldsymbol{X}_1 - \boldsymbol{X}_{\mathsf{N}(1)}\|_2^{\beta_2}\bigg] \\
&\lesssim \mathbb{E}\left[\left(1 + \|\boldsymbol{X}_1\|_2^{\beta_1} + \|\boldsymbol{X}_{\mathsf{N}(1)}\|_2^{\beta_1}\right) \|\boldsymbol{X}_1 - \boldsymbol{X}_{\mathsf{N}(1)}\|_2^{\beta_2} \mathbb{1}\left\{E_n^c\right\}\right] \\
&\quad + \mathbb{E}\left[\left(1 + \|\boldsymbol{X}_1\|_2^{\beta_1} + \|\boldsymbol{X}_{\mathsf{N}(1)}\|_2^{\beta_1}\right) \|\boldsymbol{X}_1 - \boldsymbol{X}_{\mathsf{N}(1)}\|_2^{\beta_2} \mathbb{1}\left\{E_n\right\}\right]
\end{aligned}
\tag{E.16}
$$

Next, for the first term, by Cauchy-Schwartz inequality we find,

$$
\mathbb{E}\left[\left(1 + \|\boldsymbol{X}_1\|_2^{\beta_1} + \|\boldsymbol{X}_{\mathsf{N}(1)}\|_2^{\beta_1}\right)\|\boldsymbol{X}_1 - \boldsymbol{X}_{\mathsf{N}(1)}\|_2^{\beta_2}\mathbb{1}\left\{E_n^c\right\}\right]
$$

$$
\leq \sqrt{\mathbb{E}\left[\left(1 + \|\boldsymbol{X}_1\|_2^{\beta_1} + \|\boldsymbol{X}_{\mathsf{N}(1)}\|_2^{\beta_1}\right)^2\|\boldsymbol{X}_1 - \boldsymbol{X}_{\mathsf{N}(1)}\|_2^{2\beta_2}\right]}\sqrt{\mathbb{P}\left(E_n^c\right)}
$$

By the tail condition from (E.1), Lemma D.2 from Deb et al. (2020) and choosing $C$ large enough we can conclude that the first term on RHS is bounded and $\mathbb{P}\left(E_n^c\right) \lesssim \exp\left(-4\log n\right) = n^{-4}$. Hence,

$$
\mathbb{E}\left[\left(1 + \|\boldsymbol{X}_1\|_2^{\beta_1} + \|\boldsymbol{X}_{N(1)}\|_2^{\beta_1}\right)\|\boldsymbol{X}_1 - \boldsymbol{X}_{N(1)}\|_2^{\beta_2}\mathbb{1}\left\{E_n^c\right\}\right] \lesssim \frac{1}{n^2}.
$$

Substituting in the bounds from (E.16) and once again using Cauchy-Schwartz inequality we get,

$$
\mathbb{E}\left[\left(1 + \|\boldsymbol{X}_1\|_2^{\beta_1} + \|\boldsymbol{X}_{\mathsf{N}(1)}\|_2^{\beta_1}\right)\|\boldsymbol{X}_1 - \boldsymbol{X}_{\mathsf{N}(1)}\|_2^{\beta_2}\right]
$$

$$
\lesssim \frac{1}{n^2} + \sqrt{\mathbb{E}\left[\left(1 + \|\boldsymbol{X}_1\|_2^{\beta_1} + \|\boldsymbol{X}_{\mathsf{N}(1)}\|_2^{\beta_1}\right)^2\right]}\sqrt{\mathbb{E}\left[\|\boldsymbol{X}_1 - \boldsymbol{X}_{\mathsf{N}(1)}\|_2^{2\beta_2}\mathbb{1}\left\{E_n\right\}\right]}
$$

$$
\lesssim \frac{1}{n^2} + \sqrt{\mathbb{E}\left[\|\boldsymbol{X}_1 - \boldsymbol{X}_{\mathsf{N}(1)}\|_2^{2\beta_2}\mathbb{1}\left\{E_n\right\}\right]} \tag{E.17}
$$

where the final bound follows by the tail condition from (E.1) and Lemma D.2 from Deb et al. (2020). To proceed with the second term define $\mathcal{N} = \mathcal{N}\left(M_n, \varepsilon\right)$ be the covering number of the ball $\mathcal{B}\left(M_n\right) = \{\boldsymbol{x} \in \mathbb{R}^d : \|\boldsymbol{x}\|_2 \leq M_n\}$ with respect to the $\|\cdot\|_2$ norm, where $\varepsilon > 0$ is the diameter of the covering balls. We now begin by expressing the expectation as a tail integral,

$$
\mathbb{E}\left[\left\|\boldsymbol{X}_1 - \boldsymbol{X}_{\mathsf{N}(1)}\right\|_2^{2\beta_2}\mathbb{1}\left\{\max\{\|\boldsymbol{X}_1\|_2, \left\|\boldsymbol{X}_{\mathsf{N}(1)}\right\|_2\}\right\} \leq M_n\right]
$$

$$
\lesssim 2\beta_2 \int_0^{2M_n} \varepsilon^{2\beta_2-1}\mathbb{P}\left(\left\|\boldsymbol{X}_1 - \boldsymbol{X}_{\mathsf{N}(1)}\right\|_2 \geq \varepsilon, \max\{\|\boldsymbol{X}_1\|_2, \left\|\boldsymbol{X}_{\mathsf{N}(1)}\right\|_2\} \leq M_n\right) d\varepsilon
$$

$$
\lesssim \varepsilon_n^{2\beta_2} + \int_{\varepsilon_n}^{2M_n} \varepsilon^{2\beta_2-1}\mathbb{P}\left(\left\|\boldsymbol{X}_1 - \boldsymbol{X}_{\mathsf{N}(1)}\right\|_2 \geq \varepsilon, \max\{\|\boldsymbol{X}_1\|_2, \left\|\boldsymbol{X}_{\mathsf{N}(1)}\right\|_2\} \leq M_n\right) d\varepsilon \tag{E.18}
$$

where the bound follows by noticing that $\varepsilon_n \leq M_n$ for large enough $C$. In the following we will bound the second term. Suppose $\mathcal{B}_1, \ldots, \mathcal{B}_{\mathcal{N}}$ are the covering balls of $\mathcal{B}\left(M_n\right)$ with respect to the $\|\cdot\|_2$ norm. Now define,

$$
\mathcal{S} := \{i : P_{\boldsymbol{X}}\left(\mathcal{B}_i\right) \leq Ck_n \log n/n\}, \tag{E.19}
$$

to be the collection of covering balls with probability under $P_{\boldsymbol{X}}$ smaller than $Ck_n\log n/n$. Then for $\varepsilon \in \left(\varepsilon_n, M_n\right)$ we have the following decomposition,

$$
\mathbb{P}\left(\left\|\boldsymbol{X}_1 - \boldsymbol{X}_{\mathsf{N}(1)}\right\|_2 \geq \varepsilon, \max\{\|\boldsymbol{X}_1\|_2, \left\|\boldsymbol{X}_{\mathsf{N}(1)}\right\|_2\} \leq M_n\right)
$$

$$
\lesssim \mathbb{P}\left(\left\|\boldsymbol{X}_1 - \boldsymbol{X}_{\mathsf{N}(1)}\right\|_2 \geq \varepsilon, \max\{\|\boldsymbol{X}_1\|_2, \left\|\boldsymbol{X}_{\mathsf{N}(1)}\right\|_2\} \leq M_n, \boldsymbol{X}_1, \boldsymbol{X}_{\mathsf{N}(1)} \in \bigcup_{i \notin \mathcal{S}}\mathcal{B}_i\right) + \mathbb{P}\left(\boldsymbol{X}_1 \in \bigcup_{i \in \mathcal{S}}\mathcal{B}_i\right)
$$

$$
\lesssim \mathbb{P}\left(\left\|\boldsymbol{X}_1 - \boldsymbol{X}_{\mathsf{N}(1)}\right\|_2 \geq \varepsilon, \max\{\|\boldsymbol{X}_1\|_2, \left\|\boldsymbol{X}_{\mathsf{N}(1)}\right\|_2\} \leq M_n, \boldsymbol{X}_1, \boldsymbol{X}_{\mathsf{N}(1)} \in \bigcup_{i \notin \mathcal{S}}\mathcal{B}_i\right) + \frac{k_n \log n}{n}\mathcal{N}, \tag{E.20}
$$

where the first inequality follows from Lemma D.2 in Deb et al. (2020) and the second inequality is a simple application of the union bound. To bound the first term note that $\left\|\boldsymbol{X}_1 - \boldsymbol{X}_{\mathsf{N}(1)}\right\|_2 \geq \varepsilon$ implies that for all $j$ such that $\boldsymbol{X}_j$ is not a $k_n$ nearest neighbor of $\boldsymbol{X}_i$, $\|\boldsymbol{X}_i - \boldsymbol{X}_j\|_2 \geq \varepsilon$.

Hence, it follows that

$$\mathbb{P}\left(\left\|\boldsymbol{X}_1 - \boldsymbol{X}_{\mathsf{N}(1)}\right\|_2 \geq \varepsilon, \max\{\left\|\boldsymbol{X}_1\right\|_2, \left\|\boldsymbol{X}_{\mathsf{N}(1)}\right\|_2\} \leq M_n, \boldsymbol{X}_1, \boldsymbol{X}_{\mathsf{N}(1)} \in \bigcup_{i \notin \mathcal{S}} \mathcal{B}_i\right)$$

$$\leq \mathbb{P}\left(\exists \ell, j_1, \ldots, j_{n-k_n-1} \text{ all distinct such that } \boldsymbol{X}_\ell \in \bigcup_{i \notin \mathcal{S}} \mathcal{B}_i, \min_{1 \leq v \leq n-k_n-1} \left\|\boldsymbol{X}_\ell - \boldsymbol{X}_{j_v}\right\|_2 \geq \varepsilon\right)$$

$$\leq \sum_{\substack{\ell, j_1, \ldots, j_{n-k_n-1} \\ \text{all distinct}}} \mathbb{P}\left(\boldsymbol{X}_\ell \in \bigcup_{i \notin \mathcal{S}} \mathcal{B}_i, \min_{1 \leq v \leq n-k_n-1} \left\|\boldsymbol{X}_\ell - \boldsymbol{X}_{j_v}\right\|_2 \geq \varepsilon\right) \qquad \text{(E.21)}$$

To bound the above probability, suppose $\mathcal{B}(\boldsymbol{X}_\ell) \in \{\mathcal{B}_i : i \notin \mathcal{S}\}$ denotes the covering ball where $\boldsymbol{X}_\ell$ lies. Then for a distinct collection of indices $\ell, j_1, \ldots, j_{n-k_n-1}$,

$$\mathbb{P}\left(\boldsymbol{X}_\ell \in \bigcup_{i \notin \mathcal{S}} \mathcal{B}_i, \min_{1 \leq v \leq n-k_n-1} \left\|\boldsymbol{X}_\ell - \boldsymbol{X}_{j_v}\right\|_2 \geq \varepsilon\right) \leq \mathbb{P}\left(\boldsymbol{X}_{j_v} \notin \mathcal{B}(\boldsymbol{X}_\ell), 1 \leq v \leq n-k_n-1\right)$$

To further bound the above probability note that,

$$\mathbb{P}\left(\boldsymbol{X}_{j_v} \notin \mathcal{B}(\boldsymbol{X}_\ell), 1 \leq v \leq n-k_n-1 | \boldsymbol{X}_\ell\right) = \left(1 - \mathbb{P}\left(\boldsymbol{X} \in \mathcal{B}(\boldsymbol{X}_\ell)|\boldsymbol{X}_\ell\right)\right)^{n-k_n-1}$$

$$\leq \left(1 - \frac{Ck_n \log n}{n}\right)^{n-k_n-1},$$

where $\boldsymbol{X} \sim P_{\boldsymbol{X}}$ is generated independent of $\boldsymbol{X}_\ell$ and the final bound follows by recalling the definition of $\mathcal{B}(\boldsymbol{X}_\ell)$ and $\mathcal{S}$. Hence recalling the bound from (E.21) we have,

$$\mathbb{P}\left(\left\|\boldsymbol{X}_1 - \boldsymbol{X}_{\mathsf{N}(1)}\right\|_2 \geq \varepsilon, \max\{\left\|\boldsymbol{X}_1\right\|_2, \left\|\boldsymbol{X}_{\mathsf{N}(1)}\right\|_2\} \leq M_n, \boldsymbol{X}_1, \boldsymbol{X}_{\mathsf{N}(1)} \in \bigcup_{i \notin \mathcal{S}} \mathcal{B}_i\right)$$

$$\leq n^{k_n+1}\left(1 - \frac{Ck_n \log n}{n}\right)^{n-k_n-1}$$

Using the fact $k_n = o(n/\log n)$ and choosing $C$ large enough we get,

$$n^{k_n+1}\left(1 - \frac{Ck_n \log n}{n}\right)^{n-k_n-1} \lesssim \frac{1}{n^2}.$$

Hence plugging this back into (E.20) we have,

$$\mathbb{P}\left(\left\|\boldsymbol{X}_1 - \boldsymbol{X}_{\mathsf{N}(1)}\right\|_2 \geq \varepsilon, \max\{\left\|\boldsymbol{X}_1\right\|_2, \left\|\boldsymbol{X}_{\mathsf{N}(1)}\right\|_2\} \leq M_n\right) \lesssim \frac{1}{n^2} + \frac{k_n \log n}{n}\mathcal{N}.$$

Recalling the definition of $\mathcal{N}$ we know that,

$$\mathcal{N} \lesssim_d \frac{(\log n)^{d/\alpha}}{\varepsilon^d}.$$

Since $\varepsilon \in (\varepsilon_n, 2M_n)$, then by definition of $\varepsilon_n$ and $M_n$ notice that,

$$\frac{1}{n^2} + \frac{k_n \log n}{n}\mathcal{N} \lesssim_d \frac{k_n \log n}{n}\frac{(\log n)^{d/\alpha}}{\varepsilon^d}.$$

Plugging this bound back in (E.18) shows that,

$$\mathbb{E}\left[\left\|\boldsymbol{X}_1 - \boldsymbol{X}_{\mathsf{N}(1)}\right\|_2^{2\beta_2} \mathbb{1}\left\{\max\{\|\boldsymbol{X}_1\|_2, \left\|\boldsymbol{X}_{\mathsf{N}(1)}\right\|_2\}\} \le M_n\right]$$
$$\lesssim_d \varepsilon_n^{2\beta_2} + \frac{k_n (\log n)^{1+d/\alpha}}{n} \int_{\varepsilon_n}^{2M_n} \varepsilon^{2\beta_2 - d - 1} d\varepsilon.$$
$$\lesssim_d \varepsilon_n^{2\beta_2} + \nu_n$$

where the final bound follows by evaluating the integral. Now substituting the bound in (E.17) and recalling (E.15) we get,

$$\mathbb{E}\left[T_{1,2}\right] \lesssim_d \frac{1}{n^2} + \varepsilon_n^{\beta_2} + \sqrt{\nu_n}$$

The proof is now completed by recalling the bound on K from Assumption E.1, (E.9) and following the combinatorial arguments from Appendix B.2 in Chatterjee et al. (2024) (also see Appendix C.3 in Deb et al. (2020)) with an application of McDiarmid's bounded difference inequality on the statistic $T_{1,2}$.

### E.1.4. PROOF OF LEMMA E.9

By a standard symmetrisation argument,

$$\mathbb{E}\left[T_{1,3}\right] \lesssim \mathbb{E}\left[\sup_{\boldsymbol{g} \in \boldsymbol{\mathcal{G}}} \left|\frac{1}{n}\sum_{i=1}^{n} \sigma_i \left\|h_{\boldsymbol{g}}\left(\boldsymbol{X}_i\right)\right\|_{\mathcal{K}}^2\right|\right]$$

where $\sigma_1, \ldots, \sigma_n$ are generated independently from Rademacher$(1/2)$. Then expanding the function $h_{\boldsymbol{g}}$ we get,

$$\mathbb{E}\left[T_{1,3}\right] \lesssim \mathbb{E}\left[\frac{1}{n}\left|\sum_{i=1}^{n} \sigma_i \mathsf{K}\left(\boldsymbol{Y}_i, \boldsymbol{Y}_i'\right)\right| + \sup_{\boldsymbol{g} \in \boldsymbol{\mathcal{G}}} \frac{1}{n}\left|\sum_{i=1}^{n} \sigma_i \mathsf{K}\left(\boldsymbol{Y}_i, \boldsymbol{g}_i\right)\right| + \sup_{\boldsymbol{g} \in \boldsymbol{\mathcal{G}}} \frac{1}{n}\left|\sum_{i=1}^{n} \sigma_i \mathsf{K}\left(\boldsymbol{g}_i, \boldsymbol{g}_i'\right)\right|\right] \tag{E.22}$$

where, for all $i \in [n]$, $\boldsymbol{Y}_i, \boldsymbol{Y}_i'$ are generated independently from $P_{\boldsymbol{Y}|\boldsymbol{X}=\boldsymbol{X}_i}$, and $\boldsymbol{g}_i = \boldsymbol{g}\left(\boldsymbol{\eta}_i, \boldsymbol{X}_i\right), \boldsymbol{g}_i' = \boldsymbol{g}\left(\boldsymbol{\eta}_i, \boldsymbol{X}_i\right)$ where $\{\boldsymbol{\eta}_i : i \in [n]\}$ and $\{\boldsymbol{\eta}_i' : i \in [n]\}$ are generated independently from $\mathrm{N}_m\left(\boldsymbol{0}, \boldsymbol{I}_m\right)$. By Khintchine's inequality,

$$\mathbb{E}\left[\frac{1}{n}\left|\sum_{i=1}^{n} \sigma_i \mathsf{K}\left(\boldsymbol{Y}_i, \boldsymbol{Y}_i'\right)\right|\right] \lesssim \frac{1}{n}\sqrt{\mathbb{E}\left[\sum_{i=1}^{n} \mathsf{K}\left(\boldsymbol{Y}_i, \boldsymbol{Y}_i'\right)^2\right]} \lesssim_{\mathsf{K}} \frac{1}{\sqrt{n}},$$

where the final bound follows by recalling that the kernel K is bounded. Substituting this bound back into (E.22) we get,

$$\mathbb{E}\left[T_{1,3}\right] \lesssim_{\mathsf{K}} \frac{1}{\sqrt{n}} + \mathbb{E}\left[\sup_{\boldsymbol{g} \in \boldsymbol{\mathcal{G}}} \frac{1}{n}\left|\sum_{i=1}^{n} \sigma_i \mathsf{K}\left(\boldsymbol{Y}_i, \boldsymbol{g}_i\right)\right|\right] + \mathbb{E}\left[\sup_{\boldsymbol{g} \in \boldsymbol{\mathcal{G}}} \frac{1}{n}\left|\sum_{i=1}^{n} \sigma_i \mathsf{K}\left(\boldsymbol{g}_i, \boldsymbol{g}_i'\right)\right|\right] \tag{E.23}$$

To further bound the last two terms consider,

$$\boldsymbol{\mathcal{G}}_n := \{\vec{\boldsymbol{g}} := (\boldsymbol{g}_1, \ldots, \boldsymbol{g}_n) : \boldsymbol{g} \in \boldsymbol{\mathcal{G}}\}$$

and,

$$\boldsymbol{\mathcal{G}}_n' := \{\vec{\boldsymbol{g}}' := (\boldsymbol{g}_1, \ldots, \boldsymbol{g}_n, \boldsymbol{g}_1', \ldots, \boldsymbol{g}_n') : \boldsymbol{g} \in \boldsymbol{\mathcal{G}}\}.$$

Moreover consider $d_{q,\infty}(\cdot, \cdot)$ be the $\ell_\infty$ distance on $\mathbb{R}^q$ for any $q \ge 1$ (see (E.8)). Now fix $\varepsilon > 0$ and let $\mathcal{C}_{n,\varepsilon}$ and $\mathcal{C}_{n,\varepsilon}'$ be the covering numbers of $\boldsymbol{\mathcal{G}}_\varepsilon$ and $\boldsymbol{\mathcal{G}}_n'$ at scale $\varepsilon$ with respect to the empirical distances $d_{n,\infty}$ and $d_{2n,\infty}$ respectively. Let $\boldsymbol{\mathcal{G}}_{n,\varepsilon}$ and $\boldsymbol{\mathcal{G}}_{n,\varepsilon}'$ be covering sets of $\boldsymbol{\mathcal{G}}_n$ and $\boldsymbol{\mathcal{G}}_n'$ respectively.

Now using the Lipschitz property of $\mathsf{K}$ we can show,

$$\mathbb{E}\left[\sup_{\boldsymbol{g}\in\boldsymbol{\mathcal{G}}}\frac{1}{n}\left|\sum_{i=1}^{n}\sigma_i\mathsf{K}\left(\boldsymbol{Y}_i,\boldsymbol{g}_i\right)\right| \mid \mathscr{D}_n\right]\lesssim_{\mathsf{K}}\varepsilon+\mathbb{E}\left[\sup_{\bar{\boldsymbol{g}}\in\boldsymbol{\mathcal{G}}_{n,\varepsilon}}\frac{1}{n}\left|\sum_{i=1}^{n}\sigma_i\mathsf{K}\left(\boldsymbol{Y}_i,\boldsymbol{g}_i\right)\right| \mid \mathscr{D}_n\right]$$

$$\lesssim\varepsilon+\frac{\sqrt{\log\mathcal{C}_{n,\varepsilon}}}{n}\sup_{\bar{\boldsymbol{g}}\in\boldsymbol{\mathcal{G}}_n}\left(\sum_{i=1}^{n}\mathsf{K}^2\left(\boldsymbol{Y}_i,\boldsymbol{g}_i\right)\right)^{1/2}$$

where $\mathscr{D}_n=\{(\boldsymbol{Y}_i,\boldsymbol{\eta}_i,\boldsymbol{X}_i):i\in[n]\}$ and the last bound follows by Lemma B.4 from Zhou et al. (2023). Recalling that $\mathsf{K}$ is bounded from Assumption E.1 we conclude,

$$\mathbb{E}\left[\sup_{\boldsymbol{g}\in\boldsymbol{\mathcal{G}}}\frac{1}{n}\left|\sum_{i=1}^{n}\sigma_i\mathsf{K}\left(\boldsymbol{Y}_i,\boldsymbol{g}_i\right)\right| \mid \mathscr{D}_n\right]\lesssim_{\mathsf{K}}\varepsilon+\sqrt{\frac{\log\mathcal{C}_{n,\varepsilon}}{n}}$$

As in (E.12), taking $\varepsilon=1/n$, invoking Theorem 12.2 from Anthony & Bartlett (1999), substituting the bounds on pseudo-dimension from Bartlett et al. (2019) and using the tower property of conditional expectations we get,

$$\mathbb{E}\left[\sup_{\boldsymbol{g}\in\boldsymbol{\mathcal{G}}}\frac{1}{n}\left|\sum_{i=1}^{n}\sigma_i\mathsf{K}\left(\boldsymbol{Y}_i,\boldsymbol{g}_i\right)\right|\right]\lesssim_{\mathsf{K}}\frac{1}{n}+\sqrt{\frac{\mathcal{B}^2\mathcal{H}\mathcal{S}\log\mathcal{S}\log\left(2e\mathcal{B}n^2\right)}{n}}.$$

Similarly we can show,

$$\mathbb{E}\left[\sup_{\boldsymbol{g}\in\boldsymbol{\mathcal{G}}}\frac{1}{n}\left|\sum_{i=1}^{n}\sigma_i\mathsf{K}\left(\boldsymbol{g}_i,\boldsymbol{g}_i'\right)\right|\right]\lesssim_{\mathsf{K}}\frac{1}{n}+\sqrt{\frac{\mathcal{B}^2\mathcal{H}\mathcal{S}\log\mathcal{S}\log\left(8e\mathcal{B}n^2\right)}{n}}.$$

Substituting the above bounds in (E.23) we get,

$$\mathbb{E}\left[T_{1,3}\right]\lesssim_{\mathsf{K}}\frac{1}{\sqrt{n}}+\sqrt{\frac{\mathcal{B}^2\mathcal{H}\mathcal{S}\log\mathcal{S}\log\left(8e\mathcal{B}n^2\right)}{n}}$$

Recalling the boundedness of the kernel $\mathsf{K}$ and using McDiarmid's bounded difference inequality we get,

$$T_{1,3}\lesssim_{\mathsf{K}}\frac{1}{\sqrt{n}}+\sqrt{\frac{\mathcal{B}^2\mathcal{H}\mathcal{S}\log\mathcal{S}\log\left(8e\mathcal{B}n^2\right)}{n}}+\sqrt{\frac{\log\left(1/\delta\right)}{n}}$$

with probability atleast $1-\delta$. Recalling the bound from (E.14) we conclude,

$$T_{1,3}\lesssim_{\mathsf{K}}\frac{1}{\sqrt{n}}+\sqrt{\frac{\mathcal{B}^2\mathcal{H}\mathcal{S}\log\mathcal{S}\log n}{n}}+\sqrt{\frac{\log\left(1/\delta\right)}{n}}$$

with probability at least $1-\delta$.

### E.2. Proof of Corollary E.5

By definition one can immediately recognise that,

$$\mathbb{E}\left[\mathrm{MMD}^2\left[\mathcal{F},P_{\hat{\boldsymbol{g}}(\boldsymbol{\eta},\boldsymbol{X})|\boldsymbol{X}},P_{\bar{\boldsymbol{g}}(\boldsymbol{\eta},\boldsymbol{X})|\boldsymbol{X}}\right] \mid \hat{\boldsymbol{g}}\right]=\mathcal{L}\left(\hat{\boldsymbol{g}}\right)\text{ a.s.}$$

Now fix $\varepsilon>0$. Then we can choose $R_\varepsilon>0$ large enough such that,

$$1-\Phi\left(R\right)^m\left(1-C_1\exp\left(-C_2R^\alpha\right)\right)\leq\frac{\varepsilon}{4}.$$

Moreover recall that $\bar{\boldsymbol{g}}$ is continuous and hence uniformly continuous in $E=[-R_\varepsilon,R_\varepsilon]^{d+m}$. Thus we know $\omega_{\bar{\boldsymbol{g}}}^E(r)\to 0$ as $r\to 0$. Hence choosing $n$ large enough and recalling Assumption E.3 shows that,

$$\sqrt{d+m}\,\omega_{\bar{\boldsymbol{g}}}^E\left(2R_\varepsilon\left(\mathcal{H}\mathcal{W}\right)^{-\frac{1}{d+m}}\right)\leq\frac{\varepsilon}{4},$$

and once again recalling Assumption E.3,

$$\frac{1}{\sqrt{n}} + \sqrt{\frac{\mathcal{B}^2 \mathcal{H} \mathcal{S} \log \mathcal{S} \log n}{n}} + \varepsilon_n^{\beta_2} + \sqrt{\nu_n} \leq \frac{\varepsilon}{4}.$$

where $\varepsilon_n, \nu_n$ are defined in Theorem E.4. Now choosing $\delta = \exp\left(-n\varepsilon^2/16\right)$ and applying the bound from Theorem E.4 we get,

$$\mathcal{L}(\hat{g}) \lesssim_{d,m,p,\mathsf{K}} \varepsilon \text{ with probability at least } 1 - \exp\left(-n\varepsilon^2/16\right) \text{ for all } n \text{ large enough.}$$

The proof is now completed by an application of the Borel-Cantelli lemma.

# F. When does Assumption (E.2) holds?

As discussed in Remark 4.3, the assumption in (E.2) (and in Assumption 4.2.4) is perhaps the most crucial assumption for convergence of the empirical estimator. This assumption was also considered in the works of Huang et al. (2022a); Deb et al. (2020); Azadkia & Chatterjee (2021); Dasgupta & Kpotufe (2014) for establishing rates of convergence of nearest neighbor based estimates. In this section we discuss when such assumptions might hold. To that end consider the following conditions.

**Assumption F.1.** Consider the following regularity conditions:

- The conditional density of $Y$ given $X = x$, say $f(\cdot|x)$ exists, is positive everywhere in its support, differentiable with respect to $x$ (for every $y$) and for all $1 \leq i \leq d$, the function $|(\partial/\partial x_i) \log f(y|x)|$ is bounded above by a polynomial in $\|y\|_2$ and $\|x\|_2$.

- For any $\ell \geq 1$, $\mathbb{E}[\|Y\|_2^\ell | X = x]$ is bounded above by a polynomial in $\|x\|_2$.

- Suppose that for all $g \in \mathcal{G}$, the conditional density of $g(\eta, X)$ given $X = x$, say $f_g(\cdot|x)$ exists and define,

$$r_g(y, x) = \frac{f_g(y|x)}{f(y|x)}$$

to be the density ratio such that $\sup_{g \in \mathcal{G}} |r_g(y, x)| \lesssim (1 + \|y\|_2^\zeta + \|x\|_2^\zeta)$ for some $\zeta > 0$. Furthermore, assume that for any $x_1, x_2 \in \mathbb{R}^d$,

$$\sup_{g \in \mathcal{G}} |r_g(y, x_1) - r_g(y, x_2)| \lesssim (1 + \|y\|_2^\gamma + \|x_1\|_2^\gamma + \|x_2\|_2^\gamma) \|x_1 - x_2\|_2, \tag{F.1}$$

for some $\gamma > 0$.

In the following we now show that the locally lipschtiz property from (E.2) (and also Assumption 4.2.4) holds whenever Assumption F.1 is satisfied.

**Proposition F.2.** *Suppose the kernel* $\mathsf{K}$ *is bounded. Then under Assumption F.1, (E.2) is satisfied with some* $C_3, \beta_1 > 0$ *and* $\beta_2 = 1$.

The main message of Proposition F.2 is that the locally Lipschitz condition in (E.2) is satisfied when the conditional density $f(\cdot \mid x)$ is a smooth function of $\|x\|_2$, and when the density ratio induced by applying any function from the class $\mathcal{G}$ exhibits sufficiently regular behavior. Similar conditions on density ratios have also been considered in prior work on conditional sampling (Zhou et al., 2023).

## F.1. Proof of Proposition F.2

Fix $x_1, x_2 \in \mathcal{X}$. Also fix $g \in \mathcal{G}$ and for notational convenience let $h = h_g$ where $h_g$ is defined in (E.2). Let $k \in \mathcal{K}$ such that $\|k\|_{\mathcal{K}}$ is bounded, then,

$$\left| \left\langle k, h(x_1) - h(x_2) \right\rangle_{\mathcal{K}} \right| = |\mathbb{E}[k(Y)(1 - r_g(Y, x_1))|X_1 = x_1] - \mathbb{E}[k(Y)(1 - r_g(Y, x_2))|X_2 = x_2]|$$

$$\leq \int |k(y)(1 - r_g(y, x_1))(f(y|x_1) - f(y|x_2))| \, dy$$

$$+ \int |k(y)(r_g(y, x_1) - r_g(y, x_2))f(y|x_2)| \, dy$$

$$\lesssim \|k\|_{\mathcal{K}} \left( \int |1 - r_g(y, x_1)| |f(y|x_1) - f(y|x_2)| \, dy \right.$$

$$\left. + \int |r_g(y, x_1) - r_g(y, x_2)| f(y|x_2) dy \right),$$

where the last inequality follows by recalling the bounds on the kernel $\mathsf{K}$, and the noticing that $|k(\boldsymbol{y})| = |\langle k, \mathsf{K}(\boldsymbol{y}, \cdot)\rangle_{\mathcal{H}_\mathsf{K}}| \lesssim_\mathsf{K} \|k\|_\mathcal{K}$. By using the mean value theorem along with the bounds on $|(\partial/\partial x_i) \log f(\boldsymbol{y}|\boldsymbol{x})|$ for all $1 \leq i \leq d$, the moment bounds from Assumption F.1, the polynomial bounds on $r_{\boldsymbol{g}}$ and (F.1) we now get,

$$|\langle k, h(\boldsymbol{x}_1) - h(\boldsymbol{x}_2)\rangle_\mathcal{K}| \lesssim \|k\|_\mathcal{K} \left(1 + \|\boldsymbol{x}_1\|_2^{\beta_1} + \|\boldsymbol{x}_2\|_2^{\beta_1}\right) \|\boldsymbol{x}_1 - \boldsymbol{x}_2\|_2,$$

for some $\beta_1 > 0$. By Theorem 4.1 from (Park & Muandet, 2020), $h(\boldsymbol{x}) \in \mathcal{K}$ for all $\boldsymbol{x} \in \mathcal{X}$. Recalling the bound on $\mathsf{K}$ it is easy to notice that $\sup_\mathcal{X} \|h(\boldsymbol{x})\|_\mathcal{K} \lesssim 1$. Hence we now conclude,

$$|\langle h(\boldsymbol{x}), h(\boldsymbol{x}_1) - h(\boldsymbol{x}_2)\rangle_\mathcal{K}| \lesssim \left(1 + \|\boldsymbol{x}_1\|_2^{\beta_1} + \|\boldsymbol{x}_2\|_2^{\beta_1}\right) \|\boldsymbol{x}_1 - \boldsymbol{x}_2\|_2.$$

# G. Uniform Concentration under Nearest Neighbor Interactions

In this section we provide a general overview about uniform concentration of non-linear statistics under nearest neighbor based weak interactions. The results presented here are crucially used for the proof of convergence of the proposed empirical sampler.

We begin by setting up the notations. Take $n \geq 2, d, m \geq 1$, let $\mathscr{X}_n := \{\boldsymbol{x}_1, \boldsymbol{x}_2, \ldots, \boldsymbol{x}_n\}$ be a collection of $n$ points in $\mathbb{R}^d$ and define $G(\mathscr{X}_n)$ to be the directed $k_n$-nearest neighbor graph on $\mathscr{X}_n$ with respect to the $\|\cdot\|_2$ norm. Moreover, consider $\mathcal{G}$ to be a collection of functions $\boldsymbol{g} : \mathbb{R}^m \times \mathbb{R}^d \to \mathbb{R}$ and for a function $h : \mathbb{R}^2 \times \mathbb{R}^2 \to \mathbb{R}$ define the non-linear statistic,

$$T_n(\boldsymbol{g}) := \frac{1}{nk_n} \sum_{i=1}^n \sum_{j \in N_{G(\mathscr{X}_n)}(i)} h(\boldsymbol{W}_{i,\boldsymbol{g}}, \boldsymbol{W}_{j,\boldsymbol{g}}) \tag{G.1}$$

where for all $i \in [n]$, $\boldsymbol{W}_{i,\boldsymbol{g}} := (Y_i, \boldsymbol{g}(\boldsymbol{\eta}_i, \boldsymbol{x}_i))$ with independent and identically distributed random variables $\{(\boldsymbol{\eta}_i, Y_i) : 1 \leq i \leq n\} \in \mathbb{R}^m \times \mathbb{R}$ and the set

$$N_{G(\mathscr{X}_n)}(i) := \{j \in [n] : \boldsymbol{x}_i \to \boldsymbol{x}_j \text{ is a directed edge in } G(\mathscr{X}_n)\}$$

for all $1 \leq i \leq n$. In the following theorem we establish uniform concentration of $T_n(\boldsymbol{g})$ around it's expectation.

**Theorem G.1.** *Consider the non-linear statistic $T_n(\boldsymbol{g})$ defined in (G.1) for all $\boldsymbol{g} \in \mathcal{G}$. Moreover, assume that the function $h : \mathbb{R}^2 \times \mathbb{R}^2 \to \mathbb{R}$ is Lipschitz continuous with Lipschitz constant $L > 0$ and is symmetric, that is $h(\boldsymbol{w}, \boldsymbol{w}') = h(\boldsymbol{w}', \boldsymbol{w})$ for any $\boldsymbol{w}, \boldsymbol{w}' \in \mathbb{R}^2$. Then,*

$$\mathbb{E}\left[\sup_{\boldsymbol{g} \in \mathcal{G}} T_n(\boldsymbol{g}) - \mathbb{E}[T_n(\boldsymbol{g})]\right] \lesssim_L \frac{1}{n}\mathbb{E}\left[\sup_{\boldsymbol{g} \in \mathcal{G}} \sum_{i=1}^n \sqrt{1 + \frac{d_i}{k_n}} Z_i \boldsymbol{g}(\boldsymbol{\eta}_i, \boldsymbol{x}_i)\right] \tag{G.2}$$

*where for all $i \in [n]$, $d_i$ is the degree (in-degree + out-degree) of the vertex $\boldsymbol{x}_i$ in $G(\mathscr{X}_n)$ and $\{Z_i : i \in [n]\}$ are generated independently from $\mathrm{N}(0, 1)$.*

*Remark* G.2. The results in Theorem G.1 can easily be extended to the case where $\boldsymbol{g} \in \mathcal{G}$ maps to $\mathbb{R}^p$ for some $p > 1$. Indeed in such setting the result from (G.2) becomes,

$$\mathbb{E}\left[\sup_{\boldsymbol{g} \in \mathcal{G}} T_n(\boldsymbol{g}) - \mathbb{E}[T_n(\boldsymbol{g})]\right] \lesssim_L \frac{1}{n}\mathbb{E}\left[\sup_{\boldsymbol{g} \in \mathcal{G}} \sum_{i=1}^n \sqrt{1 + \frac{d_i}{k_n}} \boldsymbol{Z}_i^\top \boldsymbol{g}(\boldsymbol{\eta}_i, \boldsymbol{x}_i)\right]$$

where $\boldsymbol{Z}_i \in \mathbb{R}^p$ for all $i \in [n]$ are now generated independently from $\mathrm{N}(\boldsymbol{0}, \boldsymbol{I}_p)$. The proof is exactly similar with additional notations and hence is omitted.

While Theorem G.1 provides bounds on uniform concentration in expectation, an application of McDiarmid's bounded difference inequality (see Theorem 6.5 of Boucheron et al. (2013)) and Lemma 2.1 from Jaffe et al. (2020) extends these results to high-probability bounds on uniform concentration in absolute difference. We formalize the result in the following.

**Corollary G.3.** *Adopt notations and settings from Theorem G.1. Moreover, assume that the function $h$ is uniformly bounded. Then for any $\delta > 0$, with probability at least $1 - \delta$,*

$$\sup_{\boldsymbol{g} \in \mathcal{G}} |T_n(\boldsymbol{g}) - \mathbb{E}[T_n(\boldsymbol{g})]| \lesssim_{L,h} \frac{1}{n}\mathbb{E}\left[\sup_{\boldsymbol{g} \in \mathcal{G}} \sum_{i=1}^n \sqrt{1 + \frac{d_i}{k_n}} Z_i \boldsymbol{g}(\boldsymbol{\eta}_i, \boldsymbol{x}_i)\right] + \sqrt{\frac{\log(2/\delta)}{n}}$$

The result from Corollary G.3 can easily be extended to the case when $\boldsymbol{g} \in \mathcal{G}$ maps to $\mathbb{R}^p$ for some $p > 1$. Indeed following the discussion from Remark G.2 one can show,

$$\sup_{\boldsymbol{g} \in \mathcal{G}} |T_n(\boldsymbol{g}) - \mathbb{E}[T_n(\boldsymbol{g})]| \lesssim_{L,h} \frac{1}{n}\mathbb{E}\left[\sup_{\boldsymbol{g} \in \mathcal{G}} \sum_{i=1}^n \sqrt{1 + \frac{d_i}{k_n}} \boldsymbol{Z}_i^\top \boldsymbol{g}(\boldsymbol{\eta}_i, \boldsymbol{x}_i)\right] + \sqrt{\frac{\log(2/\delta)}{n}}$$

holds with probability at least $1 - \delta$.

## G.1. Proof of Theorem G.1.

To begin with we set up some additional notations. For simplicity we take $N(i) = N_{G(\mathscr{X}_n)}(i)$ for all $i \in [n]$. Define,

$$t(\bar{\boldsymbol{w}}_n) := \frac{1}{nk_n} \sum_{i=1}^{n} \sum_{j \in N_{G(\mathscr{X}_n)}(i)} h(\boldsymbol{w}_i, \boldsymbol{w}_j) \text{ for all } \bar{\boldsymbol{w}}_n := (\boldsymbol{w}_1, \ldots, \boldsymbol{w}_n) \in \mathbb{R}^{2n}.$$

Then note that $T_n(\boldsymbol{g}) = t(\bar{\boldsymbol{W}}_{n,\boldsymbol{g}})$ where $\bar{\boldsymbol{W}}_{n,\boldsymbol{g}} := (\boldsymbol{W}_{1,\boldsymbol{g}}, \ldots, \boldsymbol{W}_{n,\boldsymbol{g}})$. Now take $\bar{\boldsymbol{W}}'_{n,\boldsymbol{g}} := (\boldsymbol{W}'_{1,\boldsymbol{g}}, \ldots, \boldsymbol{W}'_{n,\boldsymbol{g}})$ to be an independent copy of $\bar{\boldsymbol{W}}_{n,\boldsymbol{g}}$ and note that,

$$\mathbb{E}\left[\sup_{\boldsymbol{g} \in \mathcal{G}} T_n(\boldsymbol{g}) - \mathbb{E}[T_n(\boldsymbol{g})]\right] \leq \mathbb{E}\left[\sup_{\boldsymbol{g} \in \mathcal{G}} t(\bar{\boldsymbol{W}}_{n,\boldsymbol{g}}) - t(\bar{\boldsymbol{W}}'_{n,\boldsymbol{g}})\right]. \tag{G.3}$$

To complete the proof it is now enough to bound the right hand side of (G.3). To this end we begin by defining a partial difference operator. Take $m \in [n]$ and for $\boldsymbol{v}, \boldsymbol{v}' \in \mathbb{R}^2$ define,

$$D_{\boldsymbol{v},\boldsymbol{v}'}^m t(\bar{\boldsymbol{w}}_n) := t(\boldsymbol{w}_1, \ldots, \boldsymbol{w}_{m-1}, \boldsymbol{v}, \boldsymbol{w}_{m+1}, \ldots, \boldsymbol{w}_n) - t(\boldsymbol{w}_1, \ldots, \boldsymbol{w}_{m-1}, \boldsymbol{v}', \boldsymbol{w}_{m+1}, \ldots, \boldsymbol{w}_n). \tag{G.4}$$

Moreover for any $i \in [n]$ let,

$$\bar{N}(i) := \{j \in [n] : \boldsymbol{x}_j \to \boldsymbol{x}_i \text{ is a directed edge in } G(\mathscr{X}_n)\}.$$

Next, we first show a Lipschitz type property for the partial difference operator $D$.

**Lemma G.4.** *Fix $m \in [n]$ and take $\bar{\boldsymbol{w}}_n := \{\boldsymbol{w}_1, \ldots, \boldsymbol{w}_n\} \in \mathbb{R}^{2n}, \bar{\boldsymbol{w}}'_n := \{\boldsymbol{w}'_1, \ldots, \boldsymbol{w}'_n\} \in \mathbb{R}^{2n}$. Then for any $\boldsymbol{v}, \boldsymbol{v}' \in \mathbb{R}^2$,*

$$\left|D_{\boldsymbol{v},\boldsymbol{v}'}^m t(\bar{\boldsymbol{w}}_n) - D_{\boldsymbol{v},\boldsymbol{v}'}^m t(\bar{\boldsymbol{w}}'_n)\right| \lesssim_L \frac{1}{nk_n} \sum_{j \in \mathcal{N}(m)} \left\|\boldsymbol{w}_j - \boldsymbol{w}'_j\right\|_2$$

*where $D$ is defined in (G.4) and $\mathcal{N}(m) := N(m) \bigcup \bar{N}(m)$ for all $m \in [n]$.*

Now we will use this partial difference operator to expand the difference $t(\bar{\boldsymbol{w}}_n) - t(\bar{\boldsymbol{w}}'_n)$. Towards that we first define a new collection combining $\bar{\boldsymbol{w}}_n$ and $\bar{\boldsymbol{w}}'_n$. For any $A \subseteq [n]$ define $\bar{\boldsymbol{w}}_n^A = (\boldsymbol{w}_1^A, \ldots, \boldsymbol{w}_n^A)$ as,

$$\boldsymbol{w}_i^A = \begin{cases} \boldsymbol{w}'_i & \text{if } i \in A \\ \boldsymbol{w}_i & \text{if } i \notin A. \end{cases}$$

Furthermore for $m \in [n]$ define,

$$F_m(\bar{\boldsymbol{w}}_n, \bar{\boldsymbol{w}}'_n) = \frac{1}{2^m} \sum_{A \subseteq [m-1]} \left(D_{\boldsymbol{w}_m, \boldsymbol{w}'_m}^m t(\bar{\boldsymbol{w}}_n^A) + D_{\boldsymbol{w}_m, \boldsymbol{w}'_m}^m t(\bar{\boldsymbol{w}}_n^{A^c})\right) \tag{G.5}$$

Then by Lemma 9 from (Maurer & Pontil, 2019) we know,

$$t(\bar{\boldsymbol{w}}_n) - t(\bar{\boldsymbol{w}}'_n) = \sum_{m=1}^{n} F_m(\bar{\boldsymbol{w}}_n, \bar{\boldsymbol{w}}'_n) \text{ for all } \bar{\boldsymbol{w}}_n, \bar{\boldsymbol{w}}'_n \in \mathbb{R}^{2n}. \tag{G.6}$$

Now for all $m \in [n]$ define an operator $\mathcal{M}_m$ as $\mathcal{M}_m \bar{\boldsymbol{w}}_n = (M_{m,1}\boldsymbol{w}_1, \ldots, M_{m,n}\boldsymbol{w}_n)$ where,

$$M_{m,i} = \begin{cases} 1/n & \text{if } i = m \\ 1/n\sqrt{k_n} & \text{if } i \in \mathcal{N}(m) \\ 0 & \text{otherwise} \end{cases} \tag{G.7}$$

and let $\mathcal{M}_m(\bar{\boldsymbol{w}}_n, \bar{\boldsymbol{w}}'_n) = (\mathcal{M}_m \bar{\boldsymbol{w}}_n, \mathcal{M}_m \bar{\boldsymbol{w}}'_n)$. These definition now lead to a Lipschitz type property for $F_m$. In particular we have the following lemma.

**Lemma G.5.** *For any $\bar{\boldsymbol{w}}_n, \bar{\boldsymbol{v}}_n, \bar{\boldsymbol{w}}'_n, \bar{\boldsymbol{v}}'_n \subseteq \mathbb{R}^{2n}$ and $m \in [n]$ we have,*

$$F_m\left(\bar{\boldsymbol{w}}_n, \bar{\boldsymbol{w}}'_n\right) - F_m\left(\bar{\boldsymbol{v}}_n, \bar{\boldsymbol{v}}'_n\right) \lesssim_{d,L} \mathbb{E}\left[\left|\boldsymbol{\mathcal{Z}}_m^{\top}\left(\mathcal{M}_m\left(\bar{\boldsymbol{w}}_n, \bar{\boldsymbol{w}}'_n\right) - \mathcal{M}_m\left(\bar{\boldsymbol{v}}_n, \bar{\boldsymbol{v}}'_n\right)\right)\right|\right]$$

*where $\boldsymbol{\mathcal{Z}}_m = \left(\mathcal{Z}_{m,1}, \ldots, \mathcal{Z}_{m,n}, \mathcal{Z}'_{m,1}, \ldots, \mathcal{Z}'_{m,n}\right)^{\top}$ with $\{\mathcal{Z}_{m,i} : 1 \leq i \leq n\}, \{\mathcal{Z}'_{m,i} : 1 \leq i \leq n\}$ generated independently from $\mathrm{N}_2\left(\boldsymbol{0}, \boldsymbol{I}_2\right)$.*

Using the decomposition from (G.6) and applying Lemma G.5 we can now replicate the proof of equation (12) in (Maurer & Pontil, 2019) to get,

$$\mathbb{E}\left[\sup_{\boldsymbol{g} \in \boldsymbol{\mathcal{G}}} t\left(\bar{\boldsymbol{W}}_{n,\boldsymbol{g}}\right) - t\left(\bar{\boldsymbol{W}}'_{n,\boldsymbol{g}}\right)\right] \lesssim_{d,L} \mathbb{E}\left[\sup_{\boldsymbol{g} \in \boldsymbol{\mathcal{G}}} \sum_{m=1}^{n} \boldsymbol{\mathcal{Z}}_m^{\top} \mathcal{M}_m\left(\bar{\boldsymbol{W}}_{n,\boldsymbol{g}}, \bar{\boldsymbol{W}}'_{n,\boldsymbol{g}}\right)\right]. \tag{G.8}$$

By definition of the operator $\mathcal{M}_m$ from (G.7) we get,

$$\sum_{m=1}^{n} \boldsymbol{\mathcal{Z}}_m^{\top} \mathcal{M}_m\left(\bar{\boldsymbol{W}}_{n,\boldsymbol{g}}, \bar{\boldsymbol{W}}'_{n,\boldsymbol{g}}\right) = \sum_{m=1}^{n} \sum_{i=1}^{n} M_{m,i} \mathcal{Z}_{m,i}^{\top} \boldsymbol{W}_{i,g} + M_{m,i} \mathcal{Z}'^{\top}_{m,i} \boldsymbol{W}'_{i,g}$$

$$= \sum_{i=1}^{n} \left[\left(\sum_{m=1}^{n} M_{m,i} \mathcal{Z}_{m,i}\right)^{\top} \boldsymbol{W}_{i,g} + \left(\sum_{m=1}^{n} M_{m,i} \mathcal{Z}'_{m,i}\right)^{\top} \boldsymbol{W}'_{i,g}\right]$$

$$\overset{d}{=} \frac{1}{n} \sum_{i=1}^{n} \sqrt{1 + \frac{d_i}{k_n}} \left[\mathcal{Z}_i^{\top} \boldsymbol{W}_{i,g} + \mathcal{Z}'^{\top}_i \boldsymbol{W}'_{i,g}\right] \tag{G.9}$$

where $\{\mathcal{Z}_i : 1 \leq i \leq n\}, \{\mathcal{Z}'_i, 1 \leq i \leq n\}$ are generated independently from $\mathrm{N}_2\left(\boldsymbol{0}, \boldsymbol{I}_2\right)$. The equality in distribution from (G.9) follows by recalling the definition of $\mathcal{N}$ from Lemma G.4, operator $\mathcal{M}$ from (G.7) and noting that for any $i \in [n]$,

$$\sum_{m=1}^{n} \mathcal{M}_{m,i}^2 = \frac{1}{n^2} + \frac{1}{n^2 k_n} \sum_{m=1}^{n} \mathbb{1}\left\{i \in \mathcal{N}\left(m\right)\right\}$$

$$= \frac{1}{n^2} + \frac{1}{n^2 k_n} \sum_{m=1}^{n} \mathbb{1}\left\{m \in \mathcal{N}\left(i\right)\right\} = \frac{1}{n^2}\left(1 + \frac{d_i}{k_n}\right)$$

where $d_i$ is the degree (in-degree + out-degree) of vertex $\boldsymbol{x}_i$ in $G\left(\mathscr{X}_n\right)$. Now substituting the expression from (G.9) in the bound from (G.8) we get,

$$\mathbb{E}\left[\sup_{\boldsymbol{g} \in \boldsymbol{\mathcal{G}}} t\left(\bar{\boldsymbol{W}}_{n,\boldsymbol{g}}\right) - t\left(\bar{\boldsymbol{W}}'_{n,\boldsymbol{g}}\right)\right] \lesssim_{d,L} \frac{1}{n} \mathbb{E}\left[\sup_{\boldsymbol{g} \in \boldsymbol{\mathcal{G}}} \sum_{i=1}^{n} \sqrt{1 + \frac{d_i}{k_n}} \left[\mathcal{Z}_i^{\top} \boldsymbol{W}_{i,g} + \mathcal{Z}'^{\top}_i \boldsymbol{W}'_{i,g}\right]\right]$$

$$\lesssim_{d,L} \frac{1}{n} \mathbb{E}\left[\sup_{\boldsymbol{g} \in \boldsymbol{\mathcal{G}}} \sum_{i=1}^{n} \sqrt{1 + \frac{d_i}{k_n}} \mathcal{Z}_i^{\top} \boldsymbol{W}_{i,g}\right]$$

$$\lesssim_{d,L} \frac{1}{n} \mathbb{E}\left[\sup_{\boldsymbol{g} \in \boldsymbol{\mathcal{G}}} \sum_{i=1}^{n} \sqrt{1 + \frac{d_i}{k_n}} Z_i \boldsymbol{g}\left(\boldsymbol{\eta}_i, \boldsymbol{x}_i\right)\right] \tag{G.10}$$

where $\{Z_i : i \in [n]\}$ are generated independently from the standard Gaussian distribution and the final inequality follows by recalling the definition of $\boldsymbol{W}_{i,g}, i \in [n]$ from (G.1). The proof is now completed by substituting the bound from (G.10) in (G.3).

### G.1.1. PROOF OF LEMMA G.4.

By definition note that,

$$D_{\boldsymbol{v},\boldsymbol{v}'}^m t(\bar{\boldsymbol{w}}_n) = \frac{1}{nk_n}\left[\sum_{j \in N(m)} h(\boldsymbol{v}, \boldsymbol{w}_j) - h(\boldsymbol{v}', \boldsymbol{w}_j) + \sum_{j \in \bar{N}(m)} h(\boldsymbol{w}_j, \boldsymbol{v}) - h(\boldsymbol{w}_j, \boldsymbol{v}')\right] \tag{G.11}$$

Then, using the Lipschitz property of $h$ we have,

$$\left| D_{\boldsymbol{v},\boldsymbol{v}'}^m t\left(\bar{\boldsymbol{w}}_n\right) - D_{\boldsymbol{v},\boldsymbol{v}'}^m t\left(\bar{\boldsymbol{w}}_n'\right)\right| = \left| \frac{1}{nk_n}\left[\sum_{j\in N(m)} h(\boldsymbol{v},\boldsymbol{w}_j) - h(\boldsymbol{v},\boldsymbol{w}_j') - h(\boldsymbol{v}',\boldsymbol{w}_j) + h(\boldsymbol{v}',\boldsymbol{w}_j')\right.\right.$$

$$\left.\left. + \sum_{j\in\bar{N}(m)} h(\boldsymbol{w}_j,\boldsymbol{v}) - h(\boldsymbol{w}_j',\boldsymbol{v}) - u(\boldsymbol{w}_j,\boldsymbol{v}') + h(\boldsymbol{w}_j,\boldsymbol{v}')\right]\right|$$

$$\lesssim_L \frac{1}{nk_n}\sum_{j\in\mathcal{N}(m)}\|\boldsymbol{w}_j - \boldsymbol{w}_j'\| \tag{G.12}$$

where recall $\mathcal{N}(m) = N(m)\bigcup\bar{N}(m)$ and $L$ is the Lipschitz constant of $h$.

### G.1.2. PROOF OF LEMMA G.5

Let the collections $\bar{\boldsymbol{w}}_n, \bar{\boldsymbol{v}}_n, \bar{\boldsymbol{w}}_n', \bar{\boldsymbol{v}}_n'$ be defined as $\bar{\boldsymbol{w}}_n := (\boldsymbol{w}_1,\ldots,\boldsymbol{w}_n)\,,\bar{\boldsymbol{v}}_n := (\boldsymbol{v}_1,\ldots,\boldsymbol{v}_n)\,,\bar{\boldsymbol{w}}_n' := (\boldsymbol{w}_1',\ldots,\boldsymbol{w}_n')$ and $\bar{\boldsymbol{v}}_n' := (\boldsymbol{v}_1',\ldots,\boldsymbol{v}_n')$. Now by Lemma 2.1 from (Jaffe et al., 2020) we know that

$$|\mathcal{N}(m)| \lesssim_d k_n \text{ for all } m\in[n]. \tag{G.13}$$

Then by recalling the definition of the partial difference operator from (G.4), the expansion from (G.11) and the bound from (G.12) we get,

$$D_{\boldsymbol{w}_m,\boldsymbol{w}_m'}^m t\left(\boldsymbol{w}^A\right) - D_{\boldsymbol{v}_m,\boldsymbol{v}_m'}^m t\left(\boldsymbol{v}^A\right)$$

$$= D_{\boldsymbol{w}_m,\boldsymbol{v}_m}^m t\left(\boldsymbol{w}^A\right) + D_{\boldsymbol{w}_m',\boldsymbol{v}_m'}^m t\left(\boldsymbol{w}^A\right) + D_{\boldsymbol{v}_m,\boldsymbol{v}_m'}^m\left(t\left(\boldsymbol{w}^A\right) - t\left(\boldsymbol{v}^A\right)\right)$$

$$\lesssim_{d,L} \frac{1}{n}\|\boldsymbol{w}_m - \boldsymbol{v}_m\| + \frac{1}{n}\|\boldsymbol{w}_m' - \boldsymbol{v}_m'\| + \frac{1}{nk_n}\sum_{j\in\mathcal{N}(m)}\|\boldsymbol{w}_j^A - \boldsymbol{v}_j^A\| \tag{G.14}$$

where the final bound follows using the Lipschitz property of $h$ and Lemma G.4. Now recalling the definition of $F_m$ from (G.5) we get,

$$F_m\left(\bar{\boldsymbol{w}},\bar{\boldsymbol{w}}'\right) - F_m\left(\bar{\boldsymbol{v}},\bar{\boldsymbol{v}}'\right)$$

$$= \frac{1}{2^m}\sum_{A\subseteq[m-1]}\left(D_{\boldsymbol{w}_m,\boldsymbol{w}_m'}^m f(\boldsymbol{w}^A) - D_{\boldsymbol{v}_m,\boldsymbol{v}_m'}^m f(\boldsymbol{v}^A) + D_{\boldsymbol{w}_m,\boldsymbol{w}_m'}^m f(\boldsymbol{w}^{A^c}) - D_{\boldsymbol{v}_m,\boldsymbol{v}_m'}^m f(\boldsymbol{v}^{A^c})\right)$$

$$\lesssim_{d,L} \frac{1}{n}\left(\|\boldsymbol{w}_m - \boldsymbol{v}_m\| + \|\boldsymbol{w}_m' - \boldsymbol{v}_m'\|\right) + \frac{1}{nk_n}\sum_{j\in\mathcal{N}(m)}\|\boldsymbol{w}_j - \boldsymbol{v}_j\| + \|\boldsymbol{w}_j' - \boldsymbol{v}_j'\| \tag{G.15}$$

$$\lesssim_{d,L} \frac{1}{n}\left(\|\boldsymbol{w}_m - \boldsymbol{v}_m\|^2 + \|\boldsymbol{w}_m' - \boldsymbol{v}_m'\|^2\right)^{1/2} + \frac{1}{n\sqrt{k_n}}\left(\sum_{j\in\mathcal{N}(m)}\|\boldsymbol{w}_j - \boldsymbol{v}_j\|^2 + \|\boldsymbol{w}_j' - \boldsymbol{v}_j'\|^2\right)^{1/2} \tag{G.16}$$

$$\lesssim_{d,L} \frac{1}{n}\left(\|\boldsymbol{w}_m - \boldsymbol{v}_m\|^2 + \|\boldsymbol{w}_m' - \boldsymbol{v}_m'\|^2 + \frac{1}{k_n}\sum_{j\in\mathcal{N}(m)}\|\boldsymbol{w}_j - \boldsymbol{v}_j\|^2 + \|\boldsymbol{w}_j' - \boldsymbol{v}_j'\|^2\right)^{1/2}$$

$$= \|\mathcal{M}_m\left(\boldsymbol{w},\boldsymbol{w}'\right) - \mathcal{M}_m\left(\boldsymbol{v},\boldsymbol{v}'\right)\| \tag{G.17}$$

$$\lesssim_{d,L} \mathbb{E}\left[\left|\boldsymbol{\mathcal{Z}}_m^\top\left(\mathcal{M}_m\left(\boldsymbol{w},\boldsymbol{w}'\right) - \mathcal{M}_m\left(\boldsymbol{v},\boldsymbol{v}'\right)\right)\right|\right] \tag{G.18}$$

where the bound in (G.15) follows from (G.14), (G.16) follows using Cauchy-Schwartz inequality, (G.17) follows by recalling the definition of operator $\mathcal{M}$ from (G.7) and finally (G.18) follows by noting that $\mathbb{E}\left[\left|\mathcal{Z}^\top\boldsymbol{v}\right|\right] = \|\boldsymbol{v}\|$ whenever $\mathcal{Z}\sim N\left(\boldsymbol{0},\boldsymbol{I}\right)$ (see Lemma 7 in (Maurer & Pontil, 2019)).

## G.2. Proof of Corollary G.3

Note that,

$$\sup_{\boldsymbol{g}\in\boldsymbol{\mathcal{G}}}\left|T_n\left(\boldsymbol{g}\right)-\mathbb{E}\left[T_n\left(\boldsymbol{g}\right)\right]\right|\leq\max\left\{\sup_{\boldsymbol{g}\in\boldsymbol{\mathcal{G}}}T_n\left(\boldsymbol{g}\right)-\mathbb{E}\left[T_n\left(\boldsymbol{g}\right)\right],\sup_{\boldsymbol{g}\in\boldsymbol{\mathcal{G}}}\mathbb{E}\left[T_n\left(\boldsymbol{g}\right)\right]-T_n\left(\boldsymbol{g}\right)\right\}. \tag{G.19}$$

Replacing $h$ by $-h$ in (G.1) and applying Theorem G.1 gives,

$$\mathbb{E}\left[\sup_{\boldsymbol{g}\in\boldsymbol{\mathcal{G}}}\mathbb{E}\left[T_n\left(\boldsymbol{g}\right)\right]-T_n\left(\boldsymbol{g}\right)\right]\lesssim_L\frac{1}{n}\mathbb{E}\left[\sup_{\boldsymbol{g}\in\boldsymbol{\mathcal{G}}}\sum_{i=1}^n\sqrt{1+\frac{d_i}{k_n}}Z_i\boldsymbol{g}\left(\boldsymbol{\eta}_i,\boldsymbol{x}_i\right)\right]. \tag{G.20}$$

Now recall that $h$ is uniformly bounded. Hence, applying McDiarmid's bounded difference inequality on both $\sup_{\boldsymbol{g}\in\boldsymbol{\mathcal{G}}}T_n\left(\boldsymbol{g}\right)-\mathbb{E}\left[T_n\left(\boldsymbol{g}\right)\right]$ and $\mathbb{E}\left[\sup_{\boldsymbol{g}\in\boldsymbol{\mathcal{G}}}\mathbb{E}\left[T_n\left(\boldsymbol{g}\right)\right]-T_n\left(\boldsymbol{g}\right)\right]$ with Theorem G.1 and (G.20) shows,

$$\sup_{\boldsymbol{g}\in\boldsymbol{\mathcal{G}}}T_n\left(\boldsymbol{g}\right)-\mathbb{E}\left[T_n\left(\boldsymbol{g}\right)\right]\lesssim_{L,h}\frac{1}{n}\mathbb{E}\left[\sup_{\boldsymbol{g}\in\boldsymbol{\mathcal{G}}}\sum_{i=1}^n\sqrt{1+\frac{d_i}{k_n}}Z_i\boldsymbol{g}\left(\boldsymbol{\eta}_i,\boldsymbol{x}_i\right)\right]+\sqrt{\frac{\log\left(2/\delta\right)}{n}} \tag{G.21}$$

with probability at least $1-\delta/2$ and,

$$\sup_{\boldsymbol{g}\in\boldsymbol{\mathcal{G}}}\mathbb{E}\left[T_n\left(\boldsymbol{g}\right)\right]-T_n\left(\boldsymbol{g}\right)\lesssim_{L,h}\frac{1}{n}\mathbb{E}\left[\sup_{\boldsymbol{g}\in\boldsymbol{\mathcal{G}}}\sum_{i=1}^n\sqrt{1+\frac{d_i}{k_n}}Z_i\boldsymbol{g}\left(\boldsymbol{\eta}_i,\boldsymbol{x}_i\right)\right]+\sqrt{\frac{\log\left(2/\delta\right)}{n}} \tag{G.22}$$

with probability at least $1-\delta/2$. The proof is now completed by combining (G.21), (G.22) and (G.19).

## H. Technical Results

**Lemma H.1.** *Take $m \geq 1$ and let $A \subseteq \mathbb{R}^m$. Let $M = \sup_{\boldsymbol{a} \in A} \sqrt{\sum_{i=1}^m a_i^2}$ where $\boldsymbol{a} = (a_1, \ldots, a_m)$. Then,*

$$\mathbb{E} \left[ \sup_{\boldsymbol{a} \in A} \frac{1}{m} \sum_{i=1}^m a_i Z_i \right] \leq \frac{R \sqrt{2 \log |A|}}{m}$$

*where $Z_1, \ldots, Z_m$ are generated independently from $\mathrm{N}(0, 1)$.*

*Proof.* Take $s \geq 0$. Then by Jensen's inequality we get,

$$\exp \left( s \mathbb{E} \left[ \sup_{\boldsymbol{a} \in A} \sum_{i=1}^m a_i Z_i \right] \right) \leq \mathbb{E} \left[ \exp \left( s \sup_{\boldsymbol{a} \in A} \sum_{i=1}^n a_i Z_i \right) \right] \leq \sum_{\boldsymbol{a} \in A} \mathbb{E} \left[ \exp \left( s \sum_{i=1}^n a_i Z_i \right) \right]$$

Using the independence of $Z_1, \ldots, Z_n$ we get,

$$\exp \left( s \mathbb{E} \left[ \sup_{\boldsymbol{a} \in A} \sum_{i=1}^m a_i Z_i \right] \right) \leq \sum_{\boldsymbol{a} \in A} \prod_{i=1}^m \mathbb{E} \left[ \exp \left( s a_i Z_i \right) \right] = \sum_{\boldsymbol{a} \in A} \prod_{i=1}^m \exp \left( \frac{s^2 a_i^2}{2} \right)$$

$$\leq |A| \exp \left( \frac{s^2 R^2}{2} \right).$$

Taking logarithm of both sides we get,

$$\mathbb{E} \left[ \sup_{\boldsymbol{a} \in A} \sum_{i=1}^m a_i Z_i \right] \leq \frac{\log |A|}{s} + \frac{s R^2}{2}.$$

Recall that our choice of $s$ was arbitrary, hence minimizing the right hand side with respect to $s$ we find,

$$\mathbb{E} \left[ \sup_{\boldsymbol{a} \in A} \sum_{i=1}^m a_i Z_i \right] \leq \frac{R \log |A|}{\sqrt{2 \log |A|}} + \frac{R^2 \sqrt{2 \log |A|}}{2R} = R \sqrt{2 \log |A|}.$$

The proof is now completed by dividing both sides by $m$. $\qquad \square$

The following classical result due to Bochner characterizes continuous positive definite functions. The version stated below is adapted from Wendland (2005, Theorem 6.6) (also see Sriperumbudur et al. (2010, Theorem 3)).

**Theorem H.2** (Bochner). *A continuous function $\psi : \mathbb{R}^p \to \mathbb{R}$ is positive definite if and only if it is the Fourier transform of a finite non-negative Borel measure $\Lambda$ on $\mathbb{R}^p$ that is,*

$$\psi(\boldsymbol{x}) = \int_{\mathbb{R}^p} e^{-\iota \boldsymbol{x}^\top \boldsymbol{\omega}} \mathrm{d}\Lambda(\boldsymbol{\omega}) \text{ for all } \boldsymbol{x} \in \mathbb{R}^p.$$

