# OpenReview forum: "One-shot Conditional Sampling: MMD meets Nearest Neighbors"
_ICML.cc/2026/Conference — ICML 2026 regular_

### Official Review · Reviewer_YNTa · 2026-02-27

**Soundness:** 3
**Presentation:** 3
**Significance:** 3
**Originality:** 3
**Overall Recommendation:** 5
**Confidence:** 4

**Summary:**

This paper introduces CGMMD (Conditional Generator using Maximum Mean Discrepancy), a framework for learning conditional generative models that enables one-shot sampling from $Y|X$. The method trains a neural generator by directly minimising an empirical conditional MMD objective. The approach combines kernel-based conditional MMD measures with nearest-neighbor–based functionals to estimate  the conditional distribution. The authors establish finite-sample guarantees for the learned generator. Empirical evaluations on multimodal benchmarks and image tasks demonstrate that CGMMD performs well and can capture conditional structures.

**Compliance With Llm Reviewing Policy:**

Affirmed.

**Key Questions For Authors:**

1. The usual concern of high-dimensional settings. The finite-sample bound in Theorem E.4 contains terms scaling as $\left(k_n \log n / n\right)^{1/d}$, which deteriorate with dimension. How does CGMMD scale in moderate or high-dimensional settings? While image outputs are high-dimensional in ambient pixel space, the paper does not include a study on how performance of CGMMD scales with dimension. In particular, it would be helpful to understand how the nearest-neighbor and MMD components behave as ambient dimension increases in synthetic settings.
2. Assumption E.3 requires network depth/width to grow with sample size. Do the theoretical guarantees meaningfully apply to the fixed-size networks used in experiments? Is there a finite-capacity analogue of the convergence result? I don't necessarily need this to increase a score, and perhaps is future work, I am just interested.
3. All experiments use a Gaussian kernel. It may be useful to include ablation to determine how sensitive CGMMD is to kernel choice and bandwidth selection.
4. The paper emphasises one-shot sampling efficiency which is clearly very appealing from a computational efficiency viewpoint. Under comparable budgets, how does the performance of CGMMD compare to some known state-of-the-art conditional diffusion models?

**Limitations:**

See weaknesses section above.

**Strengths And Weaknesses:**

Strengths:
- The paper is well written, with good citations and well defined objects.
- The method enables principled one-shot sampling, requiring only a single forwaed pass at inference time.
- Theoretical contributions include non-asymptotic finite-sample guarantees and convergence in MMD under explicit assumptions.
- The uniform concentration result for nearest-neighbour–based functionals is technically interesting and may be of independent value.

Weaknesses:
- The theoretical convergence rate deteriorates with dimension through terms such as $(k_n \log n / n)^{1/d}$, raising scalability concerns.
- The theory requires network capacity (depth/width) to grow with sample size, limiting direct applicability to fixed architectures used in practice.
- The method relies on nearest-neighboir–based functionals, again raising curse of dimensionality concerns.
- Small point: Kernel sensitivity is usually important when using MMD and is not explored in this paper; all experiments use a Gaussian kernel.
- Comparisons to conditional diffusion models are limited for same computational budget.

---

> ### Author Rebuttal · Authors · 2026-03-31
>
> Thank you for your helpful comments and positive assessment. Below, we respond to your comments and questions individually.
>
> ## Role of intrinsic dimension in CGMMD guarantees
>
> We appreciate the Reviewer’s comment about the role of dimension in the convergence guarantee and performance of CGMMD. We believe that such dependence is inherent to the conditional sampling problem, similar to such fundamental hardness observed in density estimation. Crucially, CGMMD adapts to the intrinsic dimension of $\mathbf{X}$, and the convergence guarantees can be tightened to account for this lower-dimensional structure when it exists. We provide a detailed discussion under **Scope of CGMMD in High-Dimensional Settings** in our response to Reviewer iyjg and refer the Reviewer there for a comprehensive explanation.
>
> ## Scope of CGMMD with fixed network
>
> The reviewer raises an interesting and important question. Recall that Assumption 4.2 (and more generally Assumption E.3) imposes two requirements on the neural network size and architecture. The second requirement remains valid even when the network parameters are fixed, whereas the first requires that either the network depth or width grows with the sample size at an appropriate rate. This condition is primarily used to ensure that the neural network class can approximate the true conditional sampler $\bar{g}$ to arbitrary precision (see [1] for details).
>
> That said, we believe that our proof technique can also be used to show that, when the target conditional sampler $\bar{g}$ is bounded and Lipschitz continuous, for any target accuracy $\varepsilon > 0$ one can choose a network architecture (independent of the sample size but depending on $\varepsilon$) such that at large samples the resulting estimator achieves an $\varepsilon$-level bound on the loss, up to constant factors.
>
> Reference
>
> [1] Shen, Zuowei, Haizhao Yang, and Shijun Zhang. Deep network approximation characterized by number of neurons. arXiv:1906.05497 (2019).
>
> ## Sensitivity of CGMMD to kernel choice and bandwidth selection
>
> We appreciate the reviewer for raising this point. Since the Gaussian kernel is a practical choice used in most real applications, we also made the same choice. That said, we have a discussion on this aspect in Appendix D. Prompted by this suggestion, we have done an experimental study to understand the sensitivity of kernel bandwidth and number of nearest neighbors on the performance of CGMMD. We provide a discussion under **Sensitivity of CGMMD to kernel bandwidth and NN parameters** in our response to Reviewer ePyC and refer the Reviewer there for a comprehensive explanation.
>
>
> ## Comparison of CGMMD with diffusion models under comparable budget
>
> We thank the reviewer for this excellent suggestion. However, designing a rigorous and fair experiment to evaluate the performance for a specific budget presents significant, non-trivial methodological challenges. Specifically, it would require modifying the base architecture to accommodate the same budget for both algorithms, which introduces additional challenges that distract from the core contribution of this paper. As this work represents the foundational introduction of CGMMD, our focus remains on establishing its theoretical guarantees and baseline empirical viability.

---

> > ### Author Rebuttal · Reviewer_YNTa · 2026-04-02
> >
> > Thank you to the authors for their responses and clarifications. I believe these additions will improve the readability and presentation of the paper. My original score stands.

---

> > > ### Author Response · Authors · 2026-04-06
> > >
> > > We thank the reviewer for their positive evaluation and constructive feedback. We will ensure that these discussions are fully integrated into the camera-ready version of the paper.

---

### Official Review · Reviewer_ePyC · 2026-03-10

**Soundness:** 4
**Presentation:** 3
**Significance:** 3
**Originality:** 2
**Overall Recommendation:** 5
**Confidence:** 2

**Summary:**

Authors propose CGMMD, a novel framework for conditional sampling that directly minimizes a Maximum Mean Discrepancy (MMD) objective without relying on adversarial training. Unlike many existing approaches, CGMMD enables one-shot conditional sample generation through a single forward pass of the generator, resulting in low test-time computational cost. They provide rigid theoretical guarantees, including bounds on the sampling loss and convergence of the learned distribution to the true conditional distribution. Empirical results on both synthetic conditional density tasks and real-world applications such as image denoising and super-resolution demonstrate that CGMMD performs competitively with existing methods.

**Compliance With Llm Reviewing Policy:**

Affirmed.

**Final Justification:**

This work presents a well-motivated approach to addressing fundamental limitations of existing generative models, with conditional MMD as its central contribution. The theoretical grounding is solid, and the technical framework developed to realize the core idea is interesting and carefully constructed. While there were a few minor concerns raised during the initial review, the authors' rebuttal was convincing and addressed these points with sufficient clarity. We have therefore raised our score to accept.

**Key Questions For Authors:**

1. It would be helpful to include an analysis of how the performance varies with different kernel bandwidths and nearest-neighbor (NN) parameters. Such results could further strengthen the robustness of the proposed algorithm.
2. In Appendix C.5, the reference to a section appears to be broken.
3. GFMN [1] shows that ImageNet pre-trained network could be served as powerful characteristic kernel instead of well-known kernels. Does this kind of way applied to CGMMD?


[1] Santos, Cicero Nogueira dos, et al. "Learning implicit generative models by matching perceptual features." *Proceedings of the IEEE/CVF International Conference on Computer Vision*. 2019.

**Limitations:**

This work presents a well-motivated approach to addressing several limitations of existing generative models, with conditional MMD. The theoretical grounding is solid, and the technical framework developed to realize the core idea is interesting. While there were a few minor questions raised during the initial review, the authors' rebuttal was convincing and addressed these points with sufficient clarity. I raise score to accept, looking forward to the potential future work as a promising new direction for generative modeling.

**Strengths And Weaknesses:**

Strengths :
1. CGMMD is built on solid theoretical foundations with rigorous convergence guarantees. The integration with a nearest-neighbor–based framework is reasonable.

Weaknesses :
1. The quantitative evaluation is somewhat limited.
2. Baselines appear to be outdated (e.g., WGAN).

---

> ### Author Rebuttal · Authors · 2026-03-31
>
> We thank the reviewer for their thoughtful and constructive feedback. Below, we provide point-by-point responses to the reviewer’s key questions.
>
> ## Scalability of CGMMD in high dimensions
>
> We acknowledge the reviewer’s comment about scalability challenges in presence of high-dimensional covariates $\mathbf{X}$. This dependence reflects a fundamental challenge in conditional sampling, akin to related issues in density estimation. Importantly, CGMMD naturally adapts to the intrinsic dimension of $\mathbf{X}$, and the convergence guarantees can be refined to reflect this lower-dimensional structure when present. We discuss this in detail under **Scope of CGMMD in High-Dimensional Settings** in our response to Reviewer iyjg and refer the Reviewer there for a more comprehensive explanation.
>
> ## Sensitivity of CGMMD to kernel bandwidth and NN parameters
>
> We thank the reviewers ePyC and YNTa for such an insightful suggestion. For expository purposes, we have conducted a simple numerical simulation to examine the sensitivity of our method to kernel bandwidth, and the number of nearest neighbors. We consider the simulation setting where $X \sim N(0, I\_5)$ and, conditional on $X$,
> $$Y = X\_1^2 + \exp\left(\frac{X\_2 + X\_3}{3}\right) + \sin(X\_4 X\_5) + \varepsilon, \quad \varepsilon \sim N(0,1).$$
>
> For evaluation, we fix a representative covariate value $X = (0,0,0,0,0)$, and compare the conditional mean and standard deviation of the learned conditional generator $\hat{P}\_{Y \mid X}$ with the true conditional distribution $P\_{Y \mid X}$. We report the average MSE of the mean and SD estimates in the following table. Since in all other experiments, we worked with the Gaussian kernel $K(y,y') = \exp(-\sigma \|y-y'\|^2)$ we make the same choice here.
>
> **MSE of Conditional Mean estimates**
>
> | $\sigma/k$ | 5 | 10 | 15 | 20 | 25 | 30 | 35 | 40 |
> |-----------|-------|-------|-------|-------|-------|-------|-------|-------|
> | 0.00316 | 0.0098 | 0.0135 | 0.0116 | 0.0046 | 0.0010 | 0.0004 | 0.0004 | 0.0004 |
> | 0.00599 | 0.0127 | 0.0153 | 0.0107 | 0.0029 | 0.0007 | 0.0005 | 0.0005 | 0.0004 |
> | 0.01136 | 0.0145 | 0.0134 | 0.0058 | 0.0010 | 0.0004 | 0.0009 | 0.0011 | 0.0008 |
> | 0.02154 | 0.0161 | 0.0090 | 0.0025 | 0.0004 | 0.0010 | 0.0025 | 0.0031 | 0.0031 |
> | 0.04084 | 0.0242 | 0.0099 | 0.0024 | 0.0006 | 0.0016 | 0.0043 | 0.0064 | 0.0100 |
> | 0.07743 | 0.0230 | 0.0055 | 0.0013 | 0.0021 | 0.0037 | 0.0099 | 0.0208 | 0.0473 |
> | 0.14678 | 0.0060 | 0.0008 | 0.0014 | 0.0075 | 0.0154 | 0.0399 | 0.0932 | 0.2395 |
>
>
>
> **MSE of Conditional SD estimates**
>
> | $\sigma/k$| 5 | 10 | 15 | 20 | 25 | 30 | 35 | 40 |
> |-----------|-------|-------|-------|-------|-------|-------|-------|-------|
> | 0.00316 | 0.3486 | 0.3509 | 0.3519 | 0.2352 | 0.1269 | 0.1303 | 0.1314 | 0.1413 |
> | 0.00599 | 0.3264 | 0.2307 | 0.1264 | 0.0691 | 0.0511 | 0.0512 | 0.0474 | 0.0462 |
> | 0.01136 | 0.2103 | 0.1268 | 0.0961 | 0.0858 | 0.0822 | 0.0702 | 0.0635 | 0.0560 |
> | 0.02154 | 0.1888 | 0.1357 | 0.1025 | 0.1065 | 0.1217 | 0.1184 | 0.1136 | 0.1044 |
> | 0.04084 | 0.0977 | 0.0741 | 0.0488 | 0.0605 | 0.0738 | 0.0948 | 0.1016 | 0.0979 |
> | 0.07743 | 0.0488 | 0.0356 | 0.0285 | 0.0324 | 0.0390 | 0.0525 | 0.0556 | 0.0421 |
> | 0.14678 | 0.0374 | 0.0288 | 0.0260 | 0.0264 | 0.0266 | 0.0191 | 0.0058 | 0.0075 |
>
> We observe that when the bandwidth is small, moderate to large neighborhood sizes achieve lower MSE values of the conditional mean while as bandwidth increases, for the same choices of neighborhood sizes, the MSE worsens. This tends to suggest that we should in fact take small bandwidths. However, for the standard deviation, we see that with small bandwidths, the MSE of SD estimates are quite large, and as bandwidth increases the estimate is much better for moderate to large neighborhood sizes. Together this suggests that a moderate bandwidth choice and a moderate neighborhood size facilitate better estimates of both the conditional mean and the standard deviation.
>
>
> ## ImageNet pre-trained network as a characteristic kernel choice
>
> We thank the reviewer for this interesting question. In Assumption 2.1, the key requirement for CGMMD is that the kernel be characteristic, i.e., it induces a one-to-one embedding of distributions. Proposition 1 in [1] shows that ImageNet-pretrained networks can satisfy this property when their feature space is dense in the space of continuous functions, and Section 3 provides supporting arguments that this condition holds. We therefore agree that such networks could be used as kernels in the CGMMD framework and will add a remark highlighting this promising future direction in the revision.
>
> References:
>
> [1] Santos, Cicero Nogueira dos, et al. "Learning implicit generative models by matching perceptual features." Proceedings of the IEEE/CVF International Conference on Computer Vision. 2019.
>
>
> ## Fixing broken reference in Appendix C.5
>
> Thank you for bringing this to our attention. We will correct it in the updated version.

---

> > ### Author Rebuttal · Reviewer_ePyC · 2026-04-02
> >
> > All of my questions and concerns have been thoroughly addressed by the authors. I find no critical limitations in the proposed CGMMD framework that would warrant rejection. The method is well-motivated, theoretically grounded, and practically effective. I am particularly encouraged by the future potential of CGMMD as a principled and efficient alternative in the landscape of conditional generative modeling. I will update my score to Accept.

---

> > > ### Author Response · Authors · 2026-04-06
> > >
> > > As the discussion period is coming to an end, we would like to thank the Reviewer ePyC for their time and constructive suggestions. We are checking in to ensure we have addressed all the concerns, as we look forward to the **potential score update mentioned previously**. Thank you again for your valuable feedback.

---

### Official Review · Reviewer_gmCe · 2026-03-13

**Soundness:** 3
**Presentation:** 2
**Significance:** 3
**Originality:** 3
**Overall Recommendation:** 4
**Confidence:** 2

**Summary:**

This paper proposes CGMMD, a conditional generative framework that learns a sampler $g(\eta,x)$ by starting from the noise outsourcing lemma, which represents the target conditional law $Y\mid X=x$ through a measurable generator driven by auxiliary noise. The key idea is then to measure the discrepancy between the true conditional distribution and the model-induced conditional distribution using ECMMD, the expected conditional maximum mean discrepancy. To make this objective usable, the authors first rewrite ECMMD in terms of RKHS kernel mean embeddings, and then further express it as an expectation of a tractable kernel-based quantity $H(W,W')$. Because in practice one typically observes only one response $Y_i$ for each covariate $X_i$, they estimate this conditional discrepancy using a k-nearest-neighbor graph on the $X$-samples, averaging the $H$-terms over nearby covariates to form an empirical loss. The final estimator is obtained by minimizing this empirical ECMMD objective over a neural-network generator class. The paper also provides a non-asymptotic error bound that decomposes into nearest-neighbor estimation error, generalization error, and neural-network approximation error, and proves that the learned conditional distribution converges to the true one in MMD. Empirically, CGMMD performs competitively on synthetic conditional sampling tasks and several image restoration problems, while retaining the practical advantage of fast one-shot generation.

**Compliance With Llm Reviewing Policy:**

Affirmed.

**Final Justification:**

I lean toward accepting this paper. I do not have specific concerns that will affect my score during the rebuttal process. So I still keep my original evaluation.

**Key Questions For Authors:**

I am currently inclined to accept this paper, and I do not have any key questions at this point that would affect my decision.

**Limitations:**

See weaknesses above.

**Strengths And Weaknesses:**

**Strengths**

* This paper presents a fairly complete work: the method is clearly motivated, the theoretical development is substantial and complete to me. Although CGMMD may not outperform state-of-the-art generative models in sample quality, its one-shot sampling efficiency is a meaningful advantage and makes the method useful in some cases.

**Weaknesses**

* The presentation of the paper could be improved. In particular, some of the technical background is split between the introduction and the technical background section, which makes the exposition feel somewhat repetitive. A more concentrated and streamlined presentation of the preliminaries would likely make the paper easier to follow.

* The theoretical guarantees are interesting, but the main error bound has an explicit dependence on the covariate dimension $d$, which suggests a potential curse-of-dimensionality issue. I understand this might comes from the behavior of KNN, but still it suggests the method may fail to scale to high dimension.

* While the method enjoys fast test-time sampling, its training objective is less lightweight than standard pointwise losses. In each mini-batch, CGMMD requires constructing a $k$-NN graph over the conditioning variables $X_i$, i.e., computing distances between covariates and identifying neighbors before evaluating the loss. This can become computationally burdensome for large batches, and for high-dimensional conditioning inputs such as images, it is also unclear how reliable nearest-neighbor structure in the raw covariate space is.

---

> ### Author Rebuttal · Authors · 2026-03-31
>
> Thank you for your positive assessment and constructive suggestions. Below, we respond in detail to the comments.
>
> ## Improved presentation of technical backgrounds
>
> We thank the reviewer for this helpful suggestion. We will revise the exposition to consolidate the technical background and streamline the presentation for improved clarity.
>
> ##  Curse of dimensionality on CGMMD
>
> We acknowledge the Reviewer’s concern about the explicit dependence of convergence guarantees on dimension $d$ of the conditioning variable $\mathbf{X}$. We believe this to be a fundamental hardness of the conditional sampling problem similar to related problems in density estimation. However, the proposed CGMMD adapts to the intrinsic dimension of $\mathbf{X}$ and the convergence guarantees can be improved to reflect this dependence whenever $\mathbf{X}$ lies on a lower dimensional structure. We provide a detailed discussion of this point in **Scope of CGMMD in High-Dimensional Settings** in our response to Reviewer iyjg, and we refer the Reviewer there for a more comprehensive explanation.
>
> ## Cost of KNN computation within mini-batch
>
> We thank the reviewer for raising this important point regarding the computational cost of constructing the $k$-NN graph during training. We agree that the $k$-NN computation within each mini-batch introduces additional overhead compared to standard pointwise objectives. Our primary goal in this work, however, is to design a method that achieves fast and efficient test-time sampling, and we accept a moderate increase in training complexity as a trade-off for this benefit.
>
> That said, this overhead can be mitigated in practice. While exact $k$-NN construction has $O(B^2)$ complexity for batch size $B$, a large body of work on approximate nearest-neighbor (ANN) methods enables significantly faster computation, often with near-linear or $O(B \log B)$ complexity, while preserving neighborhood structure to high accuracy (see Appendix D). Such approaches are standard in large-scale learning and can be readily incorporated into our framework.
>
> ## Alternative to computing kNN graph on the raw feature space
>
> We understand the reviewer’s concern with computing kNN graph with high-dimensional conditioning variables (e.g., images), we agree that nearest-neighbor structure in the raw input space may be unreliable and computationally expensive. In such settings, a standard and effective approach is to first map the inputs into a lower-dimensional representation using a learned embedding (e.g., via a ResNet encoder; see for example [1,2]), and then construct the $k$-NN graph in this feature space. This is well-motivated by the observation that high-dimensional data often lies near a lower-dimensional manifold, making neighborhood relationships more meaningful after representation learning. In fact, we follow exactly this strategy in our STL10 experiment, where the $k$-NN graph is constructed in a learned embedding space rather than the raw image space. Empirically, this leads to stable neighborhood structure and good downstream performance (e.g., high-quality reconstructed samples), suggesting that the approach is effective in practice.
>
> [1] Ren, Y., Zhu, J., Li, J., & Luo, Y. (2016). Conditional generative moment-matching networks. Advances in Neural Information Processing Systems, 29
>
> [2] Li, Yujia, Kevin Swersky, and Rich Zemel. "Generative moment matching networks." In International conference on machine learning, pp. 1718-1727. PMLR, 2015.

---

> > ### Author Rebuttal · Reviewer_gmCe · 2026-04-03
> >
> > I thank the authors for their detailed response. I have no further concerns and I will keep my positive score.

---

> > > ### Author Response · Authors · 2026-04-06
> > >
> > > We are pleased that our responses clarified the reviewer’s concerns. We will ensure that these clarifications and the resulting improvements are fully incorporated into the camera-ready version of the manuscript.

---

### Official Review · Reviewer_iyjg · 2026-03-16

**Soundness:** 3
**Presentation:** 3
**Significance:** 3
**Originality:** 3
**Overall Recommendation:** 4
**Confidence:** 3

**Summary:**

This paper proposes CGMMD (Conditional Generator using Maximum Mean Discrepancy), a framework that learns a conditional generator by directly minimizing an empirical estimate of the ECMMD between the true conditional distribution and the generator's conditional distribution. The key idea is to use a nearest-neighbor-based estimator of ECMMD. On the theory side, the paper provides finite-sample error bounds and convergence guarantees for the learned conditional sampler under some assumptions. Empirically, it demonstrates the good performance on synthetic conditional-density examples, image denoising, super-resolution, inpainting, and a small comparison against diffusion-based conditional generation.

**Compliance With Llm Reviewing Policy:**

Affirmed.

**Key Questions For Authors:**

please refer to above

**Limitations:**

This paper don't include the limitation part. I suggest authors to include potential weakness in high dimensional setup in the manuscript

**Strengths And Weaknesses:**

This paper is well written and easy to follow. The paper provides non-asymptotic bounds, consistency results, and convergence in MMD which seems fine to me and more rigorous than many empirical conditional generation work. It contains intensive theoretical results showing that this method is mathematically guaranteed to convergence under mild assumptions. Although I acknowledge the profound theoretical work in this paper, empirical evaluations are concerning. The paper makes fairly strong claims about conditional sampling, but the experiments are mostly small-scale toy problems using dataset such as MNIST and Fashion MNIST. Also, modern conditional diffusion or flow-based methods on realistic benchmarks should be included as baseline method. Also, while it shows better efficiency compared to diffusion model, when I see the comparison with distilled diffusion, its efficiency gain gets less stark and still shows worse generation quality than distilled diffusion. I understand this but also I doubt the utility of this model in the real world. Also, nearest-neighbor methods scale poorly in high dimensions. In terms of novelty, I think this paper is novel enough to be accepted.

---

> ### Author Rebuttal · Authors · 2026-03-31
>
> Thank you for a positive assessment and sharing your valuable comments. Below we give our detailed response.
>
> ## Scope of CGMMD in high-dimensional setup
>
> We agree that our convergence guarantees depend on the dimension $d$ of the conditioning variable $\mathbf{X}$. As noted by Reviewer YNTa, this dependence is common in high-dimensional settings and we believe this reflects an inherent hardness of conditional sampling rather than a limitation of our method. Similar dimensional dependence also appears in conditional sampling with Wasserstein GANs [1]. More broadly, the curse of dimensionality is widely regarded as a fundamental hardness in related problems, including classical density estimation [2], conditional density estimation [3,4], and conditional mean estimation [2]. Importantly, CGMMD adapts to the **intrinsic dimension** of the conditioning variable. As suggested by the reviewers we will add the following discussion in the revision.
>
> Let $B(t)$ denote the ball of radius $t$ centered at the origin. We say that $X$ has intrinsic dimension at most $d\_{\mathrm{int}}$ with constant $C>0$ if for all $t>0$ and all $\varepsilon\in(0,t]$, $B(t)\cap \operatorname{support}(X)$ can be covered by at most $C(t/\varepsilon)^{d\_{\rm int}}$ closed balls of radius $\varepsilon$.
>
> This notion has been studied in [5,6] and is related to the *Assouad dimension*. In particular, if $X$ is supported on a $d\_0$-dimensional hyperplane ($d\_0 \le d$), then $X$ has intrinsic dimension $d\_{\rm int}\le d\_0$, and this also extends to lower-dimensional manifolds [6]. It can easily be shown that our convergence guarantees depend on $d\_{\rm int}$ rather than $d$ yielding improved guarantees when $\mathbf{X}$ lies on a lower-dimensional structure. Under the tenet of manifold hypothesis [7], many high-dimensional datasets lie near low-dimensional manifolds, making this adaptation practically important.
>
> In the updated version we will also add an empirical validation for the adaptation of CGMMD to intrinsic dimension.
>
> References:
>
> [1] Song, Shanshan, Tong Wang, Guohao Shen, Yuanyuan Lin, and Jian Huang. "Wasserstein generative regression." Journal of the Royal Statistical Society Series B: Statistical Methodology 88, no. 1 (2026): 330-351.
>
> [2] Tsybakov, Alexandre B. "Nonparametric estimators." In Introduction to Nonparametric Estimation, pp. 1-76. New York, NY: Springer New York, 2008.
>
> [3] Blair Bilodeau, Dylan J. Foster, Daniel M. Roy "Minimax rates for conditional density estimation via empirical entropy," Annals of Statistics 51(2), 762-790, (April 2023).
>
> [4] Li, Michael, Matey Neykov, and Sivaraman Balakrishnan. "Minimax optimal conditional density estimation under total variation smoothness." Electronic Journal of Statistics 16, no. 2 (2022): 3937-3972.
>
> [5] Deb, Nabarun, Promit Ghosal, and Bodhisattva Sen. "Measuring association on topological spaces using kernels and geometric graphs." arXiv preprint arXiv:2010.01768 (2020).
>
> [6] Huang, Zhen, Nabarun Deb, and Bodhisattva Sen. "Kernel Partial Correlation Coefficient---a Measure of Conditional Dependence." Journal of Machine Learning Research 23, no. 216 (2022): 1-58.
>
> [7] Whiteley, Nick, Annie Gray, and Patrick Rubin-Delanchy. "Statistical exploration of the manifold hypothesis." Journal of the Royal Statistical Society Series B: Statistical Methodology (2026): qkag055.
>
>
> ## Remark on empirical performance of CGMMD
> The primary goal of our work is to introduce the ECMMD loss as a principled objective for learning generative models, which has not been explicitly explored before. Our theoretical guarantees are intended to support and motivate its use in this setting. We agree that, in its current form, the method may not be able to compete directly with highly optimized generative models from the literature. Rather, we view this work as a first step towards adapting such loss functions with modern generative modeling techniques laying the groundwork for integrating more powerful methods in future.
>
> In this context, we also understand the reviewer’s concern regarding comparisons with distilled diffusion models. However, our focus here is on introducing CGMMD as a one-shot conditional generation framework, along with its theoretical grounding and initial empirical validation to demonstrate its flexibility. The goal is not to achieve comparable efficiency or benchmark against highly optimised diffusion-based methods at this stage. Extending our approach to improve efficiency and conducting thorough comparisons with advanced diffusion models and fast samplers are natural next steps, and form a key part of our planned future research.

---

> > ### Author Rebuttal · Reviewer_iyjg · 2026-04-03
> >
> > I appreciate the authors' response. I am happy to keep my score

---

> > > ### Author Response · Authors · 2026-04-06
> > >
> > > Thanks again for your feedbacks. We will definitely incorporate all your suggestions in the camera-ready version of the paper.

---

### Decision · Program_Chairs · 2026-04-30

**Decision:**

Accept (regular)

**Comment:**

This paper proposes a novel analysis extending MMD-based generative modeling to the conditional setting. The reviewers agree that the paper is interesting and that the theoretical developments are worthy of being disseminated. However, the reviewers also point to some limitations in the poor scaling with respect to the input dimension, and question the significance of the work in light of recent developments in generative modeling. The authors should improve the experimental paper and clarify that it serves as an illustration of the theoretical developments and is not actually SOTA.